# Intent-Aware Planning in Heterogeneous Traffic via Distributed Multi-Agent Reinforcement Learning

**Xiyang Wu**
University of Maryland
wuxiyang@umd.edu

**Rohan Chandra**
University of Texas at Austin
rchandra@utexas.edu

**Tianrui Guan**
University of Maryland
rayguan@umd.edu

**Amrit Singh Bedi**
University of Maryland
amritbd@umd.edu

**Dinesh Manocha**
University of Maryland
dmanocha@umd.edu

**Abstract:** Navigating safely and efficiently in dense and heterogeneous traffic scenarios is challenging for autonomous vehicles (AVs) due to their inability to infer the behaviors or intentions of nearby drivers. In this work, we introduce a distributed multi-agent reinforcement learning (MARL) algorithm for joint trajectory and intent prediction for autonomous vehicles in dense and heterogeneous environments. Our approach for intent-aware planning, iPLAN, allows agents to infer nearby drivers' intents solely from their local observations. We model an explicit representation of agents' private *incentives*: *Behavioral Incentive* for high-level decision-making strategy that sets planning sub-goals and *Instant Incentive* for low-level motion planning to execute sub-goals. Our approach enables agents to infer their opponents' behavior incentives and integrate this inferred information into their decision-making and motion-planning processes. We perform experiments on two simulation environments, Non-Cooperative Navigation and Heterogeneous Highway. In Heterogeneous Highway, results show that, compared with centralized training decentralized execution (CTDE) MARL baselines such as QMIX and MAPPO, our method yields a $4.3\%$ and $38.4\%$ higher episodic reward in *mild* and *chaotic* traffic, with $48.1\%$ higher success rate and $80.6\%$ longer survival time in *chaotic* traffic. We also compare with a decentralized training decentralized execution (DTDE) baseline IPPO and demonstrate a higher episodic reward of $12.7\%$ and $6.3\%$ in *mild* traffic and *chaotic* traffic, $25.3\%$ higher success rate, and $13.7\%$ longer survival time.

**Keywords:** Autonomous Driving, Multi-agent Reinforcement Learning, Representation Learning

## 1 Introduction

In this work, we consider the task of trajectory planning for autonomous vehicles in dense and heterogeneous traffic. High density is typically measured in the number of vehicles per square meter and high heterogeneity refers to a large variance in agents' driving styles ranging from aggressive to conservative, vehicle dynamics, and vehicle types [1]. For example, these agents may include two-wheelers, cars, buses, and trucks. The key challenge to efficient trajectory planning in such environments is to be able to accurately infer the behavior of these heterogeneous agents [2]. Therefore, many solutions perform trajectory planning by jointly predicting the agents' future *trajectories* along with their *intent* [3].

Trajectory prediction is the task of predicting the future states of an agent [4] which typically consists of spatial coordinates, and heading angle, but may also include first-order information such as

7th Conference on Robot Learning (CoRL 2023), Atlanta, USA.

velocity. Intent prediction focuses on inferring neighboring agents' behavior using local information [5]. In the context of autonomous driving, some studies have approached intent prediction by classifying driving behaviors into predefined classes [6, 2] such as aggressive or conservative.

Although many methods for joint trajectory and intent prediction [7, 8, 3, 5] have been extensively studied for planning in both industry and academia, most of the existing approaches are trained and evaluated on datasets like The Waymo Open Motion Dataset [9] and the NuScenes dataset [10], which primarily consist of homogeneous traffic and lack variation in driver behavior [3]. As a result, these methods [7, 8, 3, 5] often struggle to reliably predict the intentions of heterogeneous agents in unstructured and dense traffic [11].

On the other hand, simulators such as CARLA are designed to generate traffic agents with diverse, kinodynamically feasible behaviors [12], addressing the lack of diverse behavior in datasets. Most of the joint trajectory and intent prediction methods evaluated on the datasets discussed above can be used with such simulators [13, 4]. But these methods typically require generating and collecting data in offline storage, which defeats the purpose of a simulator [14]. Complementary to these offline approaches, simulators [12] also offers the capability to model multiple agents and their interactions simultaneously via multi-agent reinforcement learning (MARL), where the learning algorithm can engage with the simulation environment [15]. MARL has demonstrated remarkable success in many different multi-agent domains such as Go [16], chess [17], poker [18], Dota2 [19], and StarCraft [20]. However, their applicability to autonomous driving has been relatively sparse [21].

Deep MARL for trajectory planning in autonomous driving only recently achieved significant momentum with the Highway-Env environment [22] proposed in the author's doctoral thesis [23]. Since then, several deep MARL approaches have been proposed [24, 25] for trajectory planning, but these methods do not extend to heterogeneous traffic and also assume agents can communicate and share information with each other. To the best of our knowledge, there is no prior decentralized training decentralized execution (DTDE) MARL approach for joint intent and trajectory prediction for AVs in heterogeneous traffic. More related works are included in Appendix B.

**Main Contributions:** In this paper, we propose a new intent-aware trajectory planning algorithm for autonomous driving in dense and heterogeneous traffic environments. We cast the autonomous driving problem as a hidden parameter partially observable stochastic game (HiP-POSG) [26, 27] and solve it using a DTDE MARL framework, called iPLAN, built around a joint intent and trajectory prediction encoder-decoder architecture. Given the current traffic conditions and historical observations, iPLAN computes the optimal multi-agent policy for each agent in the environment, relying solely on local observations without weight-sharing or communication.

Our main contributions include:

1. To the best of our knowledge, we propose the first DTDE MARL algorithm for joint trajectory and intent prediction for autonomous vehicles in dense and heterogeneous environments. Our algorithm is fully decentralized without weight sharing, communication, or centralized critics, and can handle variable agents across episodes.

2. We model an explicit representation of agents' private incentives that include $(i)$ *Behavioral Incentive* for high-level decision-making strategy that sets planning sub-goals and $(ii)$ *Instant Incentive* for low-level motion planning to execute sub-goals. These incentives enable behavior-aware motion forecasting, which is more suited for heterogeneous traffic.

3. We perform experiments on two simulation environments, Non-Cooperative Navigation [28] and Heterogeneous Highway [22]. The results show that, compared to centralized training decentralized execution (CTDE) MARL baselines like QMIX and MAPPO, our method yields a $4.3\%$ and $38.4\%$ higher episodic reward in *mild* and *chaotic* traffic and is $48.1\%$ more successful with an $80.6\%$ longer survival time in *chaotic* traffic in Heterogeneous Highway. Compared to the DTDE baseline IPPO, we demonstrate a higher episodic reward of $12.7\%$ and $6.3\%$ in *mild* and *chaotic* traffic, a $25.3\%$ higher success rate, and $13.7\%$ longer survival time in the Heterogeneous Highway.

## 2   Problem Formulation

**Problem Setting and Assumptions:** We consider a multi-agent scenario with $N \geq 2$ non-cooperative agents [29], *i.e.*, agents are controlled by individual policies that maximize their own reward without weight sharing or communication. In each episode, agents interact with one another and gain general experience without any prior knowledge about a specific agent from previous episodes. Agents' strategies remain the same within one episode, though strategies may evolve between episodes. We assume that all agents are driven by motivations behind their actions. These motivations can arise from instantaneous reactions to environmental changes or more enduring preferences. We denote them as *incentives* for agents' strategies. While these incentives are private and not explicitly known to other agents, they can be discerned through observing agents' strategies that offer insights into the incentives behind agents' actions. In this work, we explicitly model these private incentives with hidden parameters representing latent states. Therefore, we formulate this problem as a multi-agent hidden parameter partially observable stochastic game [30], or HiP-POSG[1].

**Task and objective:** We consider the tuple

$$\left\langle N, \mathcal{S}, \{\mathcal{A}_i\}_{i=1}^N, \{\mathcal{O}_i\}_{i=1}^N, \{\Omega_i\}_{i=1}^N, \{\mathcal{Z}_i\}_{i=1}^N, \{f_i\}_{i=1}^N, \mathcal{T}, \{r_i\}_{i=1}^N, \gamma \right\rangle, \tag{1}$$

where $N$ is the number of agents. $\mathcal{S}$ is the set of states. $\mathcal{A}_i$ is the set of actions for agent $i$. $\mathcal{O}_i$ is the observation set of agent $i$ of the global state $S \in \mathcal{S}$, generated by agent $i$'s observation function $\Omega_i : \mathcal{S} \to \mathcal{O}_i$. In our problem, agent $i$'s observation $\mathbf{o}_i^t$ at time $t$ could be further specified as $\mathbf{o}_i^t = \{o_{i,j}^t\}_{j \in \mathcal{N}_i}$, where $\mathcal{N}_i$ refers to the set of agents $j$ in the neighborhood of $i$. The bold $\mathbf{o}_i^t$ denotes the set of agent $i$'s observation of its neighbors at time $t$. We denote the sequence of agent $i$'s historical observations $o_{i,j}$ of opponent $j$ up to time $t$ as $h_{i,j}^t = \{o_{i,j}^k\}_{k=1}^t$. The bold $\mathbf{h}_i^t = \{\mathbf{o}_i^k\}_{k=1}^t$ denotes agent $i$'s observation history of its neighbors. Here, we indicate that agent $i$'s observation history of agent $j$ only consists of its observation of agent $j$'s states, while agent $j$'s actions and rewards are unobservable information by others. $\mathcal{Z}_i$ denotes the latent state space that represents the *incentive* of agent $i$'s strategy. $f_i : \mathcal{O}_i^1 \times \mathcal{O}_i^2 \times \ldots \times \mathcal{O}_i^t \times \mathcal{Z}_j \to \mathcal{Z}_j$ is agent $i$'s incentive inference function that makes an estimation $\hat{z}_{i,j}$ of its opponent $j$'s actual incentive $z_j$ from its observation history of opponent $h_{i,j}^t$ up to time $t$ and its past estimation of $z_j$. Here, we assume agent $i$'s estimations of agent $j$'s incentive $\hat{z}_{i,j}$ belongs to the same latent state space $\mathcal{Z}_j$ as agent $j$'s actual incentive $z_j$. $\mathcal{T} : \mathcal{S} \times \mathcal{A}_1 \times \mathcal{A}_2 \times \ldots \times \mathcal{A}_N \to \Delta(\mathcal{S})$ is the (stochastic) transition matrix between global states. $r_i : \mathcal{S} \times \mathcal{A}_1 \times \mathcal{A}_2 \times \ldots \times \mathcal{A}_N \to \mathbb{R}$ is the reward function for agent $i$. $\gamma$ is the reward discount factor. Agent $i$ decides its action $a_i \in \mathcal{A}_i$ with policy $\pi_i : \mathcal{O}_1^t \times \mathcal{O}_2^t \times \ldots \times \mathcal{O}_N^t \times \mathcal{Z}_1 \times \mathcal{Z}_2 \times \ldots \times \mathcal{Z}_N \to \Delta(\mathcal{A}_i)$ with its observations $\mathbf{o}_i^t$, own incentive $z_i$, and estimated opponents' incentives $\hat{z}_{i,j}^t$ at time $t$.

The objective of agent $i$ is to find the optimal policy $\pi_i^*$, maximizing its $\gamma$-discounted cumulative rewards over an episode of length $T$. The objective equation is given by

$$\pi_i^* = \arg\max_{\pi_i} \mathbb{E}_{\pi_i} \left[ \sum_{t=1}^T \gamma^t r_i \left( s^t, \{a_i^t\}_{i=1}^N \right) \right] \tag{2}$$

where $r_i$ is the reward function of agent $i$.

**Incentive Latent Representation.** In this work, we assume that agents' actions are motivated by $(i)$ long-term planning tied to an agent's driving behavior or personality and $(ii)$ short-term collision avoidance related to the current traffic state. To this end, we decouple agent $i$'s incentive $z_i$ into a vector $z_i = \{\beta_i, \zeta_i\}$. Our formulation is related to the task and motion planning literature [31] where the behavior incentive follows a high-level decision-making strategy with the goal of setting planning sub-goals whereas the instant incentive refers to the low-level motion planning with the goal of executing the sub-goals. The behavior incentive biases the motion forecasting in a behavior-aware manner such that it is better suited for heterogeneous traffic.

---

[1]an extension of the HiP-POMDP [26, 27]

**Behavioral Incentive** $\beta_i$ models drivers' driving styles which are deeply rooted in their *personalities* [32]. Given the observations for the previous few seconds, behavior incentive performs high-level decision-making and plans actions, or sub-goals, and asks, "*What's the most likely action of this driver to take next?*". The answer is encoded via $\hat{\beta}_i^t$. This tells an agent whether it should speed up in empty traffic or slow down in dense traffic. It also is able to recognize conservative drivers and the possible need to overtake. Therefore, this incentive is able to reason between aggressive and conservative drivers.

**Instant Incentive** $\zeta_i$ signifies drivers' instantaneous responses to proximate traffic, taking into account the positions and speeds of neighboring vehicles. Instant incentive then asks, "*How should I execute this sub-goal/high-level action/plan using my controller so that I'm safe and still on track towards my goal?*". Instant incentive measures classical efficiency metrics defined in robotics literature such as collision avoidance (safety), distance from goal, and smoothness.

**Incentive Inference** To cater to two different incentives, we split agent $i$'s incentive inference function $f_i$ into two distinct functions, $f_{i,\beta}$ and $f_{i,\zeta}$: $\hat{\beta}_{i,j}^t \sim f_{i,\beta}(\cdot|h_{i,j}^t, \hat{\beta}_{i,j}^{t-1})$ uses agent $i$'s historical observation $h_{i,j}^t$ of opponent $j$ up to time $t$ and its previous estimation of opponent $j$'s behavioral incentive $\hat{\beta}_{i,j}^{t-1}$ to estimate opponent $j$'s new behavioral incentive $\hat{\beta}_{i,j}^t$ at time $t$. $\hat{\zeta}_{i,j}^t \sim f_{i,\zeta}(\cdot|o_{i,j}^t, \hat{\beta}_{i,j}^t, \hat{\zeta}_{i,j}^{t-1})$ uses agent $i$'s observation $o_{i,j}^t$ of opponent $j$ at time $t$, its current estimation over opponent $j$'s behavioral incentive $\hat{\beta}_{i,j}^t$ and its previous estimation of opponent $j$'s instant incentive $\hat{\zeta}_{i,j}^{t-1}$ to estimate opponent $j$'s new instant incentive $\hat{\zeta}_{i,j}^t$ at time $t$. With the estimation of opponents' incentives, agent $i$'s policy $a_i^t \sim \pi(\cdot|\mathbf{o}_i^t, \hat{\boldsymbol{\beta}}_i^t, \hat{\boldsymbol{\zeta}}_i^t)$ decides its action $a_i^t$ with its local observation, ego incentive, and estimations over opponents' incentives. Here, $\hat{\boldsymbol{\beta}}_i^t$ denotes the combination of agent $i$'s behavioral incentive $\beta_i$ and its estimations over all its opponent agents' behavioral incentives $\{\hat{\beta}_{i,j}^t\}_{j=1,j\neq i}^N$ at time $t$. $\hat{\boldsymbol{\zeta}}_i^t$ denotes the combination of agent $i$'s instant incentive $\zeta_i$ and its estimations over all its opponent agents' instant incentives $\{\hat{\zeta}_{i,j}^t\}_{j=1,j\neq i}^N$ at time $t$.

## 3   iPLAN: Methodology

We demonstrate the overall architecture of our proposed framework in Figure 1. Agents interact with the environment with continuous state space $\mathcal{S}$. Here, we denote that an agent's state includes its ID, current position, and current velocity. An agent's observation includes the states of its neighbors within its observation scope. An agent $i$ records its historical observations of its opponents' states for incentive inference. With historical observations $h_{i,j}^t$, and intermediate observations $\mathbf{o}_i^t$, agent $i$ estimates opponent $j$'s behavioral incentive $\beta_j$ and instant incentive $\zeta_j$. The controller of agent $i$ decides action $a_i^t$ based on its local observation $\mathbf{o}_i^t$, ego, and opponents' estimated behavioral incentives $\hat{\boldsymbol{\beta}}_i^t$, and instant incentives $\hat{\boldsymbol{\zeta}}_i^t$. The action space $\mathcal{A}$ of the environment is discrete and consists of the following high-level actions: {*lane left*, *idle*, *lane right*, *faster*, *slower*} in our Heterogeneous Highway environment, or {*idle*, *up*, *down*, *left*, *right*} in our Non-cooperative Navigation environment (details in Appendix C), while a low-level motion controller (*e.g.*, IDM model [33]) converts the high-level actions into a sequence of $x, y$ coordinates.

### 3.1   Behavioral Incentive Inference

The behavioral incentive inference module intends to estimate opponents' behavioral incentives by generating latent representations from their historical states. At time step $t$, agent $i$ queries a sequence of historical observations $h_{ij}^t$ for opponent $j$ from its observation history profile as the input of the behavioral incentive inference module. For ease of computing, we truncate the full historical interaction sequence into a fixed-length sequence that includes the observation history from the previous $t_h$ steps. We introduce an encoder $\mathcal{E}_i$ to update opponents' behavioral incentive estimation and a decoder $\mathcal{D}_i$ to predict opponents' state sequences in the next $t_h$ steps with current historical observations and behavioral incentive estimation. In practice, we parameterize encoder $\mathcal{E}_i$

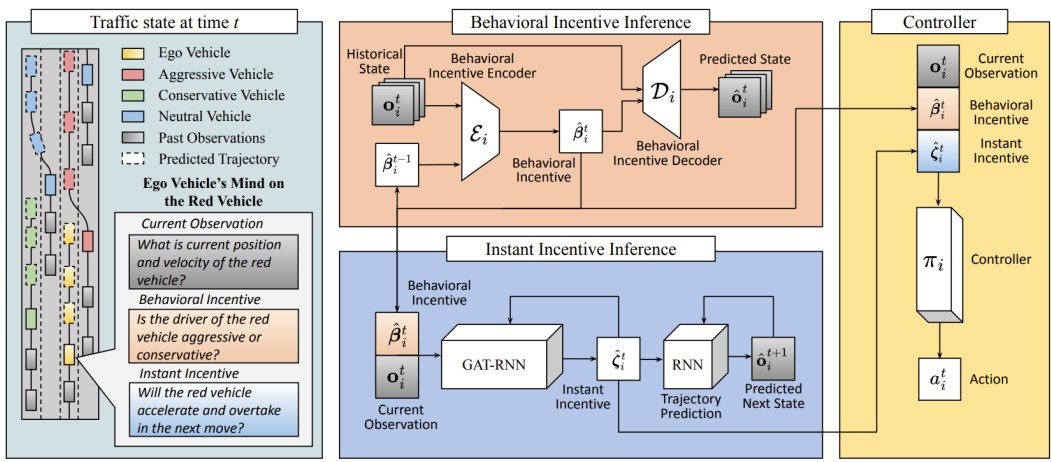

Figure 1: **Intent-aware planning in heterogeneous traffic:** At time $t$, we show current vehicle states in solid colors: ego vehicles $i$ (solid yellow vehicle), aggressive vehicles (solid red), conservative vehicles (solid green), and neutral vehicles (solid blue). The future states of each vehicle are shown with dotted colors. At time step $t$, the ego-agent observes nearby vehicles and infers their behavioral and instant incentives. The behavioral incentive inference (red block) uses agent $i$'s historical observations $\mathbf{h}_i^t$ of other vehicle states (stacked gray boxes of current observations, $\mathbf{o}_i^t$) to infer their behavioral incentives and predict future state sequences with behavioral incentive inferences. The instant incentive inference (blue block) uses agent $i$'s current observations $\mathbf{o}_i^t$ (single gray box) and its inference of others' behavioral incentives $\hat{\boldsymbol{\beta}}_i^t$ (single red box) to infer other vehicles' instant incentives $\hat{\boldsymbol{\zeta}}_i^t$ for trajectory prediction. Agent $i$'s controller (yellow block) selects its action $a_i^t$ with its current observations $\mathbf{o}_i^t$ (gray) and its inference of others' behavioral incentives $\hat{\boldsymbol{\beta}}_i^t$ (red) and instant incentives $\hat{\boldsymbol{\zeta}}_i^t$ (blue).

with $\theta_{\mathcal{E}_i}$, and decoder $\mathcal{D}_i$ with $\theta_{\mathcal{D}_i}$. Hence, the encoder $\mathcal{E}_i$ approximates the behavioral incentive inference function $\hat{\beta}_{i,j}^t \sim f_\beta(\cdot | h_{i,j}^t, \hat{\beta}_{i,j}^{t-1})$.

To capture the sequential nature within opponents' state observation sequences, the encoder $\mathcal{E}_i$ employs a recurrent network that processes $h_{ij}^t$ as a time series. This produces a new estimate of the behavioral incentive of opponent $j$. As insights from cognitive science suggest, the human social focus remains relatively stable [34]. Thus, we interpret the behavioral incentive inference for opponents as a gradual process, converging towards the true behavioral incentives of opponents without abrupt transitions between updates. Starting with an initial neutral estimation of opponents' behavioral latent states, agents propose new estimates for opponents' behavioral incentives at each time step. However, they employ a gentle update strategy, using an additional coefficient $\eta$, to refine the behavioral incentive estimates. This approach allows agents to produce more accurate estimates of opponents' behavioral incentives, managing the variability between consecutive updates, which in turn ensures more stable agent policies.

$$\hat{\beta}_{i,j}^t = \eta \mathcal{E}_i(h_{i,j}^t, \hat{\beta}_{i,j}^{t-1}) + (1 - \eta)\hat{\beta}_{i,j}^{t-1}. \tag{3}$$

The decoder $\mathcal{D}_i$ uses another recurrent network that concatenates agent $i$'s historical observations $h_{ij}^t$ of opponent $j$ with its current behavioral incentive estimation $\hat{\beta}_{ij}^t$. The output is the predicted state sequence $\hat{h}_{i,j}^{t+t_h}$ of opponent $j$ from $t$ to $t + t_h$. We train our encoder and decoder with behavioral incentive inference loss $\mathcal{J}_{\beta_i}$, given by an average L1-norm error between the predicted state sequence $\hat{h}_{i,j}^{t+t_h} = \mathcal{D}_i(h_{i,j}^t, \hat{\beta}_{i,j}^t)$ and the ground truth $h_{i,j}^{t+t_h}$.

$$\mathcal{J}_{\beta_i} = \min_{\mathcal{E}_i, \mathcal{D}_i} \frac{1}{N t_h} \sum_{j=1}^{N} \left\| \mathcal{D}_i(h_{i,j}^t, \hat{\beta}_{i,j}^t) - h_{i,j}^{t+t_h} \right\|_1. \tag{4}$$

## 3.2 Instant Incentive Inference for Trajectory Prediction

The instant incentive inference module intends to estimate opponents' instant incentives from current observations of surrounding agents and their behaviors, which is used for trajectory prediction. Similar to the behavioral incentive inference, we introduce another encoder-decoder structure with encoder $\phi_i$ parameterized by $\theta_{\phi_i}$ and decoder $\psi_i$ parameterized by $\theta_{\psi_i}$. The encoder $\phi_i$ approximates the instant incentive inference function $\hat{\zeta}_{i,j}^t \sim f_{i,\zeta}(\cdot|o_{i,j}^t, \hat{\beta}_{i,j}^t, \hat{\zeta}_{i,j}^{t-1})$ from agent $i$'s current observations $\mathbf{o}_i^t$ of agent $i$, current behavioral incentive estimations $\hat{\boldsymbol{\beta}}_i^t$, and previous instant incentive estimations $\hat{\boldsymbol{\zeta}}_i^{t-1}$. The instant latent state encoder $\phi_i$ uses a sequential structure with two networks. The first network is a Graph Attention Network (GAT) [35]. For agent $i$, GAT reads its observation $\mathbf{o}_i^t$ at time $t$ and the current behavioral incentive estimation $\hat{\boldsymbol{\beta}}_i^t$. The output of GAT is fed to an undirected graph $\mathcal{G}_i^t$ that represents instantaneous interactions among agents at time $t$. Every node in $\mathcal{G}_i^t$ represents an agent in the environment, while the attention weight over the edge between node $i$ and node $j$ encodes the interaction between agent $i$ and $j$ with its relative importance. The second part of the encoder $\phi_i$ is a recurrent neural network (RNN) to extract the temporal information from interaction history. The RNN uses the graphical representation $\mathcal{G}_i^t$ of interactions as the input and previous instant incentive estimation $\hat{\boldsymbol{\zeta}}_i^{t-1}$ as the hidden state. The output hidden state of this RNN $\hat{\boldsymbol{\zeta}}_i^t$ is the updated instant incentive estimation over all opponents of agent $i$.

The decoder $\psi_i$ predicts all opponents' trajectories over a pre-defined length $t_p$ from instant incentive estimations $\hat{\boldsymbol{\zeta}}_i^t$. We use another RNN that takes agent $i$'s current observation $\mathbf{o}_i^t$ as the input and its current instant incentive estimation $\hat{\boldsymbol{\zeta}}_i^t$ as the hidden state. The first output of this RNN is the prediction of opponents' states $\hat{\mathbf{o}}_i^{t+1}$ at the next time step $t+1$. Then we use $\hat{\mathbf{o}}_i^{t+1}$ as the new input of RNN and iteratively predict opponents' states. The sequence of opponents' state predictions $\{\hat{\mathbf{o}}_i^{t+k}\}_{k=1}^{t_p} \sim \psi_i(\mathbf{o}_i^t, \hat{\boldsymbol{\zeta}}_i^t)$ is the trajectory prediction from $t+1$ to $t+t_p$ for all opponents of agent $i$. We train our encoder and decoder with instant incentive inference loss $\mathcal{J}_{\zeta_i}$, given by an average L1-norm error between predicted trajectories $\{\hat{\mathbf{o}}_i^{t+k}\}_{k=1}^{t_p}$ and ground truth trajectories $\{\mathbf{o}_i^{t+k}\}_{k=1}^{t_p}$.

$$\mathcal{J}_{\zeta_i} = \min_{\phi_i, \psi_i} \frac{1}{N t_p} \sum_{j=1}^{N} \sum_{k=0}^{t_p-1} \left\| \psi_i(\mathbf{o}_i^t, \phi_i(\mathbf{o}_i^t, \hat{\boldsymbol{\beta}}_i^t, \hat{\boldsymbol{\zeta}}_i^{t-1})) - \mathbf{o}_i^{t+k+1} \right\|_1 \tag{5}$$

## 3.3 Implementation

The pseudocode (Algorithm 1) and flow graph (Figure 3) of our algorithm are given in Appendix A. For each environmental step $t$ in the execution (line 4), agent $i$ gathers its current and historical observations $\mathbf{o}_i^t$ and $\boldsymbol{h}_i^t$ (line 6), and uses this information to infer their opponents' behavioral incentives $\boldsymbol{\beta}_i^t$ and instant incentives $\boldsymbol{\zeta}_i^t$ (lines 7 and 8). After that, agent $i$'s policy $\pi_i$ selects action $a_i^t$ (line 9). The backbone algorithm for each agent's controller is PPO [36], which includes a policy network $\pi_i$ and a critic network $Q_i$. For each gradient step in training, agent $i$ updates its policy $\pi_i$ and critic $Q_i$ (line 15) with sampled trajectories, computes the behavioral incentive inference loss $\mathcal{J}_{\beta_i}$ (line 16) to update its behavioral incentive inference encoder $\theta_{\mathcal{E}_i}$ and decoder $\theta_{\mathcal{D}_i}$ with $\mathcal{J}_{\beta_i}$, and uses instant incentive inference loss $\mathcal{J}_{\zeta_i}$ (line 17) to update its instant incentive inference encoder $\theta_{\phi_i}$ and decoder $\theta_{\psi_i}$.

## 4 Empirical Results and Discussion

We perform experiments over two non-cooperative environments, Non-Cooperative Navigation [28] and Heterogeneous Highway [22]. Experiments are designed from two perspectives. The first is to compare our approach's performance with other CTDE and DTDE MARL approaches in non-cooperative environments. In this paper, we compare our method with two CTDE MARL baselines, QMIX [37] and MAPPO [38], and one DTDE MARL baseline, IPPO [39]. QMIX uses a central network to assign credits among agents with respect to their Q-values and global states. MAPPO

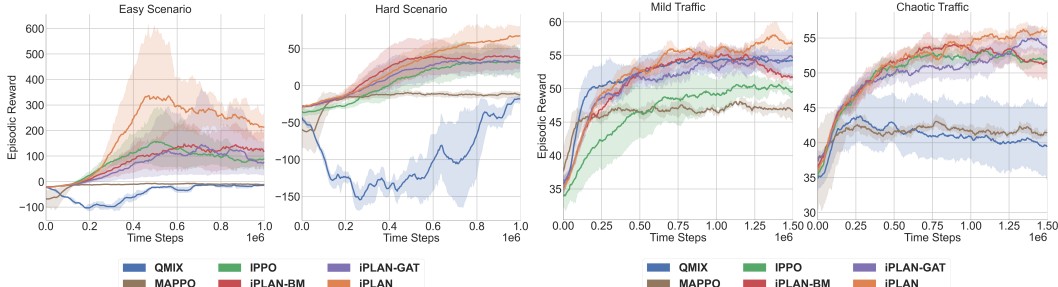

(a) **Non-Cooperative Navigation:** with 3 agents in the (left) *easy* and (right) *hard* scenarios. 50 steps/episode.

(b) **Heterogeneous Highway:** with 5 agents in (left) *mild* and (right) *chaotic* scenarios. 90 steps/episode.

Figure 2: Comparison of average episodic reward in the Non-Cooperative Navigation and Heterogeneous Highway environments. **Conclusion:** iPLAN (orange) outperforms CTDE approaches like QMIX (blue) and MAPPO (brown) as well as IPPO (green) in heterogeneous traffic environments.

uses a central critic that reads the observation of all agents and generates a critic value to update distributed actors. IPPO uses a distinct PPO policy to control each agent without any centralized training, weight-sharing, communication, or inference module. The other perspective is to show the necessity of instant and behavioral incentive inference, especially under highly heterogeneous scenarios. We further design two scenarios with different heterogeneity levels in both environments and perform ablation studies over two variants of our method, including iPLAN-BM a vanilla IPPO controller without the instant incentive inference module, and iPLAN-GAT, a vanilla IPPO controller without behavioral incentive inference module. Details regarding the experiment environment design are given in Appendix C. Further details regarding implementation, visual results, module design, and hyper-parameter study are given in Appendix D, E, F, and G.3, respectively.

## 4.1 Results on Non-Cooperative Navigation

Figure 2a compares episodic rewards in *easy* and *hard* scenarios. iPLAN outperforms other methods with low deviation. iPLAN-GAT and vanilla IPPO have larger deviations, indicating the benefit of behavioral incentive inference in stabilizing strategies. QMIX and MAPPO perform poorly with negative episodic rewards in both scenarios. In Non-Cooperative Navigation, agents are attracted to the closest landmark at each time step, allowing multiple agents to target the same landmark simultaneously. As there is no consensus in destination assignment, agents must observe and infer others' strategies to modify their own. This reliance on observations and inference contributes to the superior performance of DTDE MARL approaches over CTDE MARL approaches in Non-Cooperative Navigation.

## 4.2 Results on Heterogeneous Highway

Figure 2b compares episodic rewards in the *mild* and *chaotic* traffic scenarios of the Heterogeneous Highway. We find that iPLAN has the best episodic reward in both the *mild* and *chaotic* traffic. iPLAN-GAT, iPLAN-BM, and vanilla IPPO have similar performances in *mild* traffic scenarios, but iPLAN-GAT is slightly worse than iPLAN in the *chaotic* traffic. Notably, two CTDE MARL baselines have much lower episodic rewards than DTDE MARL approaches in *chaotic* traffic, and QMIX has a significant collapse compared with its performance in *mild* traffic.

In addition to the episodic reward curve comparison, we evaluate our method and baselines over several navigation metrics, including:

**Episodic Average Speed**. Agents' average speed during their lifetime in an episode. Agents are encouraged to drive faster when driving between 20 and 30 $m/s$.

**Average Survival Time**. The average time steps passed over all agents before they collide or reach the end of this episode. Longer survival time reflects agents' better ability to avoid collisions.

**Success Rate**. The percentage of vehicles that still stay collision-free when an episode ends.

Table 1 shows navigation metrics for *mild* and *chaotic* traffic. High speed (closer to 30) correlates with low survival time and success rate. This is because aggressive reward-exploiting policies increase collision risk, reducing long-term reward. Approaches like iPLAN and iPLAN-GAT drive slower (closer to 20) for safety and higher episodic reward. Instant incentive inference improves episodic reward and success rates, especially in *chaotic* traffic. iPLAN maintains similar success rates but a higher average speed in *mild* traffic, being more conservative and dependent in heterogeneous traffic. Comparing iPLAN and iPLAN-GAT, iPLAN drives faster in both scenarios for higher episodic reward. iPLAN-GAT has a longer survival time in *mild* traffic, but the opposite in *chaotic* traffic. This indicates that agents are more dependent on their instant incentive inference in *mild* traffic when opponents' trajectories are more predictable, and more dependent on their behavioral incentive inference in *chaotic* traffic due to aggressive vehicles' unpredictable behaviors. QMIX performs well in *mild* traffic but poorly in *chaotic* traffic (success rate $< 20\%$) due to environmental heterogeneity effect on its credit assignment.

## 5    Conclusion, Limitations, and Future Work

This paper presents a novel intent-aware distributed MARL algorithm tailored for planning and navigation in heterogeneous traffic. We model two distinct incentives, the behavioral incentive and the instant incentive, for agents' strategies. Our approach enables agents to infer their opponents' behavior incentives and integrate this inferred information into their decision-making and motion-planning processes. Results show that our approach shows a promising result in the two environments we use, Non-Cooperative Navigation and Heterogeneous Highway, with a better performance in episodic reward curves and navigation metrics

|  | Approach | Avg. Speed ($m/s$) | Avg. Survival Time (# Time Steps) ↑ | Success Rate (%) ↑ |
|---|---|---|---|---|
| Mild | QMIX [37] | $21.24 \pm 0.09$ | $\mathbf{75.98 \pm 3.67}$ | $67.50 \pm 6.34$ |
|  | MAPPO [38] | $\mathbf{27.85 \pm 0.40}$ | $48.94 \pm 3.11$ | $32.81 \pm 5.22$ |
|  | IPPO [39] | $22.63 \pm 0.17$ | $66.13 \pm 4.13$ | $49.06 \pm 7.35$ |
|  | iPLAN-GAT | $22.05 \pm 0.11$ | $75.54 \pm 3.61$ | $\mathbf{68.44 \pm 6.64}$ |
|  | iPLAN-BM | $22.61 \pm 0.16$ | $64.11 \pm 4.28$ | $45.63 \pm 6.33$ |
|  | iPLAN | $22.91 \pm 0.15$ | $70.56 \pm 3.81$ | $\mathbf{68.44 \pm 5.86}$ |
| Chaotic | QMIX [37] | $27.06 \pm 0.47$ | $39.38 \pm 2.64$ | $19.69 \pm 3.72$ |
|  | MAPPO [38] | $\mathbf{29.46 \pm 0.05}$ | $42.31 \pm 2.43$ | $16.25 \pm 3.76$ |
|  | IPPO [39] | $22.28 \pm 0.13$ | $67.01 \pm 3.64$ | $42.50 \pm 7.12$ |
|  | iPLAN-GAT | $20.91 \pm 0.13$ | $71.24 \pm 3.83$ | $61.88 \pm 6.41$ |
|  | iPLAN-BM | $21.65 \pm 0.28$ | $63.20 \pm 3.51$ | $35.31 \pm 5.66$ |
|  | iPLAN | $21.61 \pm 0.16$ | $\mathbf{76.20 \pm 3.33}$ | $67.81 \pm 5.91$ |

Table 1: **Navigation metrics in Heterogeneous Highway:** Metrics are averaged over $64$ episodes with $0.95$ confidence. iPLAN outperforms all other approaches in its highest success rate and survival time, though it tends to be conservative in its average speed.

than baselines. Nonetheless, it is important to acknowledge certain limitations within our research:

Firstly, our evaluation has been confined to a simulation environment. Real-world autonomous driving scenarios present a diverse array of traffic situations and driving behaviors, potentially posing challenges for the generalization of our approach to unfamiliar strategies unseen during training. Furthermore, attempting to predict the complete state of a multi-agent system in real-world contexts may introduce the risk of significant and potentially hazardous errors, as agents might inaccurately reconstruct or predict states.

Secondly, the design of our framework centers on the use of two incentives for representing and inferring the objectives of other drivers in motion planning. However, in certain scenarios, such as in more straightforward or mixed conditions, the necessity of dual incentives remains ambiguous *i.e.* it might be that a singular incentive set is adequate. Future research could delve deeper into the design of inference models. Additionally, while our contributions are empirically substantiated, they currently lack a solid theoretical foundation. Therefore, forthcoming research initiatives should prioritize the establishment of theoretical safety and convergence bounds for our approach.

**Acknowledgments**

We would like to thank Caroline Wang for helpful discussions and feedback during the course of this paper. This research was supported by Army Cooperative Agreement W911NF2120076.

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

# A Algorithm

## A.1 Pseudocode

---

**Algorithm 1** iPLAN: Intent-aware Planning in Heterogeneous Traffic via Distributed MARL

---

**Require:** Number of agents $N$, Number of experiences $K$ for experience replay, Length of historical observation sequence $t_h$, Length of trajectory prediction $t_p$
 1: **Initialize:** Agent $i$'s network parameters $\theta_{\pi_i}, \theta_{Q_i}, \theta_{\mathcal{E}_i}, \theta_{\mathcal{D}_i}, \theta_{\phi_i}, \theta_{\psi_i}, i = 1, 2, \ldots, N$
 2: **Initialize:** Replay Buffer $\mathcal{B} \leftarrow \emptyset$, Incentive Inferences $\boldsymbol{\beta}_i^0 \leftarrow \overrightarrow{0}, \boldsymbol{\zeta}_i^0 \leftarrow \overrightarrow{0}$ for $i = 1, 2, \ldots, N$
 3: **for** each environmental step $t$ **do**
 4:     **for** $i = 1, 2, \ldots, N$ **do**
 5:         Gather current and historical observations $\mathbf{o}_i^t$ and $\boldsymbol{h}_i^t$
 6:         Infer behavioral incentives $\hat{\boldsymbol{\beta}}_i^t$ with $\mathcal{E}_i(\boldsymbol{h}_i^t, \hat{\boldsymbol{\beta}}_i^{t-1})$
 7:         Infer instant incentives $\hat{\boldsymbol{\zeta}}_i^t$ with $\phi_i(\mathbf{o}_i^t, \hat{\boldsymbol{\beta}}_i^t, \hat{\boldsymbol{\zeta}}_i^{t-1})$
 8:         Select action $a_i^t$ with $\pi_i(\cdot | \mathbf{o}_i^t, \hat{\boldsymbol{\beta}}_i^t, \hat{\boldsymbol{\zeta}}_i^t)$
 9:     **end for**
10: **end for**
11: **for** each gradient step **do**
12:     Sample $K$ experiences from the replay buffer $\mathcal{B}$
13:     **for** $k = 1, 2, \ldots, K$ **do**
14:         **for** $i = 1, 2, \ldots, N$ **do**
15:             **// Update PPO controller**
16:             Perform experience replay on experience $k$
17:             Update policy $\theta_{\pi_i}$ and critic $\theta_{Q_i}$ of the PPO controller
18:             **// Update behavioral incentive inference module**
19:             **for** each step $t^k$ in experience $k$ **do**
20:                 Gather historical observation sequence $\boldsymbol{h}_i^{t^k}$ from experience $k$
21:                 Infer behavioral incentives $\hat{\boldsymbol{\beta}}_i^{t^k}$ with $\mathcal{E}_i(\boldsymbol{h}_i^{t^k}, \hat{\boldsymbol{\beta}}_i^{t^k-1})$
22:                 Predict future observation sequence $\hat{\boldsymbol{h}}_i^{t^k+t_h}$ with $\mathcal{D}_i(\boldsymbol{h}_i^{t^k}, \hat{\boldsymbol{\beta}}_i^{t^k})$
23:                 Use predicted $\hat{\boldsymbol{h}}_i^{t^k+t_h}$ and ground-truth $\boldsymbol{h}_i^{t^k+t_h}$ to compute $\mathcal{J}_{\beta_i}$ in (4)
24:                 Update behavioral incentive encoder $\theta_{\mathcal{E}_i}$ and decoder $\theta_{\mathcal{D}_i}$ with $\mathcal{J}_{\beta_i}$
25:             **end for**
26:             **// Update instant incentive inference module**
27:             **for** each step $t^k$ in experience $k$ **do**
28:                 Gather current observation $\mathbf{o}_i^{t^k}$ and behavioral incentives $\hat{\boldsymbol{\beta}}_i^{t^k}$ from experience $k$
29:                 Infer instant incentives $\hat{\boldsymbol{\zeta}}_i^{t^k}$ with $\phi_i(\mathbf{o}_i^{t^k}, \hat{\boldsymbol{\beta}}_i^{t^k}, \hat{\boldsymbol{\zeta}}_i^{t^k-1})$
30:                 Predict future trajectories $\{\hat{\mathbf{o}}_i^{t^k+j}\}_{j=1}^{t_p}$ with $\psi_i(\mathbf{o}_i^{t^k}, \hat{\boldsymbol{\zeta}}_i^{t^k})$
31:                 Use predicted $\{\hat{\mathbf{o}}_i^{t^k+j}\}_{j=1}^{t_p}$ and ground-truth $\{\mathbf{o}_i^{t^k+j}\}_{j=1}^{t_p}$ to compute $\mathcal{J}_{\zeta_i}$ in (5)
32:                 Update instant incentive encoder $\theta_{\phi_i}$ and decoder $\theta_{\psi_i}$ with $\mathcal{J}_{\zeta_i}$
33:             **end for**
34:         **end for**
35:     **end for**
36: **end for**
37: **Output:** $\mathcal{E}_i^*, \phi_i^*, \pi_i^*$ for each $i$.

---

## A.2 Flow Diagram

To illustrate our algorithm in Algorithm 1, we create a flow diagram to visualize the execution and training procedure performed by agent $i$ in iPLAN. Details could be found in Fig. 3.

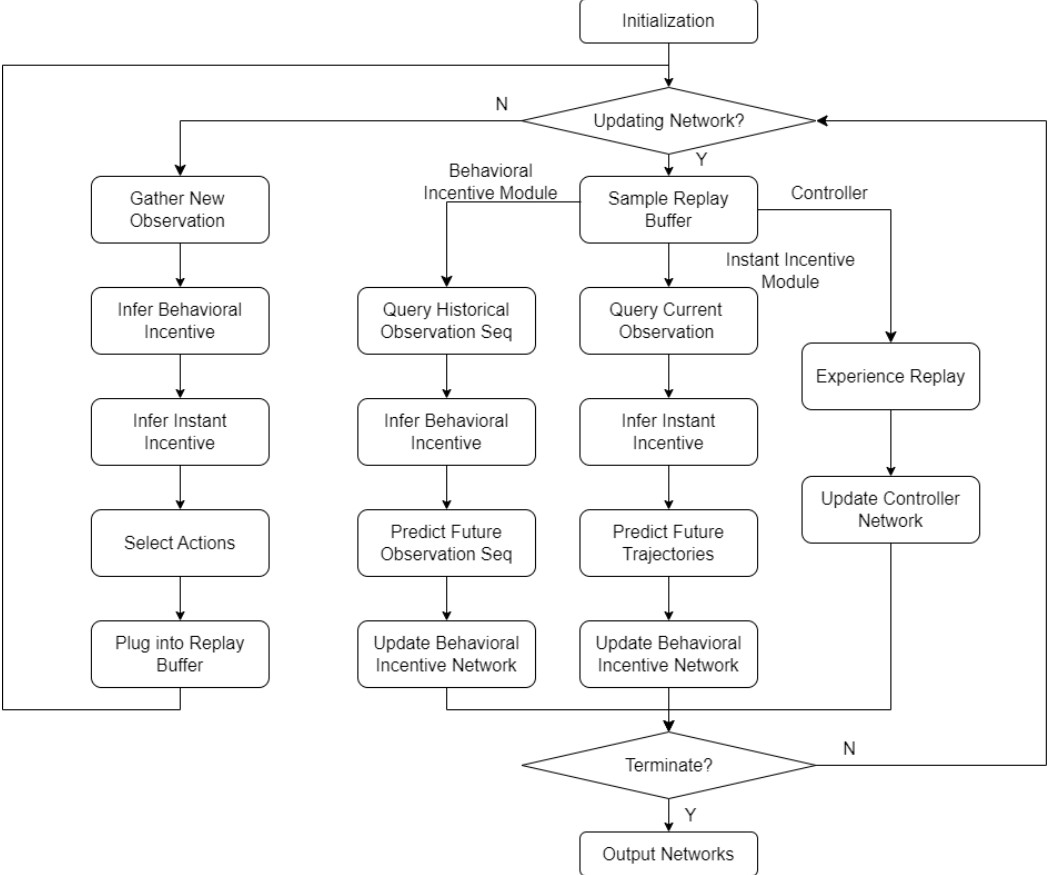

Figure 3: Flow Diagram for Algorithm 1

## B   Related Work

**Trajectory and Intent Prediction for Autonomous Driving.** Trajectory prediction is a fundamental task in autonomous driving [40, 41, 42]. TraPHic and RobustTP [43, 8] use an LSTM-CNN framework to predict trajectories in dense and complex traffic. TNT [44] uses target prediction, motion estimation, and ranking-based trajectory selection to predict future trajectories. DESIRE [4] uses sample generation and trajectory ranking for trajectory prediction. PRECOG [13] combines conditioned trajectory forecasting with planning objectives for AVs. Additionally, many methods focus on intent prediction to gain a better understanding of interactions between vehicles when predicting trajectories. Intent prediction can be done by physical-based methods like Kalman filter [45] or Monte Carlo [46], classical machine learning like Gaussian processes (GP) [47, 48], Hidden Markov Model (HMM) [49], and Monte Carlo Tree Search (MCTS) [50], or deep learning-based methods such as Trajectron++ and CS-LSTM [7, 51]. [52] uses a Seq2Seq framework to encode agents' observations over neighboring vehicles as their social context for trajectory forecasting and decision-making. [53] uses temporal smoothness in attention modeling for interactions and a sequential model for trajectory prediction. However, most methods overlook variations in driving behaviors, which deteriorates their reliability in heterogeneous traffics.

**Intent-aware Multi-agent Reinforcement Learning.** As a large-scale and non-cooperative [54] scenario, the awareness of opponents' incentives is quite important when implementing MARL in autonomous driving. Intent-aware multi-agent reinforcement learning [5] estimates an intrinsic value that represents opponents' intentions for communication [55] or decision-making. Many intent inference modules are based on Theory of Mind (ToM) [56] reasoning or social attention-related mechanisms [57, 58]. [59] uses ToM reasoning over opponents' reward functions from their historical behaviors in performing multi-agent inverse reinforcement learning (MAIRL). [60] uses game theory ideas to reason about other agents' incentives and help decentralized planning among strategic agents. However, many prior works oversimplify the intent inference and make some prior assumptions about the content of intent. In the real world, agents' incentives are more complex and intractable during interactions among large groups of agents, so a more general and high-level incentive representation is needed in intent-aware MARL.

**Opponent Modeling.** Opponent modeling [61] in multi-agent reinforcement learning usually deploys various inference mechanisms to understand and predict other agents' policies. Opponent modeling could be done by either estimating others' actions and safety via Gaussian Process [62] or by generating embeddings representing opponents' observations and actions [63]. Inferring opponents' policies helps to interpret peer agents' actions [64] and makes agents more adaptive when encountering new partners [65]. Notably, many works [66, 67] reveal the phenomenon whereby ego agents' policies also influence opponents' policies. To track the dynamic variation of opponents' strategies made by an ego agent's influence, [68, 69] propose the latent representation to model opponents' strategies and influence based on their findings on the underlying structure in agents' strategy space. [70] provides a causal influence mechanism over opponents' actions and defines an influential reward over actions with high influence over others' policies. [71] proposes an optimization objective that accounts for the long-term impact of ego agents' behavior in opponent modeling. A considerable limitation of many current methodologies is the underlying assumption that agents continually interact with a consistent set of opponents across episodes. This assumption is a misfit for real-world autonomous driving contexts. On roads, drivers constantly come across different vehicles and drivers, necessitating the ability to infer the intentions of new opponents with minimal prior knowledge.

## C   Experiment Details

### C.1   Non-Cooperative Navigation

Non-Cooperative Navigation is developed based on the Multi-agent Particles Environment (MPE) [28]. $n$ agents are required to maximize their coverage over $n$ landmarks without any ex-

plicit cooperation or inter-agent communication mechanism. Instead of being assigned some pre-determined landmarks as their destinations, agents are attracted to the immediate closest landmark at each time step. This indicates that an agent's destination is not fixed in an episode and that multiple agents can be attracted to a specific landmark simultaneously. Agents should properly select their intended landmarks, reach and stay at their intended landmarks, and avoid any conflicts with other agents. The length of each episode is 50 steps. Agents and landmarks are randomly initialized within a $2 \times 2$ world space. All plots in Non-Cooperative Navigation are averaged over 5 random seeds.

In Non-Cooperative Navigation, there are three different kinds of agents that are controllable by MARL policies and one kind of agent that is controlled by the pre-defined random policy taking random actions at each time step. Table 2 shows the parameters of different kinds of agents; their major differences come from their sizes and acceleration values:

| Agent Type | Size | Acceleration |
|---|---|---|
| Normal | 0.08 | 1.0 |
| Tiny | 0.06 | 1.1 |
| Bulky | 0.10 | 0.9 |
| Random | 0.08 | 1.0 |

Table 2: Parameters for Agents in Non-cooperative Navigation

**Scenarios.** Two scenarios with different heterogeneity levels are included in this paper:

- **Easy:** 1 Normal agent, 1 Tiny agent, and 1 Bulky agent.

- **Hard:** 1 Normal agent, 1 Tiny agent, 1 Bulky agent, and 1 Random agent.

Note that all agents in the *easy* scenario are controllable. One uncontrollable agent exists along with three controllable agents in the *hard* scenario, which makes this scenario more heterogeneous.

**Observation Space.** Non-Cooperative Navigation is a fully-observable environment with a continuous observation space for each agent. The observation vector of an agent is composed of state vectors of all entities within the world space, including the states of all agents and landmarks. Here, we denote the state of an entity in Non-Cooperative Navigation as a vector with its ID, current position, and velocity. Within agent $i$'s observation vector, the positions of all entities are their positions with respect to agent $i$. Agent $i$'s ego state vector locates it at the top of its observation vector and uses its own absolute position in the world space. For those CTDE MARL algorithms requiring the global state, the global state is the collection of all entities' state vectors composed of their IDs, absolute positions, and velocities in the world space.

**Action Space.** Non-Cooperative Navigation has a discrete action space with 5 identical high-level actions, $\{idle, up, down, left, right\}$. Taking action in any direction (*i.e.*, all actions except *idle*) makes this agent accelerate by one step size in that direction. The acceleration step size varies in different kinds of agents.

**Reward.** Each agent has an individual reward function in Non-Cooperative Navigation. An agent gets a penalty that equals its distance from the closest landmark in the environment at each time step. Notably, multiple agents may get this penalty with respect to their distances to a specific landmark if this landmark is the closest to all of them. If a collision happens between two agents, both will receive a penalty of $-5$. If an agent reaches the scope with a distance of less than $0.1$ to any landmarks, this agent receives a positive reward of $10$. We denote this scope as the *rewarding scope*. If all controllable agents reach and stay within the *rewarding scope* without conflicts, they all receive a positive reward of $100$.

## C.2  Heterogeneous Highway

Heterogeneous Highway is developed based on Highway-env [22], which is a 2D autonomous driving simulator based on PyGame. Traffic scenarios in our environment are designed based on the Highway scenario given by Highway-env with simulated vehicles driving on a multi-lane highway. The objective of vehicles controlled by MARL algorithms is to maintain a collision-free trajectory with a proper speed between 20 and 30 $m/s$ when driving through heterogeneous traffic. Uncontrollable vehicles are controlled by three different behavior-driven vehicle models modified from models proposed in [72], and we denote them as *Normal*, *Aggressive*, and *Conservative* vehicles. Their major differences come from their kinematic features, given in Table 3.

| Kinematic Parameters | Normal | Aggressive | Conservative |
|---|---|---|---|
| Max Speed ($m/s$) | 40 | 50 | 40 |
| Default Speed Range ($m/s$) | [23, 25] | [35, 40] | [23, 25] |
| Max Acceleration ($m/s^2$) | 6.0 | 9.0 | 5.0 |
| Desired Acceleration ($m/s^2$) | 3.0 | 6.0 | 2.0 |
| Desired Deceleration ($m/s^2$) | −5.0 | −9.0 | −4.0 |
| Desired Front Distance ($m$) | $5.0 + l$ | 0.5 | $8.0 + l$ |
| Time Wanted (Before Stop) ($s$) | 1.5 | 1.2 | 1.8 |

Table 3: Kinematics for the behavior-driven vehicle model used in Heterogeneous Highway scenarios. All vehicles are assumed to have the same size $l$.

The length of each episode is 90 steps. Initially, vehicles are randomly placed throughout the world space with a density of 1. All results in Heterogeneous Highway have averaged over 5 random seeds.

**Scenarios.** Two scenarios under *mild* and *chaotic* traffic are included in this paper. Each scenario has 5 controllable vehicles and 50 behavior-driven vehicles uniformly distributed over an 8-lane highway. The compositions of different behavior-driven vehicles relate to the heterogeneity of traffic. The *mild* traffic has mostly normal-behaving vehicles, so we consider this scenario more homogeneous. In the *chaotic* traffic scenario, more aggressive vehicles exist, which makes the environment more heterogeneous. Here are the propositions of each kind of behavior-driven vehicle in *mild* and *chaotic* traffic scenarios:

- **Mild:** 80% Normal vehicles + 10% Aggressive vehicles + 10% Conservative vehicles.
- **Chaotic:** 40% Normal vehicles + 30% Aggressive vehicles + 30% Conservative vehicles.

**Observation Space.** Heterogeneous Highway is a partially-observable environment in that agents can only observe 15 other vehicles within their predefined observation scope. The observation scope for each agent is 100 $m$ in both directions of the x-axis and 20 $m$ in both directions of the y-axis. Each agent has a continuous observation space. The observation vector of an agent is composed of stacked state vectors of all vehicles within its observable scope. Here, we denote a state vector of a vehicle as a vector with its ID, current position, and velocity in the world space. For agent $i$'s observation vector, its ego state vector locates it at the top of its observation vector and uses its own absolute position in the world space. The remaining state vectors are state vectors of vehicles observed by agent $i$ using their positions relative to agent $i$. The global state for CTDE MARL baselines is made up of concatenated state vectors of all controllable and uncontrollable vehicles within the environment.

**Action Space.** The action space for each controllable agent is discrete with 5 distinct actions, {*lane left*, *idle*, *lane right*, *faster*, *slower*}. Vehicles convert their high-level discrete action orders into a sequence of $x, y$ coordinates when taking actions. All vehicles' low-level motion models follow the Kinematic Bicycle Model [73], and their kinematic parameters are given in Table 3.

**Reward.** For DTDE MARL algorithms, each agent receives an individual reward, while for CTDE MARL approaches, all agents receive a global reward by summing their individual rewards together. Once an agent collides with other vehicles, this agent gets a −1 penalty. Agents are encouraged to

keep right, and an agent gets a linear reward from 0 to 0.1 with respect to its distance to the rightmost lane. Agents are encouraged to keep a speed within the rewarding speed range of 20 to 30 $m/s$. At each time step, an agent is rewarded with respect to its speed within the reward speed range. If an agent can reach a speed of 30 or higher at this time step, it gets a reward of 0.4. If an agent keeps a speed of 20 or lower at this time step, it gets a reward of 0.

# D  Visual Results

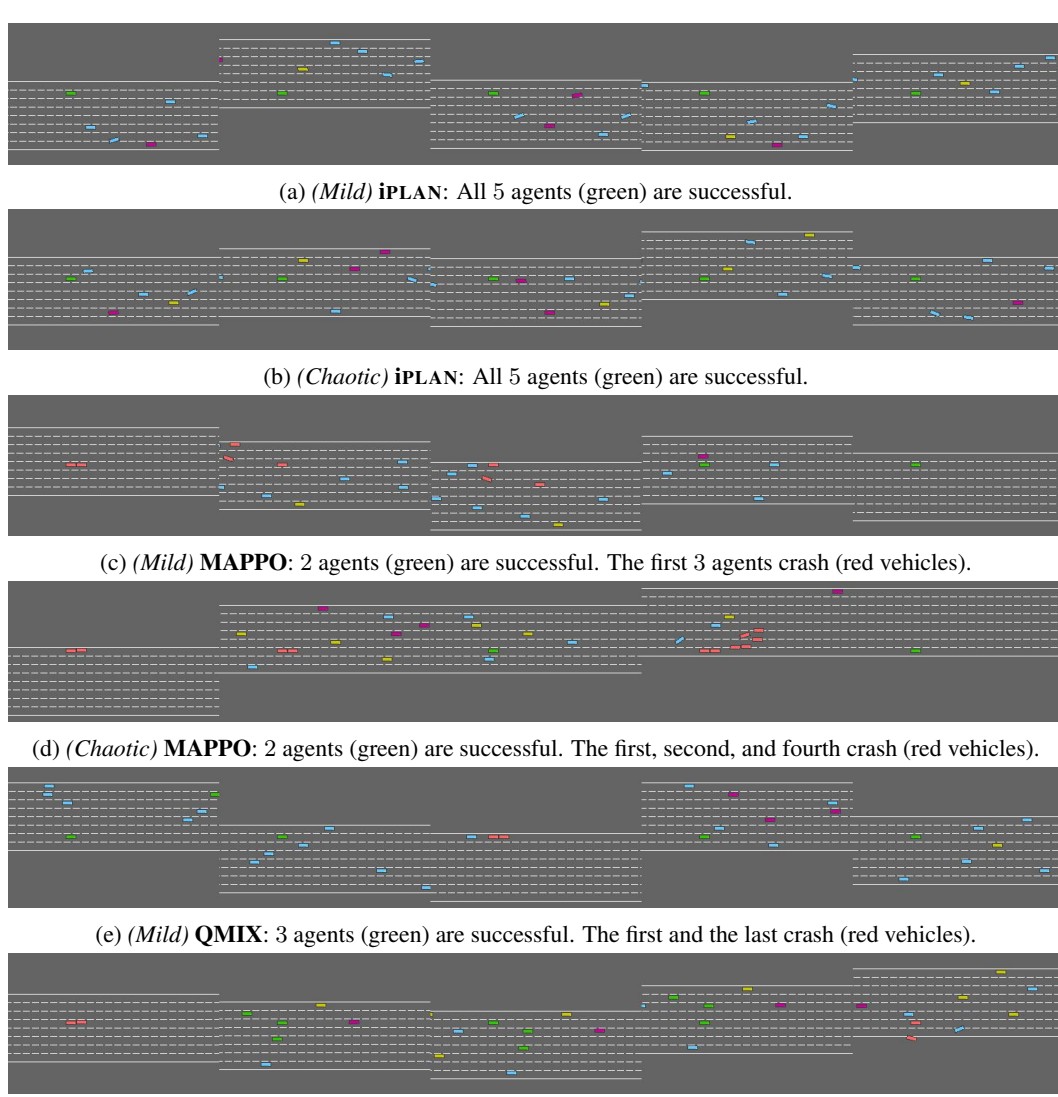

(a) *(Mild)* **iPLAN**: All 5 agents (green) are successful.

(b) *(Chaotic)* **iPLAN**: All 5 agents (green) are successful.

(c) *(Mild)* **MAPPO**: 2 agents (green) are successful. The first 3 agents crash (red vehicles).

(d) *(Chaotic)* **MAPPO**: 2 agents (green) are successful. The first, second, and fourth crash (red vehicles).

(e) *(Mild)* **QMIX**: 3 agents (green) are successful. The first and the last crash (red vehicles).

(f) *(Chaotic)* **QMIX**: 4 agents (green) are successful. The third vehicle crashes (red vehicle).

Figure 4: **Qualitative results on Heterogeneous Highway:** We visually compare the performance of iPLAN with QMIX and MAPPO. Each baseline is tested with multiple learning agents shown in green, and each figure above shows 5 such learning agents from their respective viewpoints. In each figure, we show cases when the green agents succeeded versus when they crashed. **Conclusion:** All 5 agents succeed using iPLAN as shown in Figures 4a and 4b whereas on average 2 or more agents crash using QMIX or MAPPO.

# E   Implementation Details

**Behavioral Incentive Inference.** The encoder of the behavioral incentive inference module uses a 1-layer GRU network with a size of 32 and generates an 8-length vector as the latent representation of the behavioral incentive. The decoder uses another 1-layer GRU network with a size of $64$ to predict future state sequences, with a dropout rate of $0.1$. The truncated length $t_h$ of the observation history is $10$ in the Heterogeneous Highway and $5$ in Non-Cooperative Navigation. The learning rate for behavioral incentive inference is $1 \times 10^{-4}$.

**Instant Incentive Inference.** The encoder of the instant incentive inference module uses a GAT with a hidden-layer size of 32 and a 1-layer GRU with a hidden-layer size of 32. The decoder uses another 32-size GRU to predict the trajectory, with a dropout of $0.1$. The trajectory prediction length $t_p$ is $5$ in the Heterogeneous Highway and $2$ in Non-Cooperative Navigation. The learning rate for instant incentive inference is $2 \times 10^{-5}$.

**IPPO Controller.** The input of the PPO controller for an agent is the flattened vector of its observation of all entities' (vehicles in Heterogeneous Highway; other agents and landmarks in Non-Cooperative Navigation) states and the inference of all other agents' (or other vehicles') behavioral incentive and instant incentive. The PPO controller has a buffer size of $256$ and a learning rate of $5 \times 10^{-4}$ for its actor and critic. All fully-connected and recurrent layers in the actor and critic of PPO have a dimension of $64$.

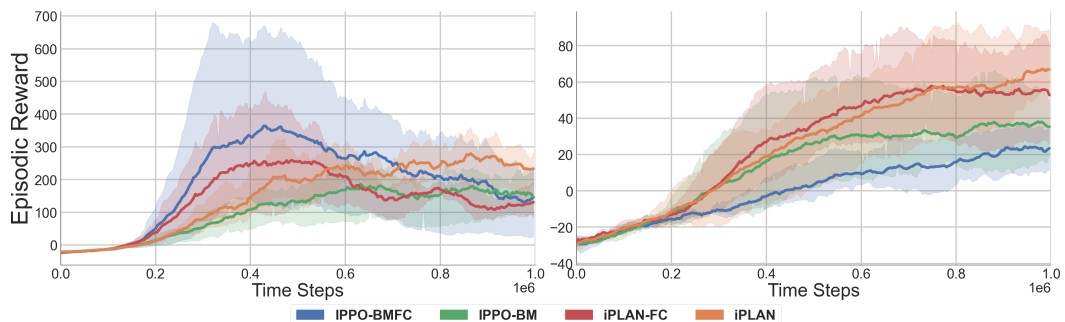

Figure 5: **Non-Cooperative Navigation with recurrent and fully-connected behavioral incentive inference modules:**  Comparing the episodic reward in the (left) *easy* and (right) *hard* scenarios. **Conclusion:** iPLAN (orange) performs better than others in the *easy* scenario. iPLAN-BM (green) outperforms iPLAN-BMFC (blue) in the *hard* scenario.

# F    Supplementary Experiments: Behavioral Incentive Inference Module

## F.1    Choice of Behavioral Incentive Inference Module

During our design process for the behavioral incentive inference module, we experimented with different architectures in the encoder-decoder framework.  Specifically, we tested the usage of a recurrent layer and a fully-connected layer. While the latter design has been utilized in prior works for similar tasks [68, 69, 65], we want to address the temporal relationship presented in the historical observation sequences. To evaluate the performance of these two designs, we conduct experiments on the comparison between iPLAN and an alternative approach that uses a fully-connected behavioral incentive inference module.

In this module, we take the flattened historical observation sequence as input and employed a 3-layer fully-connected network with a hidden layer dimension of $64$ as the encoder. This encoder generates an $8$-length latent representation of the behavioral incentive. Additionally, we use another 3-layer fully-connected network with the same hidden layer dimension as the decoder to predict future state sequences for opponents. The learning rate for this alternative behavioral incentive inference module is set to $1 \times 10^{-4}$.

We depict the episodic rewards over both environments in Figure 5 and Figure 6. In these figures, the approach employing the fully-connected network in the behavioral incentive inference module is denoted as iPLAN-FC, and the same notation applies to iPLAN-BMFC. The results indicate that incorporating the recurrent layer improves the performance of the behavioral incentive inference module.  Specifically, our approach (iPLAN, orange curve) demonstrates better performance than iPLAN-FC (red curve). Similarly, iPLAN-BM (green curve) outperforms iPLAN-BMFC (blue curve) in general.

## F.2    Soft Updating Policy

Another important aspect to consider in our behavioral incentive inference module design is the updating policy for behavioral incentives. Drawing inspiration from previous works [68, 69, 65], we divide the behavioral incentive inference within an episode into multiple sub-episodes. We aim to update the behavioral incentive inferences at the end of each sub-episode. This updating policy is referred to as the *hard-updating policy*, in contrast to the *soft-updating policy*, which treats the behavioral incentive inference as a converging procedure and iteratively updates the behavioral incentive inferences.

In our experiments, we evaluate the performance of iPLAN and an alternative method, iPLAN-Hard, which employs a hard-updating policy. In iPLAN-Hard, the behavioral incentive inference module updates the behavior incentives at specific time intervals (e.g., $t = 10, 20, 30, \ldots$), while the behav-

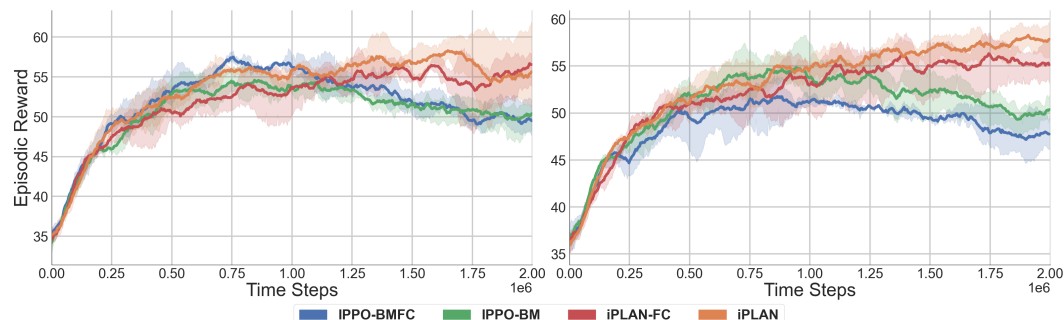

Figure 6: **Heterogeneous Highway with recurrent and fully-connected behavioral incentive inference modules:** Comparing the episodic reward in the (left) *mild* and (right) *chaotic* traffic scenarios. **Conclusion:** Approaches using recurrent behavioral incentive inference modules, including iPLAN (orange) and iPLAN-BM (green), outperform those using fully-connected behavioral incentive inference modules.

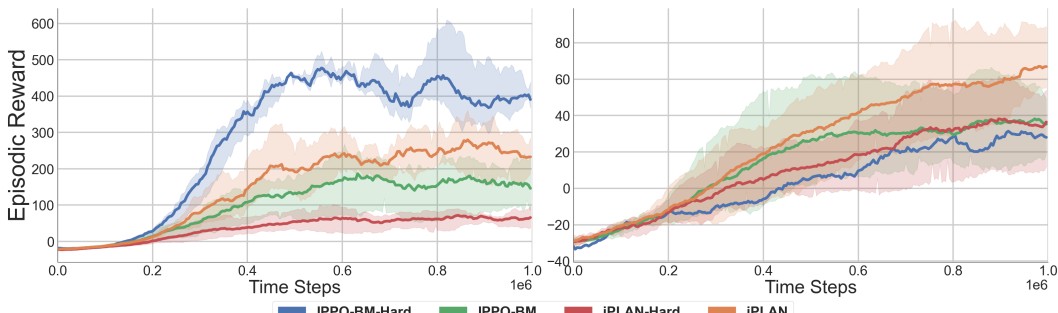

Figure 7: **Non-Cooperative Navigation with and without soft-updating policy:** Comparing the episodic reward in the (left) *easy* and (right) *hard* scenarios. **Conclusion:** iPLAN-BM-Hard (blue) performs the best in the *easy* scenario and the worst in the *hard* scenario. iPLAN (orange) has a better performance in general.

ior incentive inferences remain unchanged between these updating points (*i.e.*, between $t = 10$ and $t = 20$). All other hyperparameters used in the behavioral incentive inference module remain the same.

Figure 7 and Figure 8 illustrate the results obtained with different behavior incentive updating policies. In Non-Cooperative Navigation, iPLAN-BM-Hard achieves the best performance in the *easy* scenario but performs the worst in the *hard* scenario. This significant gap between scenarios may stem from its inability to capture heterogeneity, considering that all agents in the *easy* scenario are controllable. On the other hand, iPLAN exhibits overall better performance, ranking second in the *easy* scenario and first in the *hard* scenario. This outcome demonstrates that the soft-updating policy helps address heterogeneity and stabilize agents' strategies. In Heterogeneous Highway, iPLAN-Hard denotes the approach that uses a hard-updating policy for behavioral incentives, and the same notation applies to iPLAN-BM-Hard. The results reveal that despite the difference in updating policies, their performances remain relatively close in *mild* traffic for both comparison pairs (iPLAN *v.s.* iPLAN-Hard, iPLAN-BM *v.s.* iPLAN-BM-Hard). However, in *chaotic* traffic, where instant incentive inference is not available, the use of the soft-updating policy leads to a substantial improvement for iPLAN. As agents become more reliant on their inference of others' behaviors and intentions in a highly heterogeneous environment, the reliability and flexibility of their behavioral incentive inference become crucial, enabling them to gain a better understanding of their surroundings.

## F.3   GAT-version Behavior Incentive Encoder

We perform an additional experiment that includes an alternative approach that uses a GAT module after the behavior incentive encoder to discuss the possibility of incorporating a graphical network in behavior incentive inference that may be helpful in addressing the changing observation set. We

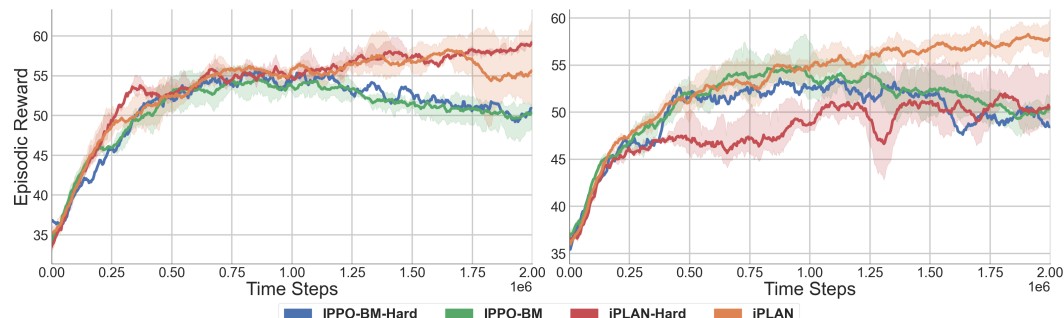

Figure 8: **Heterogeneous Highway with and without soft-updating policy:** Comparing the episodic reward in the (left) *mild* and (right) *chaotic* traffic scenarios. **Conclusion:** iPLAN (orange) that uses a soft-updating policy for behavioral incentive inference module greatly outperforms its alternative approach iPLAN-Hard (red) that uses a hard-updating policy.

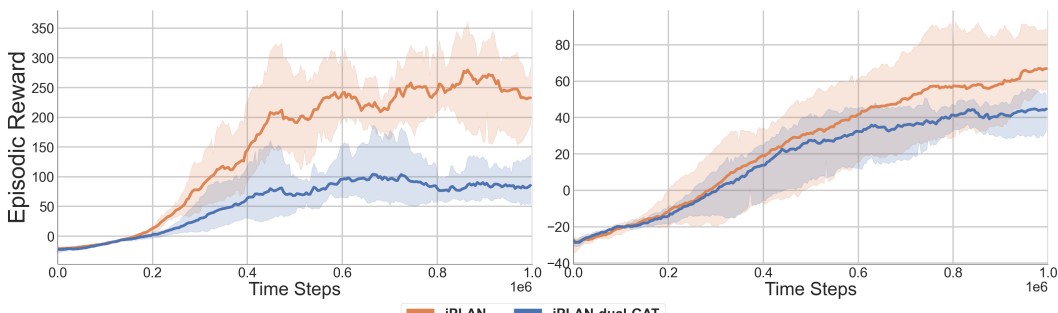

Figure 9: **Non-Cooperative Navigation with and without graphical attention network in the behavioral incentive inference module of iPLAN:** Comparing the episodic reward in the (left) *easy* and (right) *hard* scenarios. **Conclusion:** Using a graphical attention network in the behavioral incentive inference module of iPLAN does not help to improve performance.

compare this alternative approach with iPLAN in the *easy* and *hard* scenarios of Non-Cooperative Navigation.

Fig. 9 presents the comparison between the two approaches. According to the result in both *easy* and *hard* scenarios, we find that iPLAN (orange) outperforms the alternative approach that uses a GAT module inside the behavior incentive encoder, an approach named iPLAN-dual-GAT (blue), with a clear margin between two episodic reward curves. Besides, using GAT in behavior incentive inference also leads to a slower execution speed due to the additional complexity in the behavior inference module. The result shows that using GAT in the behavior incentive inference does not help to address the changing observation and additional network parameters introduced by the GAT module deteriorate the performance of iPLAN in practice.

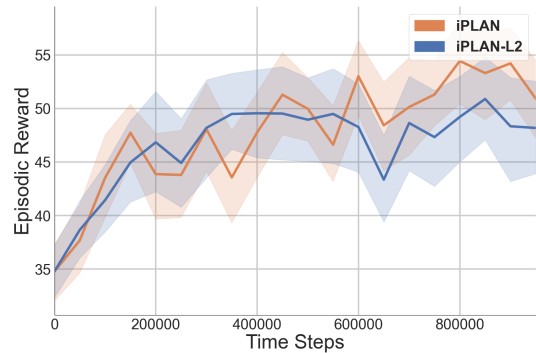

Figure 10: **Using L2 norm in loss function of behavioral and instant incentive inference module of iPLAN:** Comparing the episodic reward curve of iPLAN (orange) and iPLAN-L2 (blue) under the *chaotic* scenario of Heterogeneous Highway, with testing episodes results generated over frozen models. **Conclusion:** iPLAN using L1 norm in the loss function in two incentive inference modules performs better than iPLAN-L2 with a clear margin between two episodic reward curves.

# G   Supplementary Experiments: Inference Module Design

## G.1   L2-Norm Loss Function

Regarding the possibility of using an alternative loss function design with a different L-p norm, we modify the loss function in Eq. (4) and Eq. (5) by using L2-norm, instead of L1 norm, in the loss function for both incentive inference modules. We name this alternative approach as iPLAN-L2. The new loss functions for both incentive inference modules are:

Behavior incentive inference loss function:

$$\mathcal{J}_{\beta_i} = \min_{\mathcal{E}_i, \mathcal{D}_i} \frac{1}{Nt_h} \sum_{j=1}^{N} \left\| \mathcal{D}_i(h_{i,j}^t, \hat{\beta}_{i,j}^t) - h_{i,j}^{t+t_h} \right\|_2. \tag{6}$$

Instant incentive inference loss function:

$$\mathcal{J}_{\zeta_i} = \min_{\phi_i, \psi_i} \frac{1}{Nt_p} \sum_{j=1}^{N} \sum_{k=0}^{t_p-1} \left\| \psi_i(\mathbf{o}_i^t, \phi_i(\mathbf{o}_i^t, \hat{\boldsymbol{\beta}}_i^t, \hat{\boldsymbol{\zeta}}_i^{t-1})) - \mathbf{o}_i^{t+k+1} \right\|_2 \tag{7}$$

| Algorithm | Mean | Std |
|:---:|:---:|:---:|
| iPLAN | 53.321 | 9.490 |
| iPLAN-L2 | 48.182 | 11.921 |

Table 4: **Standard statistical test:** Standard statistical test over the episodic reward of iPLAN and iPLAN-L2 under the *chaotic* scenario of Heterogeneous Highway when training step = 950, 000. Perform standard statistical test of iPLAN and iPLAN-L2 (iPLAN using L2 norm in loss function) **Conclusion:** iPLAN using L1 norm in the loss function in two incentive inference modules performs better than iPLAN-L2

We perform evaluation experiments under the *chaotic* scenario of the Heterogeneous Highway. We train iPLAN and iPLAN-L2 models and test both models' performance over frozen models. We perform 32 testing episodes at the testing phase each time. The random seed we use is 59582679. Fig. 10 shows the episodic reward curve over all testing phases performed, while Table. 4 provides the standard statistical test results over frozen models of iPLAN and iPLAN-L2 after 950, 000 training steps. From the result, we could conclude that the current loss function design of iPLAN, i.e. using L1-norm in both incentive inference modules, leads to a better performance than the alternative approach using L2-norm in loss functions of both incentive inference modules.

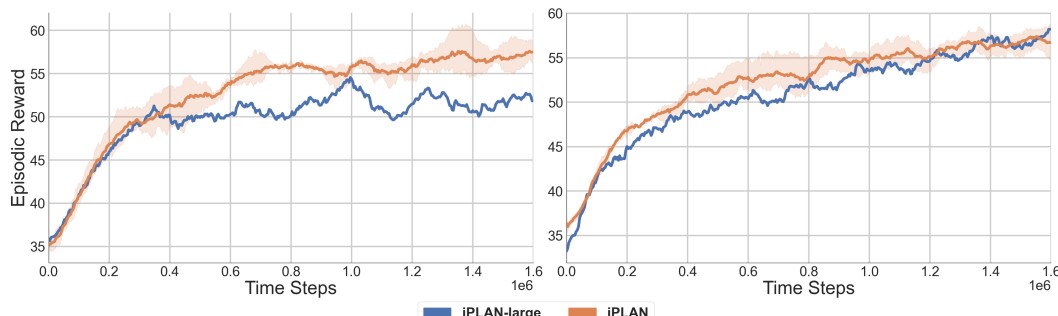

Figure 11: **Heterogeneous Highway with different learning rates for instant incentive inference module:** Comparing the episodic reward in the (left) *mild* and (right) *chaotic* traffic scenarios (with 1.6M training time steps). **Conclusion:** Using a smaller learning rate in instant incentive inference (iPLAN, orange) has a better performance in the *mild* traffic

## G.2 Weight Sharing

Regarding allowing weight sharing in iPLAN modules, we design an alternative approach of iPLAN that shares weights between different agents' behavior and instant incentive inference modules. We name this alternative approach as iPLAN-weight-sharing. We perform evaluation experiments under the *chaotic* scenario of the Heterogeneous Highway. We provide standard statistical test results over frozen models of iPLAN and iPLAN-weight-sharing at $1,200,000$ training steps. We perform 32 testing episodes at the testing phase each time. The random seed we use is $59582679$. We compute the p-values of the results by comparing the results of alternative approaches with iPLAN.

| Algorithm | Mean | Std | p-value |
|:---:|:---:|:---:|:---:|
| iPLAN | 56.540 | 10.141 | - |
| iPLAN-weight-sharing | 52.876 | 11.848 | 0.195 |

Table 5: **Standard statistical test:** Standard statistical test over the episodic reward of iPLAN and iPLAN-weight-sharing under the *chaotic* scenario of Heterogeneous Highway when training step = $1,200,000$. Perform standard statistical test of iPLAN and iPLAN-weight-sharing.

Table. 5 shows the standard statistical test results performed over frozen models after $1,200,000$ training steps. Results show that performing weight-sharing over inference modules degrades performance. This is primarily due to the inherent challenges in harmonizing policies within a diverse agent team. As discussed in our response to Weakness 2.2, even subtle disparities in controller policies can lead to significant variances in incentive inference modules. Weight sharing does not rectify this discrepancy. Furthermore, upholding distinct incentive inference modules without resorting to weight sharing effectively manages the innate diversity of the multi-agent system, making the approach more adept for intricate, heterogeneous systems.

## G.3 Supplementary Experiments: Hyper-Parameter Study

### G.3.1 Learning Rate in Instant Incentive Inference

Figure 11 compares the episodic rewards when using different learning rates for instant incentive inference. iPLAN (orange curve) uses a learning rate of $2 \times 10^{-5}$ and iPLAN-large (blue curve) uses a learning rate of $1 \times 10^{-4}$. The result shows that using a smaller learning rate in instant incentive inference has a better performance in practice.

### G.3.2 Hidden Layer Dimension in Behavioral Incentive Inference

Figure 12 presents a comparison of the effect of hidden layer dimensions used in behavior incentive inference. In this figure, we denote the alternative approach iPLAN that utilizes a hidden layer

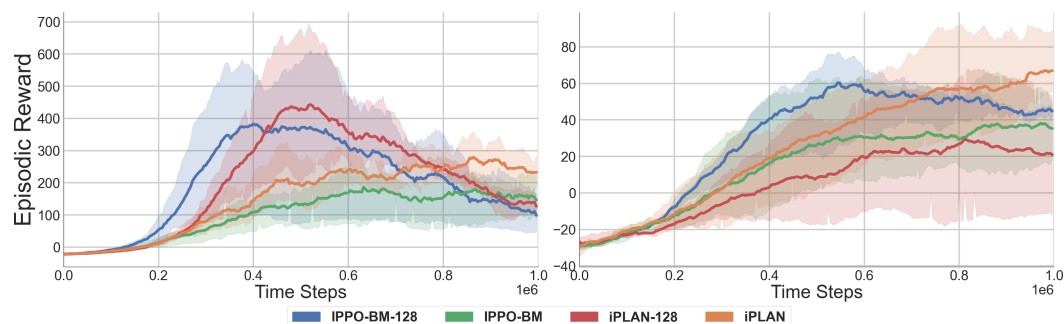

Figure 12: **Non-Cooperative Navigation with different hidden layer dimensions for behavioral incentive inference module:** Comparing the episodic reward in the (left) *easy* and (right) *hard* scenarios. **Conclusion:** Approaches like iPLAN-BM-128 (blue) and iPLAN-128 (red) that use a larger hidden layer dimension for behavioral incentive inference do not address the heterogeneity well and suffer from the overfitting problem.

dimension of 128 as iPLAN-128, and the same notation applies to the alternative approach iPLAN-BM-128 of iPLAN-BM.

In the *easy* scenario, both iPLAN-BM-128 (blue curve) and iPLAN-128 (red curve) exhibit significantly better performance than their counterparts using a hidden layer dimension of 64 in the first half of training. However, their episodic rewards experience a substantial decline in the second half, resulting in a lower ultimate episodic reward compared to iPLAN. This observation suggests that these models are overfitting in the *easy* scenario.

In the *hard* scenario, iPLAN (orange curve) outperforms iPLAN-128 (red curve) and iPLAN-BM-128 (blue curve), as the episodic reward of iPLAN-BM-128 begins to decrease when iPLAN's curve is still increasing. This phenomenon demonstrates that using a larger hidden layer dimension does not necessarily lead to performance improvement, as it can exacerbate the overfitting problem. Additionally, a larger hidden layer dimension may not effectively address the heterogeneity in a more complex and heterogeneous environment, such as the *hard* scenario.

Overall, the results indicate that carefully selecting the hidden layer dimension is crucial. While a larger dimension may offer some benefits, it can also lead to overfitting and failure in addressing the challenges posed by heterogeneity in certain scenarios.

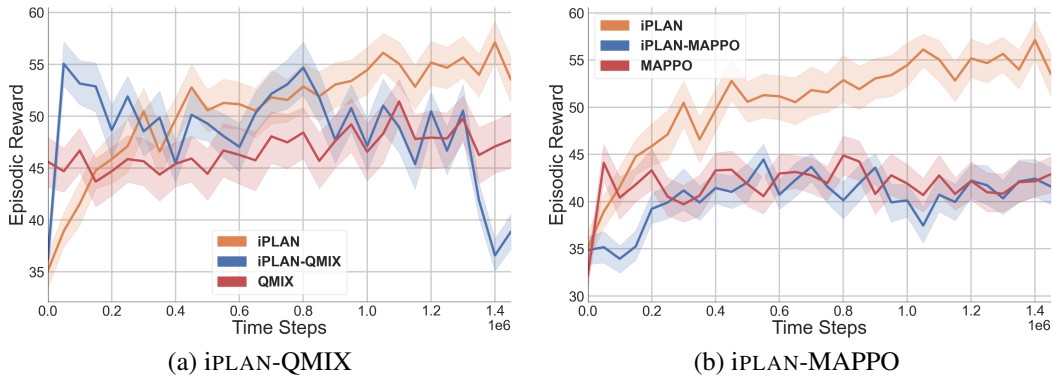

|  | (a) iPLAN-QMIX | | (b) iPLAN-MAPPO |

Figure 13: **Heterogeneous Highway with the CTDE version of iPLAN:** Comparing the episodic reward in the *chaotic* scenario with approaches incorporating iPLAN (orange) with QMIX (left) and MAPPO (right). **Conclusion:** Incorporating iPLAN with centralized credit assignment MARL approaches does not help to achieve better performance.

## H Supplementary Experiments: CTDE Version of iPLAN

Regarding evaluating iPLAN with a centralized critic (MAPPO) and a mixing network (QMIX), in the controller, we perform supplementary experiments that incorporate iPLAN incentive inference module with two CTDE MARL baselines, QMIX and MAPPO. We name the alternative approach combining QMIX and iPLAN as iPLAN-QMIX. Similarly, we name the alternative approach combining MAPPO and iPLAN as iPLAN-MAPPO. We evaluate the performance of alternative approaches under the *chaotic* scenario of the Heterogeneous Highway. We compute and visualize the episodic reward curve over a rigorous testing phase performed 32 testing episodes using frozen models. The random seed we use is 59582679.

| Algorithm | Success Rate (%) | Avg. Reward | Avg. Survival Time (# Time Steps) | Avg. Speed (m/s) |
|---|---|---|---|---|
| QMIX | $38.13 \pm 8.37$ | $54.29 \pm 3.12$ | $46.01 \pm 5.23$ | $23.50 \pm 0.30$ |
| iPLAN-QMIX | $54.38 \pm 7.79$ | $50.46 \pm 3.40$ | $64.96 \pm 3.65$ | $23.88 \pm 0.19$ |
| iPLAN | $64.38 \pm 9.12$ | $56.54 \pm 3.51$ | $74.92 \pm 4.86$ | $21.99 \pm 0.17$ |

Table 6: **Navigation metrics of QMIX, iPLAN-QMIX, and iPLAN under *chaotic* scenario of Heterogeneous Highway:** Comparing the navigation metrics of QMIX, iPLAN-QMIX, and iPLAN acquired in the *chaotic* scenario over frozen models after $1,200,000$ training time steps. **Conclusion:** iPLAN shows a better performance than the other two approaches, in terms of success rate, average episodic reward, and average survival time.

| Algorithm | Success Rate (%) | Avg. Reward | Avg. Survival Time (# Time Steps) | Avg. Speed (m/s) |
|---|---|---|---|---|
| MAPPO | $26.88 \pm 7.06$ | $43.70 \pm 3.50$ | $44.60 \pm 3.71$ | $29.93 \pm 0.02$ |
| iPLAN-MAPPO | $23.75 \pm 5.86$ | $42.22 \pm 3.09$ | $42.20 \pm 3.29$ | $29.93 \pm 0.02$ |
| iPLAN | $64.38 \pm 9.12$ | $56.54 \pm 3.51$ | $74.92 \pm 4.86$ | $21.99 \pm 0.17$ |

Table 7: **Navigation metrics of MAPPO, iPLAN-MAPPO, and iPLAN under *chaotic* scenario of Heterogeneous Highway:** Comparing the navigation metrics of MAPPO, iPLAN-MAPPO, and iPLAN acquired in the *chaotic* scenario over frozen models after $1,200,000$ training time steps. **Conclusion:** iPLAN shows a better performance than the other two approaches, in terms of success rate, average episodic reward, and average survival time.

Fig. 13 presents the episodic reward variation throughout training. We train all models for 1.5 million time steps. Fig. 13 (a) presents the result of iPLAN (orange), QMIX (red), and iPLAN-QMIX (blue). The result shows that iPLAN achieves a better overall performance, compared with the other two approaches, and the CTDE version of iPLAN, iPLAN-QMIX, does not achieve a better

performance, compared with QMIX. Fig. 13 (b) presents the result of iPLAN (orange), MAPPO (red), and iPLAN-MAPPO (blue). The result shows that iPLAN outperforms the other two approaches with a large margin between episodic reward curves, and iPLAN-MAPPO does not have a better performance compared with vanilla MAPPO.

We also compute navigation metrics over QMIX, iPLAN-QMIX, and iPLAN, and MAPPO, iPLAN-MAPPO, and iPLAN under the *chaotic* scenario. We compute results generated by 32 testing episodes over frozen models after $1,200,000$ training time steps. Table. 6 shows the results for QMIX, iPLAN-QMIX, and iPLAN and Table. 7 shows the results for MAPPO, iPLAN-MAPPO, and iPLAN. The result shows that iPLAN shows a much better performance than the other two approaches, in terms of success rate, average episodic reward, and average survival time, when evaluating the frozen model after $1,200,000$ training time steps.

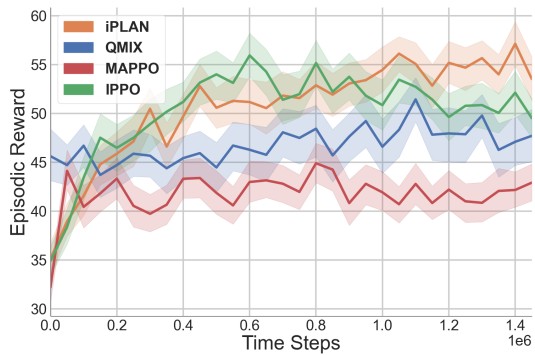

Figure 14: **Episodic reward curve of iPLAN and baselines with a rigorous testing phase.** Comparing the episodic reward curve of iPLAN (orange) and baselines, including QMIX (blue), MAPPO (red), and IPPO (green) under the *chaotic* scenario of the Heterogeneous Highway, with testing episodes results generated over frozen models. **Conclusion:** iPLAN shows a better performance than all other baselines in terms of episodic reward.

# I    Supplementary Experiments: Standard Statistical Test

To address concerns regarding performing the standard statistical tests over iPLAN and baselines, including QMIX, MAPPO, and IPPO, and perform the rigorous testing phase, we refined our codebase and performed a rigorous testing phase over frozen models every $50,000$ training step. We present the results over testing phases over 32 testing episodes throughout the training. We perform evaluation experiments under the *chaotic* scenario of the Heterogeneous Highway. The random seeds we use are 59582679, 763887655, and 312261940.

Fig. 14 shows the Episodic reward curves of iPLAN and baselines when performing a rigorous testing phase. From the result, we find that iPLAN (orange) shows a better performance than all other baselines in terms of episodic reward and there is a clear margin between iPLAN and other baselines without overlap in their error bar.

To better present the results, we provide standard statistical test results over frozen models of iPLAN and baselines, QMIX, MAPPO, and IPPO, at $200,000$ (Table. 8), $500,000$ (Table. 9), $1,000,000$ (Table. 10), and $1,450,000$ (Table. 11) training steps. We compute the p-values of the results by comparing the results of alternative approaches with iPLAN (Tables could be found on the next page).

Standard statistical test results show that iPLAN outperforms all baselines included in our paper in terms of episodic reward, and p-values ($< 0.05$) suggest results are statistically significant.

| Algorithm | Mean | Std | p-value |
|:---:|:---:|:---:|:---:|
| iPLAN | 45.866 | 11.863 | - |
| QMIX | 44.700 | 12.022 | 0.5017 |
| MAPPO | 43.317 | 10.990 | 0.1261 |
| IPPO | 46.463 | 11.700 | 0.7271 |

Table 8: **Standard statistical test:** Standard statistical test over the episodic reward generated by frozen models of iPLAN and baselines, QMIX, MAPPO, and IPPO, under the *chaotic* scenario of the Heterogeneous Highway when training step = $200,000$. Perform standard statistical test of iPLAN and iPLAN-weight-sharing.

| Algorithm | Mean | Std | p-value |
|:---:|:---:|:---:|:---:|
| iPLAN | 50.568 | 10.379 | - |
| QMIX | 44.455 | 13.008 | $4.357 \times 10^{-4}$ |
| MAPPO | 41.855 | 10.785 | $5.150 \times 10^{-8}$ |
| IPPO | 53.986 | 12.287 | $3.968 \times 10^{-2}$ |

Table 9: **Standard statistical test:** Standard statistical test over the episodic reward generated by frozen models of iPLAN and baselines, QMIX, MAPPO, and IPPO, under the *chaotic* scenario of the Heterogeneous Highway when training step = $500,000$. Perform standard statistical test of iPLAN and iPLAN-weight-sharing.

| Algorithm | Mean | Std | p-value |
|:---:|:---:|:---:|:---:|
| iPLAN | 54.445 | 11.718 | - |
| QMIX | 46.579 | 13.555 | $2.978 \times 10^{-5}$ |
| MAPPO | 41.911 | 10.192 | $2.681 \times 10^{-13}$ |
| IPPO | 50.836 | 12.023 | $3.751 \times 10^{-2}$ |

Table 10: **Standard statistical test:** Standard statistical test over the episodic reward generated by frozen models of iPLAN and baselines, QMIX, MAPPO, and IPPO, under the *chaotic* scenario of the Heterogeneous Highway when training step = $1,000,000$. Perform standard statistical test of iPLAN and iPLAN-weight-sharing.

| Algorithm | Mean | Std | p-value |
|:---:|:---:|:---:|:---:|
| iPLAN | 53.514 | 11.252 | - |
| QMIX | 47.695 | 13.002 | $1.162 \times 10^{-3}$ |
| MAPPO | 42.903 | 9.401 | $3.160 \times 10^{-11}$ |
| IPPO | 49.502 | 10.207 | $1.080 \times 10^{-2}$ |

Table 11: **Standard statistical test:** Standard statistical test over the episodic reward generated by frozen models of iPLAN and baselines, QMIX, MAPPO, and IPPO, under the *chaotic* scenario of the Heterogeneous Highway when training step = $1,450,000$. Perform standard statistical test of iPLAN and iPLAN-weight-sharing.

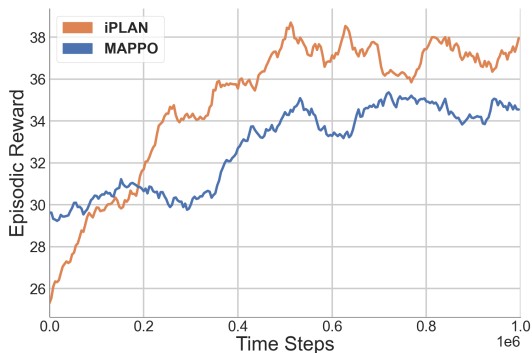

Figure 15: **iPLAN evaluation under an advanced chaotic scenario of Heterogeneous Highway:** Comparing the episodic reward of iPLAN (blue) (denote as iPLAN-VH) in the *chaotic-VH* scenario with the episodic reward of iPLAN (orange) and MAPPO (red) in the *chaotic* scenario. **Conclusion:** iPLAN shows a converging trend in the *chaotic-VH* scenario with a lower episodic reward than the other two.

## J  Supplementary Experiments: More Complex Traffic

Regarding the possibility of implementing iPLAN under a more complex domain, we perform a supplementary experiment that evaluates iPLAN under a more challenging traffic scenario. This advanced setting, which we have termed *chaotic-VH*, mirrors the existing traffic distribution of behavior-driven vehicles (*Normal*: *Aggressive*: *Conservative* = 4 : 3 : 3) in the current chaotic scenario, but with a vehicle density that is twice as dense as our previously studied chaotic scenario. Due to computational constraints during the rebuttal phase, our exploration was limited to a single random seed over 1 million time steps. Despite this, the results of iPLAN evaluated under *chaotic-VH* are promising.

Fig. 15 shows the episodic reward curve for all three experiments, while Table. 12 provides the navigation metrics over these approaches evaluated over 32 testing episodes on frozen models trained for 1 million time steps. We observe that though having a lower episodic reward curve than it used to be in *chaotic*, iPLAN outperforms MAPPO in iPLAN-VH, in terms of episodic reward curve, average episode length, and success rate.

| Algorithm | Success Rate (%) | Avg. Reward | Avg. Survival Time (# Time Steps) | Avg. Speed ($m/s$) |
|---|---|---|---|---|
| iPLAN | $50.63 \pm 9.33$ | $41.15 \pm 4.38$ | $56.19 \pm 6.62$ | $19.77 \pm 0.88$ |
| MAPPO | $18.13 \pm 4.37$ | $32.49 \pm 2.80$ | $35.80 \pm 3.42$ | $24.02 \pm 0.92$ |

Table 12: **Navigation metrics of iPLAN and MAPPO under an advanced chaotic scenario of Heterogeneous Highway:** Comparing the navigation metrics of iPLAN and MAPPO acquired in the *chaotic-VH* scenario **Conclusion:** iPLAN shows a promising performance under the *chaotic-VH* scenario as it has better performance than MAPPO.

# K  Further Discussion of Empirical Results

**Centralized versus Decentralized Training Regime**. In this work, we operated in the decentralized training regime, based on the assumption that agents should learn navigation policies in a DTDE manner without centralization in training. Empirically, we find that CTDE MARL approaches perform worse as the environmental heterogeneity increases due to the absence of consensus among agents in heterogeneous environments. On the other hand, the awareness of opponents' strategies becomes more important in agents' decision-making when the environment is heterogeneous, especially the awareness of agents' instant reactions to surroundings. This need for increased awareness makes intent-aware distributed MARL algorithms perform better in these environments.

To further investigate the empirical performance of CTDE and DTDE approaches under our problem setting, we conduct experiments integrating two incentive inference modules of iPLAN with two CTDE approaches, QMIX and MAPPO, and compare its performance with iPLAN and other baselines. We include the experiment details and results in Appendix H. Results show that integrating iPLAN inference module in CTDE approaches does not help to achieve a better performance in the *chaotic* scenario of the Heterogeneous Highway than the current DTDE version of iPLAN.

**Decoupled Incentive Inference**. Individually, the incentives yield some benefit over a baseline controller. For example, we find that both the behavior and instant incentive inference modules individually help to achieve a higher reward, especially in more heterogeneous environments (See Figure 2). However, our system works best when both incentives are jointly activated, for example in Table 1, we find that the success rate drops significantly for iPLAN-GAT, compared to iPLAN ($61.88\%$ versus $67.81\%$). This clearly indicates autonomous vehicles need the behavior incentive module to survive in the more heterogeneous chaotic traffic scenario.

