# OpenReview forum: "Intent-Aware Planning in Heterogeneous Traffic via Distributed Multi-Agent Reinforcement Learning"
_robot-learning.org/CoRL/2023/Conference — CoRL 2023 Oral_

### Official Review · Reviewer_vQKv · 2023-07-01

**Confidence:** 3
**Originality:** Very Good
**Technical Quality:** Very Good
**Clarity Of Presentation:** Very Good
**Impact:** 3

**Recommendation:**

Strong Accept: I recommend accepting the paper and will argue for my recommendation even if other reviewers hold a different opinion.

**Review:**

## Strengths

* The problem setting is very clearly stated. I like the framing of the problem as a HiP-POSG. Overall, the formalisms are precise and the notation, while extensive, is helpful in my opinion.
* The problem setting is also incredibly difficult. Getting something to work at all with multiple agents and partial observability is a feat.
* The related work is very thorough and the paper seems well positioned in the literature.
* Besides a few details mentioned below, the figures are very helpful.


## Weaknesses

* 3 random seeds is not sufficient to draw meaningful conclusions.
* Relatedly, the empirical results provide only tentative support for the main hypothesis that decoupling behavioral and instant incentive inference is better than a unified inference scheme. In particular, the performance of iPlan versus IPPO-GAT does not seem convincingly better.
* Although a graph attention network is used to model instant interactions during instant incentive inference, my understanding is that each agent’s observation set is otherwise fixed in terms of the opponents that it is tracking. A more realistic setting would have the observation set change, and the model would need to be updated accordingly -- for example, the behavior incentive inference model may also need to be a graph neural network. I don’t think the model needs to be changed for this paper, but I do think this assumption should be clarified. I also think that this criticism of other works should be applied to this work as well: “Other works… require the involvement of the same group of agents between episodes, which is not practical in the real world.”
* In Figure 1 and part of the text, the behavior incentive inference model is said to “reconstruct” historical states. I think the use of the word “reconstruct” here is quite confusing. As far as I understand (from L203 in particular), the model is actually _predicting_ future states from earlier states. Alternatively, we could say that it is reconstructing _trajectories_, rather than states, which made me think that individual states are being auto-encoded (which wouldn’t make sense). I was especially confused in Figure 1 because it appears that single time steps are being reconstructed, since only $t$ appears in the orange box.
* The GIFs in the supplementary material are helpful to understand the environments tested, but they are also a bit overwhelming without context, especially given that multiple animations are concatenated together and each individually is quite busy. It would be very helpful to have the video draw the viewer’s attention to the important parts.
* I like the separation between behavioral incentives and instant incentives, but there is a third component of the hidden state -- vehicle type / kinematics -- and it’s not clear where that component comes into play. I wonder if a third “stream” would make sense. For example, incentives drift over time (fast and slow), but kinematic constraints are almost always constant. I also want to better understand where this component manifests in the current architecture. Is it one or both of the incentive streams?
* (I wrote the following, but then I realized that my concern was addressed in the experiments with the inclusion of randomly-behaving agents. I’m including the concern anyway in case the authors want to head it off earlier in the paper.) Other decentralized approaches that use parameter sharing or global rewards are criticized in the introduction because they “introduce strong and unrealistic assumptions for applications like autonomous driving.” I am not completely convinced that these mechanisms require stronger assumptions than the ones required here. This work assumes that all of the agents are being trained using the same MARL approach. This posits that all cars on the road are autonomous and that there is one autonomous driving company that determined the software on those cars. As far as I understand, parameter sharing and global rewards would only require the same assumption. Perhaps the approach in this work could be extended to a setting where not all agents are using the same MARL algorithm, but we would need to see experimental evidence for that (nevermind!).

### Minor
* Equation 1 uses $\mathcal{F}_i$ but the subsequent text uses $f_i$. Similarly there is some switching between $\mathcal{R}_i$ and $r_i$.
* There are a few awkward word choices in the paragraph L207-216: “temporal relation”, “blank prior”, “authentic estimation”.

**Quality Of The Limitations Section:**

Limitations are addressed clearly

**Questions For Rebuttal:**

1. Can you clarify the relationship between the environments used in experiments and those in prior work? To what extent are the environments new or changed in this work?
2. Is my understanding about the fixed set of agent observations mentioned above correct?
3. The main disadvantage of centralized approaches versus decentralized ones seems to be practical infeasibility. In the experiments though, feasibility concerns can be cast aside because we’re in a simple simulation setting. Nonetheless, we see that the proposed decentralized approach actually does better than centralized baselines. What should we conclude from this? As a thought experiment, if we ignore feasibility concerns, would it be possible to make a centralized version of iPlan that actually performs better than any other method in this paper? Or is there something about decentralization that actually improves performance?
4. Would it be accurate to say IPPO-GAT-BM = iPlan? I ask because I am wondering if IPPO-GAT and IPPO-BM are truly ablations of iPlan, or more like extensions of IPPO. My preference would be true ablations.
5. Can you comment on the proposed approach from a Theory of Mind perspective? My understanding is that the ego agent reasons about opponent’s and opponent’s opponents. Does the “recursion” stop there? Or is the system somehow able to reason about opponents reasoning about opponents reasoning about opponents and so on?

**Robotics Focus:**

Highly relevant to robotics but no hardware experiments

**Summary Of Paper:**

This paper considers autonomous driving in a decentralized multi-agent setting where agents vary in their instant and behavioral incentives. The ego agent is unaware of these incentives but can estimate them from observations. A two-stream architecture for incentive inference is proposed. Experiments in two non-cooperative simplified autonomous driving domains compare to several baselines and ablations.


**Summary Of Recommendation:**

Overall, the paper is very well written and carefully done. If the results were a bit stronger, particularly in terms of including more trials (random seeds), I would consider recommending strong accept.

### After Rebuttal

The original submission was solid, and the paper was improved substantially during the rebuttal period. I appreciate the engagement of the authors and the other reviewers.

---

> ### Author Response · Authors · 2023-08-10
> **Response to Reviewer vQKv [Part 1]**
>
> We would like to express our gratitude to the reviewer for dedicating their valuable time and effort towards reviewing our manuscript and helping us to strengthen the manuscript. We deeply appreciate the feedback provided, and we have provided thorough responses to all the reviewer's questions below.
>
> ### Response to Weaknesses
> > **Weakness 1:** 3 random seeds is not sufficient to draw meaningful conclusions.
>
> **Response to Weakness 1:** Thank you for bringing this issue to our attention. Given the constraints on computational resources during the rebuttal phase, we expanded our experiments to include 2 additional random seeds, while we have to shorten the training period from 2M time steps to 1.5M time steps. Consequently, we have updated Fig. 2 in our paper to reflect the new episodic reward curves, which is now derived from a comprehensive set of 5 random seeds.
>
> > **Weakness 2:** Relatedly, the empirical results provide only tentative support for the main hypothesis that decoupling behavioral and instant incentive inference is better than a unified inference scheme. In particular, the performance of iPlan versus IPPO-GAT does not seem convincingly better.
>
> **Response to Weakness 2:** We agree with your observations regarding the performance difference between iPLAN and IPPO-GAT in both scenarios of Heterogeneous Highway. We explain below why that is so. That being said, it's important to emphasize that iPLAN considerably outshines IPPO-GAT in both easy and hard scenarios of Non-Cooperative Navigation.
>
> The contrasting performances of iPLAN and IPPO-GAT across these environments can be attributed to their differing problem settings. Notably, Non-Cooperative Navigation presents a more diverse kinematic model for agents. In Non-Cooperative Navigation, agents vary in their kinematics and shape, resulting in a more heterogeneous multi-agent system, which iPLAN is designed for. Besides, the complexity of Non-Cooperative Navigation is relatively subdued compared to Heterogeneous Highway, which hosts 6-7 distinct entities versus the latter's 55. The richer variety in agent behavior models combined with a less complex multi-agent system emphasizes the agents' reliance on long-term behavioral incentives inference of opponents over short-term instant incentive inference, favoring methods that harness behavioral incentive inferences.
>
> Conversely, in Heterogeneous Highway, all controllable and uncontrollable vehicle agents operate under a unified low-level kinematic and entity model. This makes the system more homogeneous. The primary source of heterogeneity stems from the diverse behavior-driven models controlling vehicles that are uncontrollable by MARL algorithms. Additionally, given the intricate and dynamic nature of the traffic scenarios here, agents tend to prioritize their adeptness in predicting instant incentives of opponents over long-term behavioral incentives inference in their decision-making, thereby favoring methodologies equipped with instant incentive inference modules.

---

> > ### Author Response · Authors · 2023-08-10
> > **Response to Reviewer vQKv [Part 2]**
> >
> > > **Weakness 3.1:** Although a graph attention network is used to model instant interactions during instant incentive inference, my understanding is that each agent’s observation set is otherwise fixed in terms of the opponents that it is tracking.
> >
> > **Response to Weakness 3.1:** Thank you for sharing your perspective on our observation settings. However, it seems there might be a misunderstanding regarding observation setting in our paper. Our problem operates within a partially observable environment, meaning each agent can only access data about other agents or entities within their observation scope. This results in a dynamic observation set that varies depending on opponents location within the observation scope at each time step.
> >
> > In our environment setting:
> >
> > 1. The environment provides the total number of vehicles present, a number that remains unchanged throughout an episode. Each vehicle is assigned a unique identifier by the environment.
> > 2. Every agent maintains a record of past observations for each vehicle in the environment. This record starts empty without any observation or any identifier of any vehicle at the onset of an episode.
> > 3. As time progresses, agents update their records based on the vehicles appeared within their observation range. Depending on the appearance order of every other vehicle in ego agent's observation scope, the storage order in this record is changing. At each time step, if a previously observed vehicle moves out of ego agent's observation scope, its record is set to 0 at this time step. Conversely, when a new vehicle enters an agent's observation scope, a new record, identifiable by this new vehicle's unique ID, is established for it, with a blank observation history before this time step.
> >
> > From this system, several conclusions can be drawn about the dynamic nature of our observation set:
> >
> > 1. Given the environment's partial observability, not all opponents are discernible at any given time.
> > 2. The sequence in which opponents become observable varies. For example, some agents may become observable earlier than others, based on the order in which they enter the observation range. This order is varying between episodes.
> > 3. The observational records for a particular opponent, as kept by the ego agent, can differ based on the duration and frequency with which that opponent stays within the agent's observational scope.
> >
> > > **Weakness 3.2:** A more realistic setting would have the observation set change, and the model would need to be updated accordingly -- for example, the behavior incentive inference model may also need to be a graph neural network. I don’t think the model needs to be changed for this paper, but I do think this assumption should be clarified.
> >
> > **Response to Weakness 3.2:**
> > We value your suggestion on enhancing our behavioral incentive inference within the context of our observation setting. However, implementing such a change would conflict with our foundational assumptions about incentives. As detailed in Section 3 of our paper (lines 156-170), our framework operates within a partially observable environment, leading to a dynamic observation set. To navigate this, we employ instant incentive inference, interpreting it as agents' real-time responses to their environment, mirrored by the evolving observation set.
> >
> > Further, our research highlights persistent factors influencing an agent's driving behavior. For instance, a consistently aggressive driver often prefers higher speeds and exhibits risk-taking tendencies, such as choosing to overtake rather than decelerate when faced with a slower vehicle ahead. We term these underlying tendencies "behavioral incentives," presuming them to be consistent traits for individual drivers. Employing the Graph Attention Networks (GAT) to target fluctuating elements in driver trajectories, rather than these consistent behavioral facets, would introduce a conflict into our foundational assumptions for driving behavior modeling.
> >
> > Regarding your proposal of adapting our behavioral incentive inference model using a graph neural network, we conducted a supplementary experiment where we integrated a GAT module after our behavioral incentive encoder to evaluate any potential impact arising from the variable observation set. We present our results in Appendix H.1 and find that the inclusion of the GAT didn't improve performance. For a comprehensive understanding, please refer to the episodic reward curve and accompanying discussion located in Appendix H.1 of our paper.

---

> > > ### Author Response · Authors · 2023-08-10
> > > **Response to Reviewer vQKv [Part 3]**
> > >
> > > > **Weakness 3.3:** I also think that this criticism of other works should be applied to this work as well: “Other works… require the involvement of the same group of agents between episodes, which is not practical in the real world.”
> > >
> > > **Response to Weakness 3.3:**
> > >
> > > We believe there's some misunderstanding of our approach. iPLAN resets agents at the beginning of each episode, devoid of any memories of past episodes. Agents are positioned to treat every opponent as a fresh entity, mandating them to infer this opponent's behavior incentives from a limited set of interactions within the current episode alone. This means opponent #1 in the present episode is distinct from opponent #1 from any prior episode. We believe this setup more closely mirrors real-world scenarios. In genuine driving situations, drivers routinely need to quickly understand and adapt to the behaviors of vehicles they likely haven't encountered before.
> > >
> > > ***This is different from existing literature*** [1, 2, 3]. For instance, the opponent #1 in a current interaction would be the *same* entity as the opponent #1 from past episodes. This context defines "the involvement of the same group of agents between episodes". Under this premise, agents could benefit from the knowledge from their historical interaction with opponents, like they could develop a more refined behavior model of opponents when engaging interactions within multi-agent systems.
> > >
> > > One of our paper's ***pivotal contributions*** is precisely this: ***leveraging behavioral incentive inference to empower agents*** to anticipate an opponent's behavioral patterns within a constrained number of steps. We think this problem setting helps to agents perform a more reliable behavior modeling under real-world traffic scnearios.
> > >
> > > [1] Xie, Annie, et al. "Learning latent representations to influence multi-agent interaction." Conference on robot learning. PMLR, 2021.
> > >
> > > [2] Wang, Woodrow Zhouyuan, et al. "Influencing towards stable multi-agent interactions." Conference on robot learning. PMLR, 2022.
> > >
> > > [3] Parekh, Sagar, Soheil Habibian, and Dylan P. Losey. "RILI: Robustly influencing latent intent." 2022 IEEE/RSJ International Conference on Intelligent Robots and Systems (IROS). IEEE, 2022.
> > >
> > > > **Weakness 4:** In Figure 1 and part of the text, the behavior incentive inference model is said to “reconstruct” historical states. I think the use of the word “reconstruct” here is quite confusing. As far as I understand (from L203 in particular), the model is actually predicting future states from earlier states. Alternatively, we could say that it is reconstructing trajectories, rather than states, which made me think that individual states are being auto-encoded (which wouldn’t make sense). I was especially confused in Figure 1 because it appears that single time steps are being reconstructed, since only appears in the orange box.
> > >
> > > **Response to Weakness 4:** Thank you for highlighting the confusion in the terminology. Indeed, the term "predict" better encapsulates the process we have implemented. We have updated Figure 1 accordingly and have modified the terminology in the paper.
> > >
> > > > **Weakness 5:** The GIFs in the supplementary material are helpful to understand the environments tested, but they are also a bit overwhelming without context, especially given that multiple animations are concatenated together and each individually is quite busy. It would be very helpful to have the video draw the viewer’s attention to the important parts.
> > >
> > > **Response to Weakness 5:** Thank you for drawing attention to the ambiguities in our GIF animations. We've accentuated the ego agent, behavior-driven vehicles, and pivotal events in the updated animations provided in the attachment. Please review and share your feedback!

---

> > > > ### Author Response · Authors · 2023-08-10
> > > > **Response to Reviewer vQKv [Part 4]**
> > > >
> > > > > **Weakness 6:** I like the separation between behavioral incentives and instant incentives, but there is a third component of the hidden state -- vehicle type / kinematics -- and it’s not clear where that component comes into play. I wonder if a third “stream” would make sense. For example, incentives drift over time (fast and slow), but kinematic constraints are almost always constant. I also want to better understand where this component manifests in the current architecture. Is it one or both of the incentive streams?
> > > >
> > > > **Response to Weakness 6:** Thanks for your insight. It suggests a promising direction for our future work. In our current paper, we have integrated the effect of "vehicle type/kinematics" into what we term a vehicle's "behavioral incentive."
> > > >
> > > > In Non-Cooperative Navigation, the distinguishing factors between agents predominantly lie in their size and acceleration capabilities. These attributes naturally takes as a part of "vehicle type/kinematics". Hence, when deriving behavioral incentives in this setting, agents are effectively addressing the impact of these "vehicle type/kinematics" components.
> > > >
> > > > Regarding the Heterogeneous Highway environment, our simulator's constraints don't allow for a diverse range of vehicle types, assuming a uniform size for all. As for vehicle kinematics, we perceive all vehicles' motions follow the same low-level kinematics model, while drivers performing dfiferent driving behavior vary in their preference in driving speed, or accelerations, both of which could be taken as an integral part of driving behavior. For instance, trajectories from vehicles that boast greater acceleration and top speeds than most, yet are controlled by a driver with standard behavior, can alternatively be seen as those of typical vehicles managed by aggressive drivers who favor faster speeds and brisk acceleration.
> > > >
> > > > Our current simulator does have its limitations, restricting our ability to delve deeply into this particular component. However, we see a future where we'll be executing tests on more intricate traffic situations, incorporating a broader spectrum of vehicle types and kinematics to simulate real-world traffic scenarios.
> > > >
> > > > ### Response to Minor Weaknesses
> > > > > **Minor Weakness 1:** Equation 1 uses $\mathcal{F}_i$ but the subsequent text uses $f_i$. Similarly there is some switching between $\mathcal{R}_i$ and $r_i$.
> > > > >
> > > > **Response to Minor Weakness 1:** Thank you for denoting our ambigurity in our symbol usages. We have now used $f_i$ and $r_i$ only everywhere.
> > > >
> > > > > **Minor Weakness 2:** There are a few awkward word choices in the paragraph L207-216: “temporal relation”, “blank prior”, “authentic estimation”.
> > > > >
> > > > **Response to Minor Weakness 2:** Thanks for pointing this out. We rephrased this paragraphed as follows. Please let us know if this is better. Replaced words are **bolded**:
> > > >
> > > > To capture the **sequential nature** within opponents' state observation sequences, the encoder $\mathcal{E_i}$ employs a recurrent network that processes $h_{ij}^t$ as a time series. This produces a new estimate of the behavioral incentive of opponent $j$. As insights from cognitive science suggest, human social focus remains relatively stable. Thus, we interpret the behavioral incentive inference for opponents as a gradual process, converging towards the true behavioral incentives of opponents without abrupt transitions between updates. Starting with an initial **neutral estimation** of opponents' behavioral latent states, agents propose new estimates for opponents' behavioral incentives at each time step. However, they employ a gentle update strategy, using an additional coefficient $\eta$, to refine the behavioral incentive estimates. This approach allows agents to produce more **accurate estimates** of opponents' behavioral incentives, managing the variability between consecutive updates, which in turn ensures more stable agent policies.

---

> > > > > ### Author Response · Authors · 2023-08-10
> > > > > **Response to Reviewer vQKv [Part 5]**
> > > > >
> > > > > ### Response to Questions for Rebuttal
> > > > >
> > > > > >  **Question 1:** Can you clarify the relationship between the environments used in experiments and those in prior work? To what extent are the environments new or changed in this work?
> > > > >
> > > > > **Response to Question 1:** Thank you for highlighting the ambiguity in our environment description. Here, we will delve deeper to address any lingering confusion concerning the uniqueness of our environments.
> > > > >
> > > > > **Non-Cooperative Navigation** is a variation of the Cooperative Navigation scenario in the Multi-agent Particle Environment. The salient distinctions between them are:
> > > > > 1. In Non-Cooperative Navigation, each agent obtains a unique reward based on its proximity to the nearest landmark. Conversely, in Cooperative Navigation, all agents share a global reward linked to the collective distance from agents to their closest landmarks.
> > > > > 2.  Non-Cooperative Navigation portrays a heterogeneous multi-agent system that agents are heterogeneous in their kinematics and shapes. In Cooperative Navigation, all agents are homogeneous in their kinematics and shapes.
> > > > > 3. For the advanced Non-Cooperative Navigation scenario, we've embedded uncontrollable, random agents to augment environmental heterogeneity. In contrast, all agents in Cooperative Navigation are controllable by MARL algorithms.
> > > > >
> > > > > **Heterogeneous Highway** is our take on an expanded multi-agent adaptation of the Highway-Env. Distinguishing features between Heterogeneous Highway and analogous environments from prior research encompass:
> > > > > 1. We've cast our challenge in a non-cooperative frame, wherein each agent garners a specific reward tied to its velocity and collision incidents, specifically for iPLAN / IPPO. This departs from the majority of previous work which hasn't ventured to define problems or design algorithms in this context.
> > > > > 2. Our environment simulates complex traffic situations, boasting a range of behavior-driven vehicles within a multi-agent milieu. Notably, both controllable and uncontrollable agents populate this landscape. In contrast, earlier studies like [1] limit their scope to single-agent settings amidst behavior-driven vehicles.
> > > > > 3. We model agents engage with unknown opponents in our environment setting and they must learn to infer opponents' intentions without much prior knowledge. Prior works in this area, like [2, 3], operates on the assumption of agents interacting with a consistent group — a setup less reflective of real-world dynamics. For a deeper dive into our assumptions within this environment, we direct you to our response to Weakness 3.3.
> > > > >
> > > > > [1] Mavrogiannis, Angelos, Rohan Chandra, and Dinesh Manocha. "B-gap: Behavior-guided action prediction for autonomous navigation." arXiv preprint arXiv:2011.03748 1.2 (2020).
> > > > >
> > > > > [2] Xie, Annie, et al. "Learning latent representations to influence multi-agent interaction." Conference on robot learning. PMLR, 2021.
> > > > >
> > > > > [3] Wang, Woodrow Zhouyuan, et al. "Influencing towards stable multi-agent interactions." Conference on robot learning. PMLR, 2022.
> > > > >
> > > > > >  **Question 2:** Is my understanding about the fixed set of agent observations mentioned above correct?
> > > > >
> > > > > **Response to Question 2:** Apologies for any confusion, but there seems to be a misunderstanding regarding agent observations. We addressed this concern in our response to Weakness 3.1. Kindly refer to that section for a comprehensive explanation.

---

> > > > > > ### Author Response · Authors · 2023-08-10
> > > > > > **Response to Reviewer vQKv [Part 6]**
> > > > > >
> > > > > > >  **Question 3:** The main disadvantage of centralized approaches versus decentralized ones seems to be practical infeasibility. In the experiments though, feasibility concerns can be cast aside because we’re in a simple simulation setting. Nonetheless, we see that the proposed decentralized approach actually does better than centralized baselines. What should we conclude from this? As a thought experiment, if we ignore feasibility concerns, would it be possible to make a centralized version of iPlan that actually performs better than any other method in this paper? Or is there something about decentralization that actually improves performance?
> > > > > >
> > > > > > **Response to Question 3:** Thank you for raising this insightful question. We frame our problem within a distributed setting, where the multi-agent system exhibits heterogeneity and intricate behavioral patterns. This context is inherently suited for fully distributed MARL algorithms over CTDE MARL algorithms.
> > > > > >
> > > > > > The inherent centralized credit assignment module in CTDE MARL algorithms can encounter difficulties in effectively distributing credits over heterogeneous agents during training under our problem setting. Moreover, as the multi-agent system escalates, algorithms with centralized credit assignment mechanisms tend to degrade in performance due to challenges in accurately allocating credit among a burgeoning set of agents. Through this perspective, distributed MARL algorithms generally exhibit superior results, especially in expansive multi-agent systems.
> > > > > >
> > > > > > Regarding the hypothetical scenario you posed, the answer is contingent on the specific problem setting. Broadly speaking, a centralized rendition of iPLAN could potentially surpass the current distributed variant in scenarios featuring a modest number of homogeneous agents. Conversely, the distributed version of iPLAN is likely to be more effective in systems characterized by a large, heterogeneous group of agents.
> > > > > >
> > > > > > To give us more insight, we've conducted experiments integrating our iPLAN module with certain CTDE MARL algorithms, including QMIX and MAPPO. Detailed results can be found below as well as in Appendix H.2. Our observations indicate that incorporating iPLAN modules with CTDE approaches, like QMIX and MAPPO, does not help to overcome the failure of centralized credit assigment modules in both CTDE approaches, and distributed version of iPLAN has a better performance in the MARL problem setting with complex, heterogeneous multi-agent system, like Heterogeneous Highway.
> > > > > >
> > > > > > Navigation Metrics over QMIX, iPLAN-QMIX (Centralized version of iPLAN based on QMIX), and iPLAN.
> > > > > > |  Algorithm  | Success Rate | Avg. Reward | Avg. Survival Time | Avg. Speed
> > > > > > | ----------- | ------- | ------- | ------- | ------- |
> > > > > > | QMIX | 38.13 $\pm$ 8.37 | 54.29 $\pm$ 3.12 | 46.01 $\pm$ 5.23  | 23.50 $\pm$ 0.30  |
> > > > > > | iPLAN-QMIX | 54.38 $\pm$ 7.79 | 50.46 $\pm$ 3.40 | 64.96 $\pm$ 3.65  | 23.88 $\pm$ 0.19  |
> > > > > > | iPLAN | 64.38 $\pm$ 9.12 | 56.54 $\pm$ 3.51 | 74.92 $\pm$ 4.86  | 21.99 $\pm$ 0.17  |
> > > > > >
> > > > > > Navigation Metrics over MAPPO, iPLAN-MAPPO (Centralized version of iPLAN combining MAPPO), and iPLAN.
> > > > > > |  Algorithm  | Success Rate | Avg. Reward | Avg. Survival Time | Avg. Speed
> > > > > > | ----------- | ------- | ------- | ------- | ------- |
> > > > > > | MAPPO | 26.88 $\pm$ 7.06 | 43.70 $\pm$ 3.50 | 44.60 $\pm$ 3.71  | 29.93 $\pm$ 0.02  |
> > > > > > | iPLAN-MAPPO | 23.75 $\pm$ 5.86 | 42.22 $\pm$ 3.09 | 42.20 $\pm$ 3.29 | 29.93 $\pm$ 0.02  |
> > > > > > | iPLAN | 64.38 $\pm$ 9.12 | 56.54 $\pm$ 3.51 | 74.92 $\pm$ 4.86  | 21.99 $\pm$ 0.17  |
> > > > > >
> > > > > > >  **Question 4:** Would it be accurate to say IPPO-GAT-BM = iPlan? I ask because I am wondering if IPPO-GAT and IPPO-BM are truly ablations of iPlan, or more like extensions of IPPO. My preference would be true ablations.
> > > > > >
> > > > > > **Response to Question 4:** Thank you for your feedback. We apologize for the confusion. Yes, IPPO-GAT and IPPO-BM are true ablations of iPLAN and IPPO-GAT-BM = iPLAN.
> > > > > >
> > > > > > * "-GAT" signifies the use of the instant incentive inference module, which is based on graphical attnetion network (GAT).
> > > > > > * "-BM" indicates the use of the behavioral incentive inference module, which performs behavior modeling (BM) tasks.
> > > > > >
> > > > > > We clarified this in the revised paper and have renamed these ablations more appropriately as iPLAN-GAT (Previous: IPPO-GAT) and iPLAN-BM (Previous: IPPO-BM) to indicate variants that exclusively use the instant incentive and the behavior incentives, respectively.

---

> > > > > > > ### Author Response · Authors · 2023-08-10
> > > > > > > **Response to Reviewer vQKv [Part 7]**
> > > > > > >
> > > > > > > >  **Question 5:** Can you comment on the proposed approach from a Theory of Mind perspective? My understanding is that the ego agent reasons about opponent’s and opponent’s opponents. Does the “recursion” stop there? Or is the system somehow able to reason about opponents reasoning about opponents reasoning about opponents and so on?
> > > > > > >
> > > > > > > **Response to Question 5:** We appreciate you raising this valuable question to discuss. Though the the apporach proposed in our paper is deeply rooted in the Theory of Mind (ToM), it's hard to analyze agents' reasoning process from this prespective and present a clear answer to where the “recursion” of reasoning stops.
> > > > > > >
> > > > > > > Similar to the problem setting in [1, 2, 3], we assume that the existance and behavior of other agents play a role in determining a specific agent's policies. Our approach is designed to model this effect on agents' policies, which performs "reasoning about opponents" in practices.
> > > > > > >
> > > > > > > From agents in a heterogeneous MARL environment learning their opponents' models, it seems they could perform at least some level of second-order reasoning. The ego agent is not just considering the actions of other agents, but also how those agents might be interpreting the ego agent's and opponent’s opponents' actions, which could be taken as second-order reasoning.
> > > > > > >
> > > > > > > However, whether this inference system goes beyond this to third-order reasoning or deeper could not be explicitly analyzed for now. It's true that ego agents' third-order neighbors' existance and behaviors have some effects on ego agent's neighboring opponents and opponents' opponents, but this effect is hard to track and anaylze from highly coupled behaviors in a multi-agent system. Besides, deep recursive reasoning can become computationally expensive and may not always provide significant benefits in many real-world scenarios, so we think first-order and second-order reasoning might be sufficient to make effective decisions in many practical situations.
> > > > > > >
> > > > > > > [1] Xie, Annie, et al. "Learning latent representations to influence multi-agent interaction." Conference on robot learning. PMLR, 2021.
> > > > > > >
> > > > > > > [2] Wang, Woodrow Zhouyuan, et al. "Influencing towards stable multi-agent interactions." Conference on robot learning. PMLR, 2022.
> > > > > > >
> > > > > > > [3] Papoudakis, Georgios, Filippos Christianos, and Stefano Albrecht. "Agent modelling under partial observability for deep reinforcement learning." Advances in Neural Information Processing Systems 34 (2021): 19210-19222.

---

> ### Author Response · Authors · 2023-08-13
> **Gentle Reminder to Reviewer vQKv**
>
> Dear Reviewer vQKv ,
>
> We appreciate the time and effort you've dedicated to reviewing our work. Your insights have been extremely useful, and we have taken care to address each concern raised. Should there be any additional points of discussion, we would be grateful for the opportunity to engage further.
>
> In light of the revisions and clarifications, we kindly and humbly request you to reconsider the evaluation of our work.
>
> Regards, Authors

---

### Official Review · Reviewer_qQsj · 2023-07-18

**Confidence:** 4
**Originality:** Good
**Technical Quality:** Very Good
**Clarity Of Presentation:** Excellent
**Impact:** 3

**Recommendation:**

Weak Accept: I recommend accepting the paper, but will not argue for my recommendation if the majority of other reviewers have a different opinion.

**Review:**

This work considers a relevant problem for the community, and focuses on an interesting approach to tackle it: driving style estimation (i.e., essentially agent modelling) to enable and rely on predictions of the other agents’ short-term actions. I found the paper overall pleasant to read, and in particular appreciated the rigor (yet readability) of Section 3. That being said, I have several concerns with the actual premises of the work and with the results, which I believe call for more time to refine the overall story and the evaluation.

In the introduction, authors state that existing MARL algorithms can be classified as either centralized (e.g., QMIX or MAPPO) or distributed (e.g., iPPO). However, this does not seem to be what is generally accepted in the field, as far as I know. “Centralized” approaches like QMIX/MAPPO only rely on information centralization during training (e.g., mixing network, or centralized critic) but remain entirely decentralized at runtime. Instead, one generally refers to QMIX/MAPPO as “centralized training, decentralized execution (CTDE),” a paradigm that could be well-suited to many AV scenarios since training is often done in simulation (where centralization is not an issue), while offering the desired purely decentralized execution capabilities and scalability at deployment time. Therefore, I was first confused by this statement and premise.

Authors then follow up by stating that distributed approaches like iPPO often rely on tricks like parameters sharing, which cannot apply in AV path planning (but without substantiating why). However, the whole approach then described in this work, which consists of three modules (behavioral incentive inference, instant incentive inference, and controller/policy), has two of these modules completely symmetric and agent-agnostic, thus naturally allowing (and likely even benefitting) from parameter sharing! That is, while each vehicle’s policy network (controller) may justifiably want to assume independent weights for heterogeneity, the inference modules could and likely should share weights, since these modules all ask the same question, e.g., “based on what your neighbors did recently, how would you classify their intention?” for the first module. More to this point, one could even argue that the policy network (controller) could also use parameter sharing without drastically restricting performance/generality…
There again, I am confused by the overall premise of the work, which drastically weakens its contribution. It seems to me that the paper relies on “perceived”/”stated” requirements of the proposed solution (no “centralization” during training, no parameters sharing) that I am not convinced have reasons to be. This distracts the reader from appreciating the actual contribution of the work, i.e., the proposed planning framework, which does not really need/builds upon any of these requirements.

Results then compare the proposed approach and its ablation variants to standard baselines (MAPPO, QMIX, and iPPO), but does so in random training scenarios instead of freezing the models and running them through a rigorous, fixed set of randomized experiments. As a result, I am not convinced by the results, where many of the curves also seem to overlap and where algorithms were tested on potentially widely different scenarios. This is not the way algorithms are/should be compared in our field, and I believe a more rigorous testing phase is necessary.

Finally, I also have major concerns with the choice of design for the incentive inference modules, as learning to reconstruct/predict the full raw state seems like a rather flawed process: since each agent’s state also contains information about neighboring agents, learning to simply L1-reconstruct/predict such state may not allow agents to really focus on the most important/key portions of it. As a result, agents may often learn to grossly reconstruct/predict states, which may involve “small” yet very impactful mistakes, leading to very poor incentive learning and downstream policies. This is a common issue in many (single-agent) model-based learning methods that rely on similar encoder-decoder structure, and I believe that the fact that this paper considers a multi-agent problem only makes this issue more important. I also believe that this aspect of the work may be the one that, once properly addressed, could lead to the largest performance improvement.

**Quality Of The Limitations Section:**

Limitations are addressed clearly

**Questions For Rebuttal:**

1. Have authors tried their proposed framework with weight sharing? If so, how do the performance compare to the proposed approach without weight sharing? What are the potential reasons behind these differences, in your opinion?

2. How about testing the proposed framework with a centralized critic (in the controller), and/or a mixing network (also in the controller)? Nothing in the proposed approach actually limits such implementation, and this could also help shed light into the potential advantages of extending the agents’ state with incentive information.

3. Can authors present results of a standard statistical test between the performance results of the different baselines? Current confidence intervals often overlap, thus lowering the confidence the reader can have in the significance of these differences.

4. More importantly: can authors present results of a proper/rigorous testing phase, where frozen models are compared on the same set of randomized experiments, with random seeds fixed? This should also include the same statistical tests for completeness.

5. Can authors discuss the choice of reconstructing a very high-dimensional raw state in the incentive inference modules, and the advantages/limitations of such an approach? Are there other options there, and/or potential directions from the (still very early) field of model-based MARL?

**Robotics Focus:**

Relevant but unlikely to deploy to hardware in near future

**Summary Of Paper:**

This paper focuses on path planning for autonomous vehicles, by training agents (individual vehicles) to learn the driving styles (termed “incentives”) of neighboring vehicles, to both make short-term path predictions about other agents and finally reason about their own path decisions. The work is demonstrated in simulation on two different traffic scenarios (each with two levels of difficulty), where it is compared to three standard algorithms (QMIX, MAPPO, and independent PPO (iPPO)) as well as with a few ablation versions of the proposed approach.

**Summary Of Recommendation:**

While this paper considers a relevant problem, which it approaches along an interesting and novel axis, I worry that the current writing is still built upon shaky premises and inaccurate statements that weaken its story. Some of the key design considerations are also debatable and may merit to be more thoroughly explained and/or investigated for further performance improvement. Results are remain questionable, as no actual testing phase was carried out and algorithms were only tested via their trend at the end of the training phase, thus on potentially widely different scenarios. Performance differences also remain rather marginal. All in all, while I believe this work has a lot of potential, I also think that it needs more time to be polished, and I sadly cannot recommend it for publication at CORL 2023 in its present form.

EDIT: Authors have made significant improvements to the paper, and after rebuttal I believe it is now ready for publication.

---

> ### Author Response · Authors · 2023-08-11
> **Response to Reviewer qQsj [Part 1]**
>
> We would like to express our gratitude to the reviewer for dedicating their valuable time and effort towards evaluating our manuscript, which has allowed us to strengthen the manuscript. We deeply appreciate the insightful feedback provided, and we have thoroughly responded to reviewer's inquiries in the responses provided below.
>
> ## Response to Weaknesses
> > **Weakness 1.1:** In the introduction, authors state that existing MARL algorithms can be classified as either centralized (e.g., QMIX or MAPPO) or distributed (e.g., iPPO). However, this does not seem to be what is generally accepted in the field, as far as I know. “Centralized” approaches like QMIX/MAPPO only rely on information centralization during training (e.g., mixing network, or centralized critic) but remain entirely decentralized at runtime. Instead, one generally refers to QMIX/MAPPO as “centralized training, decentralized execution (CTDE)...”
>
> **Response to Weakness 1.1:** Thank you for pointing out this ambiguity! The confusion is in line 36 of the paper. We have clarified this confusion by rephrasing this paragraph to:
>
> *Most popular MARL algorithms follow the centralized training, decentralized execution (CTDE) paradigm that use centralized credit assignment networks or communicate information among agents, resulting in better convergence properties. However, centralized MARL algorithms are not practical for autonomous driving scenarios, as implementing a central controller for vehicles distributed across a large-scale traffic scenario is infeasible. An alternative paradigm that performs decentralized training, decentralized execution (DTDE), of which distributed MARL algorithms are a part of, allow agents to train and act independently, but training and convergence of such methods are challenging due to the well-documented non-stationary problem.*

---

> > ### Author Response · Authors · 2023-08-11
> > **Response to Reviewer qQsj [Part 3]**
> >
> > > **Weakness 2.1:** Authors then follow up by stating that distributed approaches like iPPO often rely on tricks like parameters sharing, which cannot apply in AV path planning (but without substantiating why).
> >
> > **Response to Weakness 2.1:** Thank you for your comment. We believe there is some misunderstanding. iPlan indeed builds on IPPO, but IPPO does not do weight sharing. Distributed approaches which rely on tricks like parameters sharing is the approach proposed in [1].
> >
> > Considering the inherent diversity in autonomous driving, sharing policy network parameters across varied agents can compromise the system's collective performance. For example, a policy designed for managing a truck isn't apt for a sedan. Experiences gained from autonomous vehicles accustomed to urban traffic might not be directly applicable to those primarily navigating rural roads. As such, our algorithms are crafted without weight sharing during training, ensuring nuanced control in multi-agent autonomous vehicle contexts.
> >
> > [1] Zhang, Kaiqing, et al. "Fully decentralized multi-agent reinforcement learning with networked agents." International Conference on Machine Learning. PMLR, 2018.
> >
> > > **Weakness 2.2:** However, the whole approach then described in this work, which consists of three modules (behavioral incentive inference, instant incentive inference, and controller/policy), has two of these modules completely symmetric and agent-agnostic, thus naturally allowing (and likely even benefitting) from parameter sharing! That is, while each vehicle’s policy network (controller) may justifiably want to assume independent weights for heterogeneity, the inference modules could and likely should share weights, since these modules all ask the same question, e.g., “based on what your neighbors did recently, how would you classify their intention?” for the first module. More to this point, one could even argue that the policy network (controller) could also use parameter sharing without drastically restricting performance/generality…
> >
> > **Response to Weakness 2.2:** Thank you for raising a very important problem to discussion. It is true that the inference module of each agent in iPLAN framework performs the same high-level task that "based on what your neighbors did recently, how would you classify their intention?" However, ***performing the same task does not necessarily equal to parameter sharing,*** because:
> > 1. Our problem setting is partially observable and each agent's observation set is not fixed. The observed states are determined by relative positions f entities, not absolute positions, so that their values are influenced by the ego agent's current location. Given our experiment's intricate environments and agent interactions, the observation sets are high-dimensional. This complexity diminishes the likelihood of convergence to a shared weight across inference modules and controllers due to the vast variations in their input.
> > 2. We set our problem on a heterogeneous multi-agent system. As [2] highlights, agents with varying kinematics typically develop unique policies when managed by DTDE MARL algorithms like IPPO. Since iPLAN maintains individual controller and inference networks for each agent, the diversity in their kinematics ensures their policy networks' evolution paths diverge, despite undertaking identical tasks.
> > 3. Even in environments where agent kinematics are mostly homogenous, like Heterogeneous Highway, minor policy deviations can have significant implications for the inference module. This is because an agent's controller module is tightly interwoven with its inference module. Without explicit weight sharing among inference modules, disparities in their policies and observations can lead to significant evolutionary differences in their inference frameworks.
> >
> > [2] Bettini, Matteo, Ajay Shankar, and Amanda Prorok. "System Neural Diversity: Measuring Behavioral Heterogeneity in Multi-Agent Learning." arXiv preprint arXiv:2305.02128 (2023).

---

> > > ### Author Response · Authors · 2023-08-11
> > > **Response to Reviewer qQsj [Part 4]**
> > >
> > > ***Empirical evidence for Weakness 2.2:*** To support our statement about weight sharing,  we perform quantitative analysis among the network similarity among agents. Specifically, we calculated the Pearson correlation coefficient for the controller, behavioral, and instant incentive inference networks across two scenarios. This metric offers a clear indicator of network similarity.
> > >
> > > Here are the Pearson correlation coefficients among agents, showing the similarity between different agents' controller, behavioral and instant incentive inference networks in the chaotic scenario of Heterogeneous Highway.
> > > | Controller | Agent 1 | Agent 2 | Agent 3 | Agent 4 | Agent 5 |
> > > | ----------- | ------- | ------- | ------- | ------- | ------- |
> > > | **Agent 1** | 1.0000  | 0.8303  | 0.8302  | 0.8299  | 0.8317  |
> > > | **Agent 2** | 0.8303  | 1.0000  | 0.8290  | 0.8298  | 0.8300  |
> > > | **Agent 3** | 0.8302  | 0.8290  | 1.0000  | 0.8292  | 0.8297  |
> > > | **Agent 4** | 0.8299  | 0.8298  | 0.8292  | 1.0000  | 0.8291  |
> > > | **Agent 5** | 0.8317  | 0.8300   | 0.8297  | 0.8291  | 1.0000  |
> > >
> > > | Behavior | Agent 1 | Agent 2 | Agent 3 | Agent 4 | Agent 5 |
> > > | ----------- | ------- | ------- | ------- | ------- | ------- |
> > > | **Agent 1** | 1.0000e+00 | -2.7426e-04| -9.4707e-03|  6.3779e-03 |  -2.6243e-03  |
> > > | **Agent 2** | -2.7426e-04 |  1.0000e+00 |-2.3460e-03 |  1.1409e-02 |  1.0865e-02  |
> > > | **Agent 3** | -9.4707e-03 | -2.3460e-03 | 1.0000e+00 | 7.6034e-03 |  4.7705e-03  |
> > > | **Agent 4** | 6.3779e-03 | 1.1409e-02 | 7.6034e-03 | 1.0000e+00 | 4.4854e-03  |
> > > | **Agent 5** | -2.6243e-03 |  1.0865e-02| 4.7705e-03| 4.4854e-03 |  1.0000e+00  |
> > >
> > >
> > > | Instant | Agent 1 | Agent 2 | Agent 3 | Agent 4 | Agent 5 |
> > > | ----------- | ------- | ------- | ------- | ------- | ------- |
> > > | **Agent 1** | 1.0000e+00 | 1.0412e-02 | 2.9131e-03 | -6.5048e-04 | -3.5517e-03 |
> > > | **Agent 2** | 1.0412e-02 | 1.0000e+00 | -3.3802e-03 | -2.4014e-04 |  4.1945e-03 |
> > > | **Agent 3** | 2.9131e-03 | -3.3802e-03 | 1.0000e+00 | -2.1079e-03 | 2.3084e-03|
> > > | **Agent 4** | -6.5048e-04 | -2.4014e-04 | -2.1079e-03 | 1.0000e+00 |  5.6384e-03 |
> > > | **Agent 5** | -3.5517e-03 | 4.1945e-03 | 2.3084e-03 | 5.6384e-03 | 1.0000e+00 |
> > >
> > > Here are the Pearson correlation coefficients among agents, showing the similarity between different agents' controller, behavioral and instant incentive inference networks in the hard scenario of Non-Cooperative Navigation.
> > > |  Controller  | Agent 1 | Agent 2 | Agent 3 |
> > > | ----------- | ------- | ------- | ------- |
> > > | **Agent 1** | 1.0000 |  0.3914 | 0.3872  |
> > > | **Agent 2** | 0.3914 | 1.0000 | 0.3868  |
> > > | **Agent 3** | 0.3872 | 0.3868 | 1.0000  |
> > >
> > > |  Behavior  | Agent 1 | Agent 2 | Agent 3 |
> > > | ----------- | ------- | ------- | ------- |
> > > | **Agent 1** | 1.0000 |  0.1379 | 0.1691  |
> > > | **Agent 2** | 0.1379 | 1.0000 | 0.1769  |
> > > | **Agent 3** | 0.1691 | 0.1769 | 1.0000  |
> > >
> > > |  Instant  | Agent 1 | Agent 2 | Agent 3 |
> > > | ----------- | ------- | ------- | ------- |
> > > | **Agent 1** | 1.0000 |  0.0045 | 0.0147  |
> > > | **Agent 2** | 0.0045 | 1.0000 | 0.0123  |
> > > | **Agent 3** | 0.0147 | 0.0123 | 1.0000  |
> > >
> > > Our findings suggest that in a highly heterogeneous multi-agent system, such as Non-Cooperative Navigation, agents are more inclined to cultivate distinct controller policies. In contrast, within a more homogeneous multi-agent system, like Heterogeneous Highway, agents demonstrate more uniformity in their controller policies, hinting at a potential inclination towards weight sharing. However, the distinctness observed in both behavioral and instant inference modules across both environments underscores their lack of weight sharing among agents.
> > >
> > > In addition to this heterogeneity assessment, our ablation study addressing the weight-sharing variant of iPLAN can be found in our response to Question 1. Kindly refer to that section for a comprehensive understanding of the weight sharing experiments.

---

> > > > ### Author Response · Authors · 2023-08-11
> > > > **Response to Reviewer qQsj [Part 5]**
> > > >
> > > > > **Weakness 2.3:** There again, I am confused by the overall premise of the work, which drastically weakens its contribution. It seems to me that the paper relies on “perceived”/”stated” requirements of the proposed solution (no “centralization” during training, no parameters sharing) that I am not convinced have reasons to be. This distracts the reader from appreciating the actual contribution of the work, i.e., the proposed planning framework, which does not really need/builds upon any of these requirements.
> > > >
> > > > **Response to Weakness 2.3:** Thank you for pointing out the ambiguous part in our problem setting. Based on our reponse regarding centralization (in Weakness 1) and parameters sharing (in Weakness 2.1 and 2.2), ***our contribution made in this paper is:***
> > > >
> > > > *We propose a novel intent-aware planning algorithm within the "decentralized training, decentralized execution (DTDE)" MARL framework. This compels agents to optimize performance based solely on their observations and inferred opponent incentives, eliminating the need for common decentralized MARL approaches such as centralized credit assignment or parameter sharing.*
> > > >
> > > > We perform experiments and find that our algorithm, iPLAN, allows multi-agent system achieve better performance in complex and heterogeneous environments, like autonomous driving in dense and heterogeneous traffic environments.
> > > >
> > > > > **Weakness 3:** Results then compare the proposed approach and its ablation variants to standard baselines (MAPPO, QMIX, and iPPO), but does so in random training scenarios instead of freezing the models and running them through a rigorous, fixed set of randomized experiments. As a result, I am not convinced by the results, where many of the curves also seem to overlap and where algorithms were tested on potentially widely different scenarios. This is not the way algorithms are/should be compared in our field, and I believe a more rigorous testing phase is necessary.
> > > >
> > > > **Response to Weakness 3:** Thank you for point out the problem in our testing phase. We have updated our testing phases and perform our testing phase over frozen models using a fixed set of random seeds (59582679, 763887655 and 312261940). Results are updated in Appendix G.4 of our paper and provided in our response to Question 3 and 4. We kindly direct you there for a thorough review.
> > > >
> > > > Regarding the scenarios, it is crucial to understand that our training and execution landscapes are more aligned than they might appear at first glance, and they are not "widely different", because:
> > > >
> > > > 1. The singular variation between episode scenarios across both environments lies in the initial positioning of agents. Every other parameter remains invariant. This means that all MARL approaches are trained and evaluated under consistent scenarios, with only the stochasticity in position initialization varying.
> > > > 2. Comparing the easy and hard scenarios of Non-Cooperative Navigation, the hard scenario simply introduces an additional random agent. Analogously, in the Heterogeneous Highway, the distinction between the mild and chaotic scenarios is the composition of behavior-driven vehicles. The chaotic scenario incorporates a higher proportion of aggressive vehicles. However, it is essential to note that the overall vehicle count and their density distribution remain consistent across both scenarios. This approach ensures that our experiments are conducted within controlled variable settings.
> > > >
> > > > For an in-depth exploration of our scenario design across both environments, please refer to Section 5.1 and Appendix A of our manuscript.

---

> > > > > ### Author Response · Authors · 2023-08-11
> > > > > **Response to Reviewer qQsj [Part 6]**
> > > > >
> > > > > > **Weakness 4:** Finally, I also have major concerns with the choice of design for the incentive inference modules, as learning to reconstruct/predict the full raw state seems like a rather flawed process: since each agent’s state also contains information about neighboring agents, learning to simply L1-reconstruct/predict such state may not allow agents to really focus on the most important/key portions of it. As a result, agents may often learn to grossly reconstruct/predict states, which may involve “small” yet very impactful mistakes, leading to very poor incentive learning and downstream policies. This is a common issue in many (single-agent) model-based learning methods that rely on similar encoder-decoder structure, and I believe that the fact that this paper considers a multi-agent problem only makes this issue more important. I also believe that this aspect of the work may be the one that, once properly addressed, could lead to the largest performance improvement.
> > > > >
> > > > > **Response to Weakness 4:** Thank you for raising this concern in our loss function. This is a very good point. While developing our incentive inference modules, we took cues from the methodology presented in [3]. This work, which introduces a similar encoder-decoder framework in their trajectory prediction module, presents an interesting divergence: they advocate for L-p norm in their paper, but their practical implementation leans on the L-1 norm. Their empirical results demonstrate that the L-1 norm is effective in their loss function design.
> > > > >
> > > > > To ensure the robustness of our approach and address your concern, we carried out experiments employing the L-2 norm in our loss function design for both inference modules, specifically under the chaotic traffic scenario of the Heterogeneous Highway.
> > > > >
> > > > > Results and discussions are included here:
> > > > >
> > > > > Standard statistical test over reward values generated by 32 testing episodes over frozen models after 950,000 training time steps. (Seed ID = 59582679)
> > > > > |  Algorithm  | Mean | Std |
> > > > > | ----------- | ------- | ------- |
> > > > > | iPLAN | 53.321 | 9.490 |
> > > > > | iPLAN-L2 | 48.182 | 11.921 |
> > > > >
> > > > > The result shows that using L2-norm in the loss function of both behavioral and instant incentive inference modules does not help to improve the performance of iPLAN inference module in practice. More details and discussions regarding experiments about loss function design could be found in Appendix G.1 of our paper.
> > > > >
> > > > > [3] Yoo, Se-Wook, et al. "GIN: Graph-Based Interaction-Aware Constraint Policy Optimization for Autonomous Driving." IEEE Robotics and Automation Letters 8.2 (2022): 464-471.
> > > > >
> > > > > ### Response to Questions for Rebuttal
> > > > >
> > > > > >  **Question 1:** Have authors tried their proposed framework with weight sharing? If so, how do the performance compare to the proposed approach without weight sharing? What are the potential reasons behind these differences, in your opinion?
> > > > >
> > > > > **Response to Question 1:** We appreciate your questions regarding performing weight sharing in our model. We conduct an experiment wherein the behavioral and instant incentive inference modules operate with shared weights under the chaotic traffic scenario of Heterogeneous Highway.
> > > > >
> > > > > Results of the experiments and discussions are included here.
> > > > >
> > > > > Standard statistical test over reward values generated by 32 testing episodes over frozen models when time step = 1,200,000 (Seed ID = 59582679)
> > > > > |  Algorithm  | Mean | Std |
> > > > > | ----------- | ------- | ------- |
> > > > > | iPLAN | 56.540 | 10.141 |
> > > > > | iPLAN-weight-sharing | 52.876 | 11.848 |
> > > > >
> > > > > Results show that performing weight sharing over inference modules degrades the performance, as the margin between the mean of the episodic reward of iPLAN and iPLAN-weight-sharing is large. Upon deeper analysis, it becomes evident that weight sharing among inference modules of varied agents does not enhance outcomes. This is primarily due to the inherent challenges in harmonizing policies within a diverse agent team. As discussed in our response to Weakness 2.2, even subtle disparities in controller policies can lead to significant variances in incentive inference modules. Weight sharing does not rectify this discrepancy. Furthermore, upholding distinct incentive inference modules without resorting to weight sharing effectively manages the innate diversity of the multi-agent system, making the approach more adept for intricate, heterogeneous systems.
> > > > >
> > > > > For a comprehensive understanding of our multi-agent system's heterogeneity and its impact on policy development, we direct you to our detailed explanation in response to Weakness 2.2 and Appendix G.2 in our paper.

---

> > > > > > ### Author Response · Authors · 2023-08-11
> > > > > > **Response to Reviewer qQsj [Part 7]**
> > > > > >
> > > > > > >  **Question 2:** How about testing the proposed framework with a centralized critic (in the controller), and/or a mixing network (also in the controller)? Nothing in the proposed approach actually limits such implementation, and this could also help shed light into the potential advantages of extending the agents’ state with incentive information.
> > > > > >
> > > > > > **Response to Question 2:** Thank you for raising this valuable question. To evaluate the performance of iPLAN module incorporating with centralized critic and a mixing network (in the controller), we perform several experiments over the following alternative approaches in the chaotic traffic scenario of Heterogeneous Highway and compare their results with iPLAN, including:
> > > > > >
> > > > > > 1. iPLAN embedded in a MAPPO framework
> > > > > > 2. iPLAN embedded in a QMIX framework
> > > > > >
> > > > > > Our observations indicate that incorporating iPLAN modules with CTDE approaches, like QMIX and MAPPO, does not help to overcome the failure of centralized credit assigment modules in both CTDE approaches, and distributed version of iPLAN has a better performance in the MARL problem setting with complex, heterogeneous multi-agent system, like Heterogeneous Highway.
> > > > > >
> > > > > > Navigation metrics over QMIX, iPLAN-QMIX (Centralized version of iPLAN based on QMIX), and iPLAN, with the statistical test over results generated by 32 testing episodes over frozen models after 1,200,000 training time steps (Seed ID = 59582679)
> > > > > > |  Algorithm  | Success Rate | Avg. Reward | Avg. Survival Time | Avg. Speed
> > > > > > | ----------- | ------- | ------- | ------- | ------- |
> > > > > > | QMIX | 38.13 $\pm$ 8.37 | 54.29 $\pm$ 3.12 | 46.01 $\pm$ 5.23  | 23.50 $\pm$ 0.30  |
> > > > > > | iPLAN-QMIX | 54.38 $\pm$ 7.79 | 50.46 $\pm$ 3.40 | 64.96 $\pm$ 3.65  | 23.88 $\pm$ 0.19  |
> > > > > > | iPLAN | 64.38 $\pm$ 9.12 | 56.54 $\pm$ 3.51 | 74.92 $\pm$ 4.86  | 21.99 $\pm$ 0.17  |
> > > > > >
> > > > > > Navigation metrics over MAPPO, iPLAN-MAPPO (Centralized version of iPLAN based on MAPPO), and iPLAN, with the statistical test over results generated by 32 testing episodes over frozen models after 1,200,000 training time steps (Seed ID = 59582679)
> > > > > > |  Algorithm  | Success Rate | Avg. Reward | Avg. Survival Time | Avg. Speed
> > > > > > | ----------- | ------- | ------- | ------- | ------- |
> > > > > > | MAPPO | 26.88 $\pm$ 7.06 | 43.70 $\pm$ 3.50 | 44.60 $\pm$ 3.71  | 29.93 $\pm$ 0.02  |
> > > > > > | iPLAN-MAPPO | 23.75 $\pm$ 5.86 | 42.22 $\pm$ 3.09 | 42.20 $\pm$ 3.29 | 29.93 $\pm$ 0.02  |
> > > > > > | iPLAN | 64.38 $\pm$ 9.12 | 56.54 $\pm$ 3.51 | 74.92 $\pm$ 4.86  | 21.99 $\pm$ 0.17  |
> > > > > >
> > > > > > More details could be found in Appendix G.3 of our paper.
> > > > > >
> > > > > > >  **Question 3:** Can authors present results of a standard statistical test between the performance results of the different baselines? Current confidence intervals often overlap, thus lowering the confidence the reader can have in the significance of these differences.
> > > > > >
> > > > > > **Response to Question 3:** Thank you for your suggestion for presenting our results between baselines with standard statistical tests. Here are standard statistical test (including mean, standard deviation and p-value) between result of different baselines (iPLAN, IPPO, QMIX and MAPPO). All results are performed over frozen models with 3 fixed random seeds by 32 testing episodes.
> > > > > >
> > > > > > Standard statistical tests over reward values generated by 32 testing episodes over frozen models after 1,450,000 training time steps (Seed ID: 59582679, 763887655 and 312261940)
> > > > > > |  Algorithm  | Mean | Std | p-value |
> > > > > > | ----------- | ------- | ------- | ------- |
> > > > > > | iPLAN | 53.514 | 11.252 | -  |
> > > > > > | QMIX | 47.695 | 13.002 | $1.162 \times 10^{-3}$   |
> > > > > > | MAPPO | 42.903 | 9.401 | $3.160 \times 10^{-11}$  |
> > > > > > | IPPO | 49.502 | 10.207 | $1.080 \times 10^{-2}$  |
> > > > > >
> > > > > > p-values (<0.05) suggest results are statistically significant.
> > > > > >
> > > > > > More details could be found in Appendix G.4 of our paper.

---

> > > > > > > ### Author Response · Authors · 2023-08-11
> > > > > > > **Response to Reviewer qQsj [Part 8]**
> > > > > > >
> > > > > > > >  **Question 4:** More importantly: can authors present results of a proper/rigorous testing phase, where frozen models are compared on the same set of randomized experiments, with random seeds fixed? This should also include the same statistical tests for completeness.
> > > > > > > >
> > > > > > > **Response to Question 4:** We appreciate your insights in performing a more rigorous testing phase in our experiment. We have included our new testing results performed over frozen model along with standard statistical tests over iPLAN and three baselines (QMIX, MAPPO, and IPPO), both in our paper and our response here.
> > > > > > >
> > > > > > > All experiments are performed under chaotic traffic scenario of Heterogeneous Highway. The random seed ID we used are: 59582679, 763887655 and 312261940. Standard statistical test performed over 32 testing episodes with model frozen. The p-value is compuated by comparing the testing result of iPLAN and baslines.
> > > > > > >
> > > > > > > Please check our updated experiment results in Appendix G.4 of our paper and see if current experiment results are rigorous.
> > > > > > >
> > > > > > > Standard statistical tests over reward values generated by 32 testing episodes over frozen models after 200,000 training time steps.
> > > > > > > |  Algorithm  | Mean | Std | p-value |
> > > > > > > | ----------- | ------- | ------- | ------- |
> > > > > > > | iPLAN | 45.866 |  11.863 | -  |
> > > > > > > | QMIX | 44.700 | 12.022 | 0.5017  |
> > > > > > > | MAPPO | 43.317 | 10.990 | 0.1261  |
> > > > > > > | IPPO | 46.463 | 11.700 | 0.7271  |
> > > > > > >
> > > > > > > Standard statistical tests over reward values generated by 32 testing episodes over frozen models after 500,000 training time steps.
> > > > > > > |  Algorithm  | Mean | Std | p-value |
> > > > > > > | ----------- | ------- | ------- | ------- |
> > > > > > > | iPLAN | 50.568 | 10.379 | -  |
> > > > > > > | QMIX | 44.455 | 13.008 | $4.357 \times 10^{-4}$   |
> > > > > > > | MAPPO | 41.855 | 10.785 | $5.150 \times 10^{-8}$  |
> > > > > > > | IPPO | 53.986 | 12.287 | $3.968 \times 10^{-2}$  |
> > > > > > >
> > > > > > > Standard statistical tests over reward values generated by 32 testing episodes over frozen models after 1,000,000 training time steps.
> > > > > > > |  Algorithm  | Mean | Std | p-value |
> > > > > > > | ----------- | ------- | ------- | ------- |
> > > > > > > | iPLAN | 54.445 | 11.718 | -  |
> > > > > > > | QMIX | 46.579 | 13.555 | $2.978 \times 10^{-5}$   |
> > > > > > > | MAPPO | 41.911 | 10.192 | $2.681 \times 10^{-13}$  |
> > > > > > > | IPPO | 50.836 | 12.023 | $3.751 \times 10^{-2}$  |
> > > > > > >
> > > > > > > Standard statistical tests over reward values generated by 32 testing episodes over frozen models after 1,450,000 training time steps.
> > > > > > > |  Algorithm  | Mean | Std | p-value |
> > > > > > > | ----------- | ------- | ------- | ------- |
> > > > > > > | iPLAN | 53.514 | 11.252 | -  |
> > > > > > > | QMIX | 47.695 | 13.002 | $1.162 \times 10^{-3}$   |
> > > > > > > | MAPPO | 42.903 | 9.401 | $3.160 \times 10^{-11}$  |
> > > > > > > | IPPO | 49.502 | 10.207 | $1.080 \times 10^{-2}$  |
> > > > > > >
> > > > > > > Standard statistical test results show that iPLAN outperforms all baselines included in our paper (QMIX, MAPPO, and IPPO) in terms of episodic reward. p-values (<0.05) suggest results are statistically significant.
> > > > > > >
> > > > > > > More details could be found in Appendix G.4 of our paper.

---

> > > > > > > > ### Author Response · Authors · 2023-08-11
> > > > > > > > **Response to Reviewer qQsj [Part 9]**
> > > > > > > >
> > > > > > > > >  **Question 5:** Can authors discuss the choice of reconstructing a very high-dimensional raw state in the incentive inference modules, and the advantages/limitations of such an approach? Are there other options there, and/or potential directions from the (still very early) field of model-based MARL?
> > > > > > > > >
> > > > > > > > **Response to Question 5:**  Thank you for raising this insightful question. The choice of reconstructing a high-dimensional raw state in the incentive inference modules was made based on several considerations:
> > > > > > > >
> > > > > > > > *  **Richness of Information:** High-dimensional raw states capture a wealth of information about the environment and the agents. By reconstructing such states, our model can access a comprehensive view of the scenario, which can be crucial for accurately inferring the incentives of agents, especially in complex and dynamic environments.
> > > > > > > > *  **Flexibility:** By working with raw states, our model remains flexible and can be applied to a variety of scenarios without needing domain-specific feature engineering. This makes the approach more generalizable across different MARL environments.
> > > > > > > > *  **Learning Hierarchies:** High-dimensional states can allow the model to learn hierarchies of features, where lower layers capture basic spatial-temporal patterns and higher layers capture more abstract concepts related to agent behavior and intentions.
> > > > > > > >
> > > > > > > > The limitations of this approach include:
> > > > > > > > * **Computational Overhead:** Handling high-dimensional data can be computationally intensive, potentially slowing down training and inference.
> > > > > > > > * **Risk of Overfitting:** With more dimensions, there's a risk that the model might overfit to the training data, especially if not enough diverse data is available.
> > > > > > > >
> > > > > > > > Regarding alternative approaches, the field of model-based MARL offers some promising directions:
> > > > > > > >
> > > > > > > > *  **Compact State Representations:** Instead of using raw states, one could employ techniques to derive compact state representations that capture the most salient features of the environment. Techniques like autoencoders or variational methods could be used to reduce dimensionality while preserving crucial information.
> > > > > > > > *  **Predictive Models:** Model-based MARL often involves learning a model of the environment. Such models could be used to predict future states based on current actions and states. This predictive capability can aid in incentive inference by anticipating future scenarios.
> > > > > > > > *  **Incorporating Domain Knowledge:** In some cases, domain-specific knowledge can be integrated into the model to guide the inference process. For instance, in traffic scenarios, known traffic rules and typical driver behaviors can be used to refine incentive predictions.
> > > > > > > >
> > > > > > > > In conclusion, while our choice of reconstructing high-dimensional raw states offers several benefits, we acknowledge the potential limitations and are excited about the prospects of integrating ideas from model-based MARL to enhance our approach further. We appreciate the opportunity to explore this aspect and will consider it in our future iterations and research.

---

> > > > > > > > > ### Comment · Reviewer_qQsj · 2023-08-14
> > > > > > > > > **Response to authors**
> > > > > > > > >
> > > > > > > > > Dear Authors,
> > > > > > > > >
> > > > > > > > > Thank you very much for your exhaustive rebuttal, which took me a few days to get through! I understand that you now would like to position your work in the DTDE framework, but am very unconvinced by this approach: DTDE is usually selected for online learning on-robot (on-vehicle), where the difficulties associated with estimating/sharing network weights/updates is justified by the fact that information just cannot be centralized. However, your problem setting is not one where online learning is feasible anyways (for obvious safety reasons), as evidenced by the fact that you train and test in a centralized simulator. The actual deployment of your approach would most likely follow the same setting (centralized training in simulation, followed by decentralized deployment). Therefore, the choice of DTDE does not make sense to me, and I stand by my earlier comments (centralization is not an issue in your case *during training*, and I don't understand the "need" to get away from it; this seems like a very subjective and unsubstantiated choice, which is even directly broken in the paper's results...).
> > > > > > > > >
> > > > > > > > > Second, while I appreciated the new results with weight sharing, my comments were a lot more nuanced than what seems to have transpired: I never said that you should share all weights, since as you (and I) mentioned, heterogeneity would likely be an issue there. However, I mentioned that some portions of the network could (and likely should) use weight sharing, especially earlier stages of feature extractions that are and should be similar among vehicles to boost cooperation. The new results, which show that fully sharing weights doesn't work as well are not a surprise to me but do not address my comment. This also goes in the same vein of this very subjective choice of getting away from the CTDE framework without justifying the need to do so convincingly.
> > > > > > > > >
> > > > > > > > > Finally, I still feel that the issue of learning to predict a full state of a multi-agent system is a little too optimistic and has a lot of room for agents to learn to over-simplify state predictions and thus hurt performances. In my opinion, the choice of a specific (L1, L2, or Lp) norm there does not matter as much as the choice of what should be learned and how.
> > > > > > > > > More generally, I feel that this work has a lot of potential but is currently not fully investigated, and as such I will need to keep my current rating as I do not see this work as being ready for publication just yet.

---

> > > > > > > > > > ### Author Response · Authors · 2023-08-14
> > > > > > > > > > **Response to Reviewer qQsj follow-up [Part 1]**
> > > > > > > > > >
> > > > > > > > > > Thank you for your feedback. We are happy to engage further!
> > > > > > > > > >
> > > > > > > > > >
> > > > > > > > > > >**Question1:** Thank you very much for your exhaustive rebuttal, which took me a few days to get through! I understand that you now would like to position your work in the DTDE framework, but am very unconvinced by this approach: DTDE is usually selected for online learning on-robot (on-vehicle), where the difficulties associated with estimating/sharing network weights/updates is justified by the fact that information just cannot be centralized. However, your problem setting is not one where online learning is feasible anyways (for obvious safety reasons), as evidenced by the fact that you train and test in a centralized simulator. The actual deployment of your approach would most likely follow the same setting (centralized training in simulation, followed by decentralized deployment). Therefore, the choice of DTDE does not make sense to me, and I stand by my earlier comments (centralization is not an issue in your case during training, and I don't understand the "need" to get away from it; this seems like a very subjective and unsubstantiated choice, which is even directly broken in the paper's results...).
> > > > > > > > > >
> > > > > > > > > > We understand your concerns regarding the positioning of our work within the DTDE framework, especially given the nature of our problem setting. Our primary intention is *not* to advocate for DTDE over CTDE based on subjective preferences. Instead, we aim to **empirically explore** and understand the performance dynamics of both paradigms in the context of heterogeneous autonomous driving. Our empirical results, both in the original and revised versions of the paper, consistently indicate that **DTDE offers superior performance in terms of average reward** when compared to CTDE in our specific setting. We invite you to revisit our results section for a detailed comparison.
> > > > > > > > > >
> > > > > > > > > > While we acknowledge that centralization might not pose significant challenges during the training phase, our findings suggest that DTDE provides certain advantages in the decentralized deployment phase, which is crucial for real-world applications. We concur that the field of Multi-Agent Reinforcement Learning (MARL) is still evolving, and **it's premature to assert a definitive stance on the superiority of one paradigm over another**. Our work aims to contribute to this ongoing discourse by presenting empirical evidence, rather than making absolute claims.
> > > > > > > > > >
> > > > > > > > > > Additionally, we are confused about what you mean by “centralized simulator” in this context. A simulator is only responsible for providing the environment and centralization refers to the algorithm. The simulators we have chosen can be both “centralized” or “decentralized” depending on how we set up the environment and algorithm. To be specific, the simulator we have chosen is “decentralized”, by your choice of definition.
> > > > > > > > > >
> > > > > > > > > > Lastly, our work contributes to expanding our knowledge of practical applications of DTDE. Decentralized training of agent policies represents a frontier yet to be fully explored in the realm of Multi-Agent Reinforcement Learning (MARL)[1]. The desire for distributed or decentralized training lies in its potential for parallelism, robustness, flexibility, and scalability — attributes that are critical for the task of autonomous driving. Such a decentralized approach becomes especially pivotal in scenarios characterized by a multitude of agents, dynamic and evolving environments [2], or when agents are required to adapt to diverse team configurations throughout their operational lifespan [3]. Through our research, we aim to shed light on the practical implications and advantages of DTDE, further enriching the discourse in this domain.
> > > > > > > > > >
> > > > > > > > > > In conclusion, our exploration into DTDE vs. CTDE is driven by empirical findings, the benefits and advantages of DTDE, and a desire to contribute to the broader understanding of MARL paradigms in the context of autonomous driving. We appreciate your feedback and hope that our clarifications address your concerns.

---

> > > > > > > > > > > ### Author Response · Authors · 2023-08-14
> > > > > > > > > > > **Response to Reviewer qQsj follow-up [Part 2]**
> > > > > > > > > > >
> > > > > > > > > > > >**Question2:** Second, while I appreciated the new results with weight sharing, my comments were a lot more nuanced than what seems to have transpired: I never said that you should share all weights, since as you (and I) mentioned, heterogeneity would likely be an issue there. However, I mentioned that some portions of the network could (and likely should) use weight sharing, especially earlier stages of feature extractions that are and should be similar among vehicles to boost cooperation. The new results, which show that fully sharing weights doesn't work as well are not a surprise to me but do not address my comment. This also goes in the same vein of this very subjective choice of getting away from the CTDE framework without justifying the need to do so convincingly.
> > > > > > > > > > >
> > > > > > > > > > > To clarify, we do **not** perform full weight sharing. We conducted separate experiments where we shared weights in the inference modules and in the controller. But as we show, iPlan still outperforms partial weight sharing in these settings. Please refer to our weight-sharing results in response to Q1 above. Our results clearly indicate **weight sharing degrades performance** compared to when weights are not shared.
> > > > > > > > > > >
> > > > > > > > > > > What you mentioned in the third paragraph of your review, quoted exactly, was:
> > > > > > > > > > > >>"That is, while each vehicle’s policy network (controller) may justifiably want to assume independent weights for heterogeneity, **the inference modules could and likely should share weights**, since these modules all ask the same question, e.g., “based on what your neighbors did recently, how would you classify their intention?” for the first module."
> > > > > > > > > > >
> > > > > > > > > > > Your question, quoted exactly, was
> > > > > > > > > > > >>"Have authors tried their proposed framework with weight sharing? If so, how do the performance compare to the proposed approach without weight sharing? What are the potential reasons behind these differences, in your opinion?”.
> > > > > > > > > > >
> > > > > > > > > > > We have answered your question exactly and to the point and we have done so with partial weight sharing which you were asking for.
> > > > > > > > > > >
> > > > > > > > > > >
> > > > > > > > > > > >**Question3:** Finally, I still feel that the issue of learning to predict a full state of a multi-agent system is a little too optimistic and has a lot of room for agents to learn to over-simplify state predictions and thus hurt performances. In my opinion, the choice of a specific (L1, L2, or Lp) norm there does not matter as much as the choice of what should be learned and how. More generally, I feel that this work has a lot of potential but is currently not fully investigated, and as such I will need to keep my current rating as I do not see this work as being ready for publication just yet.
> > > > > > > > > > >
> > > > > > > > > > > Thanks for the discussion. We agree that there could be room for alternative approaches that may or may not work better in practice. While this is definitely an interesting direction for future work, we strongly believe this is **not a “weakness”**, rather, a separate point of discussion on its own and merits its own discussion.
> > > > > > > > > > >
> > > > > > > > > > > More specifically, just because there may exist better ways of learning the representation does not mean what we have done is weak, as clearly demonstrated by our results.
> > > > > > > > > > >
> > > > > > > > > > > In [4, 5], when performing a similar behavior modeling task as we did, like predicting a full state of a multi-agent system from the latent representation with encoder-decoder, the authors are using a similar setting as we did in the driving scene, like Driving (2D), agents could observe their own positions and other agent’s position at each time-step (Check Append B.3 of [4]). In Driving (CARLA) (Append B.4 of [4]), at each time step, the ego agent observes both the opponent’s and its own position, directional velocity, and angular velocity. All of these are quite similar to our work.
> > > > > > > > > > > ___
> > > > > > > > > > >
> > > > > > > > > > > [1] Wang, Caroline, Ishan Durugkar, Elad Liebman, and Peter Stone. "DM2: Decentralized Multi-Agent Reinforcement Learning via Distribution Matching." (2023).
> > > > > > > > > > >
> > > > > > > > > > >
> > > > > > > > > > > [2] Marinescu, Andrei, Ivana Dusparic, and Siobhán Clarke. "Prediction-based multi-agent reinforcement learning in inherently non-stationary environments." ACM Transactions on Autonomous and Adaptive Systems (TAAS) 12.2 (2017): 1-23.
> > > > > > > > > > >
> > > > > > > > > > > [3] Thrun, Sebastian. "Lifelong learning algorithms." Learning to learn. Boston, MA: Springer US, 1998. 181-209.
> > > > > > > > > > >
> > > > > > > > > > > [4] Xie, Annie, et al. "Learning latent representations to influence multi-agent interaction." Conference on robot learning. PMLR, 2021.
> > > > > > > > > > >
> > > > > > > > > > > [5] Wang, Woodrow Zhouyuan, et al. "Influencing towards stable multi-agent interactions." Conference on robot learning. PMLR, 2022.

---

> > > > > > > > > > > > ### Comment · Reviewer_qQsj · 2023-08-14
> > > > > > > > > > > > **Response to authors**
> > > > > > > > > > > >
> > > > > > > > > > > > Dear Authors,
> > > > > > > > > > > >
> > > > > > > > > > > > I understand your points about the CTDE/DTDE debate in your answer here, but still do not understand/agree with what is written in the paper: "However, centralized MARL algorithms are not practical for autonomous driving scenarios, as implementing a central controller for vehicles distributed across a large-scale traffic scenario is infeasible." This is just not true, and is not what is happening in CTDE. CTDE methods do not have a "central controller," they only centralize information during training to stabilize and speed up convergence to decentralized policies. In particular, CTDE approaches have been shown to train tens/hundreds of agents, so scalability/centralization is usually not an issue there. This should be corrected in the paper.
> > > > > > > > > > > >
> > > > > > > > > > > > More broadly, my issue is that the choice of DTDE in your work seems to really only be a choice of "weight sharing" vs "no weight sharing": agents in CTDE and DTDE both learn decentralized policies, the only difference is that you propose that agents learn fully independent, decentralized policies. This does not make sense to me, as this would then mean that there is either 1) no cooperation among agents, since all agents have learned independently from each other, or (more likely) 2) that agents have learned to collaborate with other independent agents, but only for a (rather narrow) subset of agents (the ones they trained alongside of). This is my main worry with the DTDE use in your approach: how would agents learn to predict/react to agents they have never seen during training, if all agents have a completely different policy?
> > > > > > > > > > > > In summary, I worry that the proposed approach is rather limited to policies seen from other agents during training, by construction, which sounds like a big limitation in practice. This is one of the key limitations addressed by weight sharing in the CTDE framework, where sharing weights helps agents constrain their policies to a smaller, common area of the weight space, often helping with generalization and/or scalability (by reducing the variance of perceived agent behaviors).
> > > > > > > > > > > >
> > > > > > > > > > > > Second, thanks a lot for the clarification about the weight sharing experiments! I apologize about missing this important aspect of your additional results.
> > > > > > > > > > > > That being said, I wonder if the loss of robustness I just discussed above may be the key trade-off there: while weight sharing may reduce overall performances, by constraining inference modules to be similar despite inner differences among agents, maybe it also ensures agents all share a common ground on how they make inferences about the system, that is more likely to generalize to new agents they have never seen during training? This may be something worth investigating. Nevertheless, I agree that weight sharing does not help improve performance, and if raw performance is the only objective then going for independent policies indeed seems better.
> > > > > > > > > > > >
> > > > > > > > > > > > For your third point, I did not see the decoder in [4] learning to output the next state of the system (only the predicted reward), but I agree that the one in [5] does. However, since [5] learns both reward and future state prediction, it is unclear whether state prediction alone would be potent enough to hold, and this is my main worry in this work as well.
> > > > > > > > > > > > That being said, given the limitation I discussed above (most likely limited generalizability beyond the decentralized policies seen at training), it still seems brittle to me to learn to predict solely the full multi-agent state. I agree that the results presented are promising, but I still am not sure that the approach has been investigated thoroughly enough to convince me yet. I will keep my weak reject rating for now, as I believe this work needs more time to be developed into a strong paper, but I will also be involved in the discussion with the other reviewers to reach a consensus.

---

> > ### Author Response · Authors · 2023-08-12
> > **Response to Reviewer qQsj [Part 2]**
> >
> > >  **Weakness 1.2:** ...a (CTDE) paradigm that could be well-suited to many AV scenarios since training is often done in simulation (where centralization is not an issue), while offering the desired purely decentralized execution capabilities and scalability at deployment time. Therefore, I was first confused by this statement and premise.
> >
> > **Response to Weakness 1.2:** We understand your concern. We frame our problem within a distributed setting, where the multi-agent system exhibits heterogeneity and intricate behavioral patterns. This context is inherently suited for fully distributed MARL algorithms over CTDE MARL algorithms. We explain in more detail below.
> >
> > The inherent centralized credit assignment module in CTDE MARL algorithms can encounter difficulties in effectively distributing credits over heterogeneous agents during training under our problem setting. Moreover, as the multi-agent system escalates, algorithms with centralized credit assignment mechanisms tend to degrade in performance due to challenges in accurately allocating credit among a burgeoning set of agents. Through this perspective, distributed MARL algorithms generally exhibit superior results, especially in expansive multi-agent systems.
> >
> > To give us more insight, we have conducted experiments integrating our iPLAN module with CTDE MARL algorithms, including QMIX and MAPPO. Detailed results can be found below. Our observations indicate that incorporating iPLAN modules with CTDE approaches, like QMIX and MAPPO, does not help to overcome the failure of centralized credit assignment modules in both CTDE approaches, and distributed version of iPLAN has a better performance in the MARL problem setting with complex, heterogeneous multi-agent system, like Heterogeneous Highway.
> >
> > Navigation metrics over QMIX, iPLAN-QMIX (Centralized version of iPLAN based on QMIX), and iPLAN, with statistical tests over results generated by 32 testing episodes over frozen models when training time step = 1,200,000 (Seed ID = 59582679)
> > |  Algorithm  | Success Rate | Avg. Reward | Avg. Survival Time | Avg. Speed
> > | ----------- | ------- | ------- | ------- | ------- |
> > | QMIX | 38.13 $\pm$ 8.37 | 54.29 $\pm$ 3.12 | 46.01 $\pm$ 5.23  | 23.50 $\pm$ 0.30  |
> > | iPLAN-QMIX | 54.38 $\pm$ 7.79 | 50.46 $\pm$ 3.40 | 64.96 $\pm$ 3.65  | 23.88 $\pm$ 0.19  |
> > | iPLAN | 64.38 $\pm$ 9.12 | 56.54 $\pm$ 3.51 | 74.92 $\pm$ 4.86  | 21.99 $\pm$ 0.17  |
> >
> > Navigation metrics over MAPPO, iPLAN-MAPPO (Centralized version of iPLAN combining MAPPO), and iPLAN, with statistical tests over results generated by 32 testing episodes over frozen models when training time step = 1,200,000 (Seed ID = 59582679)
> > |  Algorithm  | Success Rate | Avg. Reward | Avg. Survival Time | Avg. Speed
> > | ----------- | ------- | ------- | ------- | ------- |
> > | MAPPO | 26.88 $\pm$ 7.06 | 43.70 $\pm$ 3.50 | 44.60 $\pm$ 3.71  | 29.93 $\pm$ 0.02  |
> > | iPLAN-MAPPO | 23.75 $\pm$ 5.86 | 42.22 $\pm$ 3.09 | 42.20 $\pm$ 3.29 | 29.93 $\pm$ 0.02  |
> > | iPLAN | 64.38 $\pm$ 9.12 | 56.54 $\pm$ 3.51 | 74.92 $\pm$ 4.86  | 21.99 $\pm$ 0.17  |
> >
> > We provide more experiment results regarding using iPLAN inference module in CTDE approaches in our response to Question 2. More details could be found in Appendix G.3 in our paper.

---

> ### Author Response · Authors · 2023-08-13
> **Gentle Reminder to Reviewer qQsj**
>
> Dear Reviewer qQsj ,
>
> We appreciate the time and effort you've dedicated to reviewing our work. Your insights have been extremely useful, and we have taken care to address each concern raised. Should there be any additional points of discussion, we would be grateful for the opportunity to engage further.
>
> In light of the revisions and clarifications, we kindly and humbly request you to reconsider the evaluation of our work.
>
> Regards,
> Authors

---

> ### Author Response · Authors · 2023-08-14
> **Summary of response to Reviewer qQsj**
>
> We have fixed the problematic statement you found in the paper. Revised paper attached.
>
> Summarizing our discussion so far, we have **addressed almost all of your concerns**--including evaluation protocol, weight sharing, and statistical significance. One remaining point is related to the utilization of CTDE vs DTDE.
>
> We respectfully **disagree** with the reviewer that DTDE doesn't makes sense. This is a well motivated setting in several papers in similar applications [1,2]. We would like to re-iterate that our paper is **not** trying to resolve the debate between the CTDE vs DTDE frameworks. They both are well motivated in literature and have their own advantages depending upon the problem context. In this work, we consider the DTDE framework and have shown it's advantages.  In that regard, we believe our paper does **not** lack strength and given that we addressed all but one of your concerns, we kindly request you to reflect that in your evaluation of our work.
> ___
> Having said that, we would like to address your final concern as well, which is:
>
> > **How do agents generalize in our DTDE paradigm, which is apparently achievable in CTDE through weight sharing?**
>
> With due respect, it seems you are making several strong and unclear assumptions about MARL, weight sharing, and generalizability:
>
> 1. **CTDE agents ALSO cannot generalize to unseen agents they have not trained with**. Weight sharing does **not** help CTDE to generalize, rather, it enables CTDE to stabalize training better than DTDE leading to better convergence properties [3]. So your point that CTDE *generalizes* better than DTDE through weight sharing is **incorrect**.
>
> 2. **Addressing generalizability is NOT the focus of this paper**. Generalizability is a central problem, not only in MARL, but more broadly in learning theory. Other techniques, like population-based training algorithms or opponent modeling techniques [4], are common approaches to achieve some degree of generalization to previously unseen agents (see overcooked[5, 6, 7]/Hanabi[8]). These methods do not use CTDE style updates, as the notion of a global value function itself doesn't make sense in a setting where teammates policies could change at test time.
>
> [1] Zhang, Kaiqing, et al. "Fully decentralized multi-agent reinforcement learning with networked agents." International Conference on Machine Learning. PMLR, 2018.
>
> [2] Li, Wenhao, et al. "F2a2: Flexible fully-decentralized approximate actor-critic for cooperative multi-agent reinforcement learning." arXiv preprint arXiv:2004.11145 (2020).
>
> [3] Wong, Annie, Thomas Bäck, Anna V. Kononova, and Aske Plaat. "Deep multiagent reinforcement learning: Challenges and directions." Artificial Intelligence Review 56, no. 6 (2023): 5023-5056.
>
> [4] Albrecht, Stefano V., and Peter Stone. "Autonomous agents modelling other agents: A comprehensive survey and open problems." Artificial Intelligence 258 (2018): 66-95.
>
> [5] Ribeiro, João G., et al. "Assisting unknown teammates in unknown tasks: Ad hoc teamwork under partial observability." arXiv preprint arXiv:2201.03538 (2022).
>
> [6] Zhao, Rui, et al. "Maximum entropy population-based training for zero-shot human-ai coordination." Proceedings of the AAAI Conference on Artificial Intelligence. Vol. 37. No. 5. 2023.
>
> [7] Lauffer, Niklas, et al. "Who Needs to Know? Minimal Knowledge for Optimal Coordination." International Conference on Machine Learning. PMLR, 2023.
>
> [8] Muglich, Darius, et al. "Equivariant networks for zero-shot coordination." Advances in Neural Information Processing Systems 35 (2022): 6410-6423.

---

> > ### Comment · Reviewer_qQsj · 2023-08-15
> > **Response to Summary**
> >
> > Dear Authors,
> >
> > Thanks for fixing this mistake in the introduction, but your new text is still incorrect: CTDE methods do not suffer from heterogeneity, only weight sharing may tend to do so. For example, QMIX is the most popular CTDE method for credit assignment and has been specifically designed and demonstrated on heterogeneous tasks (many of the SMAC benchmarks are heterogeneous, and in general QMIX aims at learning independent policies without weight sharing).
> > There is still a lot of confusion around these central terms in the paper that worry me: CTDE is not the same as weight sharing, CTDE is not learning/using a centralized controller, DTDE is not the same as learning independent policies (DTDE just lets agents learn in a decentralized manner, without explicit centralization of experiences; DTDE can use weight sharing, but then decentralized consensus methods are needed to allow agents to share weights during training), etc.
> >
> > More importantly, I never said that the CTDE framework is better. I was merely defending it from inaccurate accusations, but I honestly do not believe it is "the superior way", do not worry. I believe this stems from a misunderstanding of my initial review.
> > From the start, I mentioned that your paper was awkward to read for me, as it seemed to build on an inaccurate premise where you tried to state without substance that a CTDE approach would not work for traffic scenarios, to justify your choice of learning independent policies (let's call this IPL, independent policy learning). I disagreed and still disagree with this statement. I do not understand why your work has to build up from overstatements, instead of just saying "we choose to do ILP" without having to "justify" this design choice with inaccuracies. As you said more recently, both approaches are likely to work, and your choice of the ILP is merely a design choice, not THE correct choice, and you do not need to motivate it this way in the intro (like is still done in the most recent version of the paper). This is one of my main issues and it has never been addressed.
> >
> > Now, thinking more about the generalizability, I once again agree with your points but did not feel heard: you are right that the scope of your paper is not to "solve generalizability" in MARL. I am not asking for that of course. But I feel that your paper is currently trying to hide/downplay its actual limitations in terms of generalizability that stem from your choice of ILP. Again, this would not be an issue if you were willing to actually reflect on these aspects and offer a more nuanced presentation in your manuscript. My issues were not ones of opinion/belief, they were about actually properly stating and discussing facts, without the need/desire to claim that your approach is "the best", that CTDE/ILP is "the superior choice", or that your approach is "foolproof" (not said, but somewhat implied by the lack of a clear discussion of its key limitations, like the generalizability discussion I offered above).
> >
> > Your approach works well. It is built on a design choice of ILP that you should not justify by saying CTDE would not work, but instead by discussing honestly their pros and cons with accurate facts (CTDE could very well work on heterogeneous systems, this is not a good argument). Most of all, you then need to properly discuss the potential limitations of your work, especially those that stem from this design choice. If I understand well, other reviewers have also expressed similar concerns about the need to discuss the advantages and limitations of some of your design choices (e.g., choice to separate the intent modules, etc.).
> >
> > You have addressed a lot of my more technical comments, but my fundamental ones are not fixed yet. I cannot "raise my score without advocating for acceptance", since as you well know raising my score would be a "weak accept", which is advocating for acceptance. This is just silly :-)
> > I would and will recommend acceptance of papers that have a solid contribution (which I believe you have), with good results (which I believe you have), but most of all that are written in a way where these points come out through a coherent and **accurate narrative** (which I do not believe is the case yet), i.e., without overstatements or over-simplifications. In my honest opinion, I believe that this last part is mostly a writing exercise, like I mentioned from the beginning, but I worry that it may take a little more than a few days as it may require a deeper reflection in which you accurately and convincingly address the points I made above. But do not worry, I also love to be proven wrong and will be reading your updated text until the end of the discussion phase, so I can offer a fair opinion of your work during the subsequent discussion phase.

---

> > > ### Author Response · Authors · 2023-08-15
> > > **Corrected narrative in Introduction, added Discussion, and discussed Limitations in revised paper**
> > >
> > > Dear Reviewer qQsj,
> > >
> > > It took a while, but we finally understood what you were trying to say. And we agree - the narrative is not correct. We should definitely not go about justifying DTDE by demoting CTDE. That is incorrect and we thank you for helping us realize this, even if it took a while!
> > >
> > > We are happy you acknowledge that we have **solid contributions with good results**. We can both agree to disagree on which factors we believe to be essential for good science (and therefore publication) :-) For example, we completely agree that good writing is essential ! But does it outweigh novelty, contributions, and results? We feel otherwise. But we accept that you think otherwise ! To each their own.
> > >
> > > That being said, now that **all reviewers have more or less agreed on the contributions, results, and clarity**, we have begun addressing the final concern about the narrative and have uploaded a new draft. Specifically, we have **corrected the narrative**, **discussed design choices** and **limitations**, and have re-written the following sections:
> > >
> > > - Section 1 (introduction)
> > > - Section 5.4 (discussion)
> > > - Section 6 (Limitations)
> > >
> > > To help readability, instead of coloring the entire sections blue, we colored the *section titles* as blue, indicating that these sections have been completely re-written.
> > >
> > > Please acknowledge if this a large step in the correct direction.
> > >
> > > Thanks !

---

> > > > ### Comment · Reviewer_qQsj · 2023-08-16
> > > > **Authors response**
> > > >
> > > > Dear Authors,
> > > >
> > > > Thanks a lot for your hard work with the rewriting of these key portions of the paper! A good approach with good results is sadly nothing without a proper narrative and clear writing to support/convey it, and papers that contain inaccuracies should not be allowed to be published (and cited) even if the approach they bring forth is good/effective. I certainly think that we can both agree on that, despite our potential weightage of writing vs idea. The good news is that I believe your paper now has both :-)
> > > >
> > > > The new text is near perfect in addressing my earlier concerns: the last thing I saw is that you refer to QMIX/MAPPO as "centralized MARL approaches", which is not the right terminology. They are certainly CTDE approaches, but "centralized MARL" refers to something different (this term is often used to discuss one learning agent outputting a high-dimensional policy for other agents' actions) and is misleading there. I would recommend you replace the mention of "centralized MARL approach" by "CTDE MARL approach" when discussing these baselines in the Introduction and other relevant places (you actually already did that in section 5.4, which is perfect). I also appreciated the improved limitations discussion in the conclusion.
> > > >
> > > > Given these changes, and the new results and discussions about weight sharing you brought forward and incorporated since my initial review, I will be happy to bump my rating to a weak accept (once the system allows us to do so).

---

> > > > > ### Author Response · Authors · 2023-08-16
> > > > > **Thank you very much !**
> > > > >
> > > > > We appreciate your efforts in improving our paper. Sorry it took a while to understand your core points :-)
> > > > >
> > > > > And yes, we agree that inaccuracies in the narrative should be corrected, and we appreciate your help in that. Relatedly, should we similarly distinguish between "decentralized MARL" and "DTDE"? In other words, can they be equated. We think no, but we'd like to hear your thoughts on this.
> > > > >
> > > > > In any case, we have changed all instances of "centralized MARL" and "decentralized MARL" to "CTDE" and "DTDE", respectively.
> > > > >
> > > > > Let us know if that works !

---

> > > > > > ### Comment · Reviewer_qQsj · 2023-08-16
> > > > > > **Authors response**
> > > > > >
> > > > > > Sounds great for CTDE! For DTDE, although decentralized MARL and DTDE are technically two separate things (many people say CTDE is decentralized MARL, since the ultimate goal is to learn decentralized policies), I think that the places you'd like to use it for (IPPO and your approach) are warranted. The only thing that is important to mention at least once is that they are DTDE and do not share weights (which isn't the same, as many DTDE methods rely on decentralized consensus to still enact weight sharing, one of the big challenges of the DTDE framework).
> > > > > >
> > > > > > I would recommend to add this mention for iPLAN clearly in the introduction (you mention information-sharing, but clarifying once this includes weight sharing would be useful), and also mention this particular detail for IPPO when you introduce it in section 5 (lines 282-283, you already say it does not centralized training, but stating no weight sharing would make it crystal clear).
> > > > > >
> > > > > > After this is done, simply using CTDE/DTDE is completely fair and clear in my opinion.

---

> > > > > > > ### Author Response · Authors · 2023-08-16
> > > > > > > **Done, thanks !**
> > > > > > >
> > > > > > > Done!

---

### Official Review · Reviewer_htJq · 2023-07-21

**Confidence:** 4
**Originality:** Good
**Technical Quality:** Good
**Clarity Of Presentation:** Fair
**Impact:** 4

**Recommendation:**

Weak Accept: I recommend accepting the paper, but will not argue for my recommendation if the majority of other reviewers have a different opinion.

**Review:**

# Strengths

## Originality

The paper proposes an interesting idea: where agents in a heterogenous MARL environment learn opponent models. Given that there is lots of work on driver modeling for autonomous driving, this is a promising approach for their domain. I would be curious to see how this handles human driving behavior.

## Significance
The underlying idea has promise and so I think there is some upside for a high-impact paper.

## Quality

Although there is some room for improvement in presentation, the experiments seem to be a reasonable test of the method.

# Weaknesses

## Clarity

[major] I had trouble following the overall presentation of the inference training method. I think that a more clear example of the method and edits to Algorithm 1 would help the reader follow and/or replicate the method.

[minor] the typesetting in lines 171-182 is a bit strange due to the inline math.

[major] I had trouble understanding the practical difference between the behavioral and instant incentive. Maybe a better description of the method would help, or a more clear caption. E.g., consider rephrasing the sentence "The instance incentive inference... [uses data] .. to infer other vehicles instance incentives." It would be helpful to make it extra clear what an example of the instant incentives is.

[major] there's not enough description of the different environments in the main paper, I recommend restructuring the paper so you can include a paragraph describing the environments in more detail

**Quality Of The Limitations Section:**

Additional details required

**Questions For Rebuttal:**

## Can you describe an example of what the behavior and instant incentives are?

I had trouble following this and would benefit from an example. I think this something like this would be good to include in the paper to make it more accessible.

## Please address the clarity issues I raised in my review.

I think the idea presented is promising, but I think that some issues with the presentation need to be addressed before I recommend acceptance.

## Are there any experiments you could run to test the value of using hierarchical incentives?

I'm curious how this would compare to a simpler method that infers a single set of incentives for each agent.

**Robotics Focus:**

Highly relevant to robotics but no hardware experiments

**Summary Of Paper:**

The paper presents a MARL approach for driving where each agent acts based on a combination of 'behavioral' incentives, which govern preferences over driving style, and 'instant' incentives, which govern the local preferences over avoiding collisions. Agents learn to predict the incentives of other agents while they learn to act. They evaluate in multi-agent driving domains and demonstrate performance improvements.

**Summary Of Recommendation:**

Overall, I think the idea has promise, but I worry that the presentation/evaluation is not quite where is should be. If the authors can address the presentation issues, I think the paper should be accepted. I think a more clear demonstration on a more complex domain could also improve the paper's impact.

Update after rebuttal/discussion: I appreciate the author’s efforts during the discussion to address my concerns about presentation and to better scope the claimed contributions. As a result, I am happy to raise my score and confidence level in the result.

---

> ### Author Response · Authors · 2023-08-11
> **Response to Reviewer htJq [Part 1]**
>
> We would like to express our gratitude to the reviewer for dedicating their valuable time and effort towards evaluating our manuscript, which has allowed us to strengthen the manuscript. We deeply appreciate the insightful feedback provided, and we have thoroughly responded to reviewer's inquiries in the responses provided below.
>
> ## Response to Weaknesses
>
> >  **Weakness 1:** [Major] I had trouble following the overall presentation of the inference training method. I think that a more clear example of the method and edits to Algorithm 1 would help the reader follow and/or replicate the method.
>
> **Response to Weakness 1:** Thank you for raising your concern regarding the ambiguity of the inference training in Algorithm 1. We have updated Algorithm 1 to clearly explain the inference and training setup. We also include a flow diagram of iPLAN, as a supplement to Algorithm 1 in Appendix F.2 of our paper.
>
>
> > **Weakness 2:** [Minor] The typesetting in lines 171-182 is a bit strange due to the inline math..
>
> **Response to Weakness 2:** Thanks! We have fixed this. Kindly check now and let us know if it looks good to you!
>
> > **Weakness 3:** [Major] I had trouble understanding the practical difference between the behavioral and instant incentive. Maybe a better description of the method would help, or a more clear caption. E.g., consider rephrasing the sentence "The instance incentive inference... [uses data] .. to infer other vehicles instance incentives." It would be helpful to make it extra clear what an example of the instant incentives is.
>
> **Response to Weakness 3:** Thank you for sharing your insights. We've enriched the description of the incentives, providing a clearer depiction of both behavioral and instant incentives their real-world implications with examples.
>
> ***Behavioral incentive:*** It is ***drivers' long-term driving strategies and habits***, invariably intertwined with their personalities and consistent driving patterns. We postulate that the behavioral incentive remains relatively **constant** over time.
>
> To illustrate, upon extended observation of a particular driver, one might note that aggressive drivers consistently maintain higher average speeds than their more moderate counterparts. Such drivers are also more prone to risky actions in similar conditions than those who drive conservatively. For instance, confronted with a slow-moving vehicle ahead, an aggressive driver might attempt an overtake, while a more cautious driver would likely reduce speed, maintaining a safe distance. The behavioral incentive aims to capture these ingrained driving tendencies and habits. We surmise that this incentive can be ascertained from prolonged observations of a particular driver's patterns, as stated in [1].
>
> ***Instant incentive:*** The instant incentive encapsulates ***drivers' immediate reactions to their surrounding traffic conditions***, including current positions and velocities of ego vehicle's neighboring vehicle. This incentive is **transient**, reflecting the dynamic nature of driving environments.
>
> Since safety is the essential concern in driving, we consider instant incentive is usually related to collision avoidance behaviors. Regardless of driving strategies and habits (aggressive or conservative), almost all drivers will perform the same bahaviors under some specific conditions. For instance, amid dense urban traffic, most drivers, irrespective of their driving tendencies—be it aggressive or cautious—opt for reduced speeds and heightened vigilance, like slowing down to accommodate an approaching vehicle rather than attempting an overtake. In this case, all drivers appear conservative in driving.
>
> Conversely, in lighter traffic settings, such as rural roads or highways during off-peak hours, drivers are prone to adopt higher speeds and might indulge in riskier maneuvers, like overtaking or maintaining a higher speed. In this case, all drivers appear aggressive in driving. The instant incentive encapsulates these immediate, context-sensitive driving decisions. It is presumed that such incentives can be discerned through real-time traffic observations.
>
> Please refer to Section 3 in our paper for our refined illustrations and examples of both behavioral and instant incentives. The updates have been highlighted in blue for your convenience.
>
> [1] Cheung, Ernest, et al. "Identifying driver behaviors using trajectory features for vehicle navigation." 2018 IEEE/RSJ International Conference on Intelligent Robots and Systems (IROS). IEEE, 2018.

---

> > ### Author Response · Authors · 2023-08-11
> > **Response to Reviewer htJq [Part 2]**
> >
> > > **Weakness 4:** There's not enough description of the different environments in the main paper, I recommend restructuring the paper so you can include a paragraph describing the environments in more detail.
> >
> > **Response to Weakness 4:** Thank you, we have restructured the paper accordingly by adding an new Section 5.1 in main paper, describing the details of our environment. We also provide our new Section 5.1 below. Please confirm if it looks better.
> >
> > *Non-Cooperative Navigation is an adaptation of the Cooperative Navigation scenario in the Multi-agent Particle Environment (MPE), This environment involves n agents independently covering n landmarks. Agents aim to choose, reach and remain at landmarks while avoiding conflict. Each agent, at every step, observes other agents' and landmarks' identifiers, positions, and velocities, selects actions from {idle, up, down, left, right}, and gets a reward based on its distance to the closest landmark. Agents face a -5 penalty if a collision happens, earn 10 if reaching a landmark, and win a 100 reward if all agents reach landmarks without conflicts. We span experiments over two scenarios. The easy scenario has 3 controllable agents varying in their sizes and kinematics, and the hard scenario adds an uncontrollable agent taking random actions apart from 3 controllable agents.*
> >
> > *Heterogeneous Highway is our enhanced multi-agent iteration of the Highway-Env's Highway scenario.  It replicates rush-hour traffic on a multi-lane highway with diverse driving behaviors. The MARL-controlled vehicles aim to navigate safely at speeds between 20 and 30 m/s amidst varied traffic. Uncontrollable vehicles fall under three behavior-driven models: Normal, Aggressive, and Conservative, distinguished by risk-taking and general speed. Each agent observes nearby vehicles' ID, position, and velocity, choosing actions from {lane left, idle, lane right, faster, slower}. Rewards are given for collision-free navigation, maintaining speed, and using the rightmost lane. We perform experiments over two scenarios with different compositions of behavior-driven vehicles. The mild scenario has 80% normal, 10% aggressive, and 10% conservative vehicles. The chaotic scenario has 40% normal, 30% aggressive, and 30% conservative vehicles.*
> >
> > ## Response to Questions
> >
> > >  **Question 1:** Can you describe an example of what the behavior and instant incentives are?
> >
> > **Response to Question 1:** Yes, we have elaborated on illustration examples for both behavioral and instant incentives in our response to Weakness 3 and included examples we propose there in Section 3 of our paper. Kindly refer to that section for a comprehensive understanding.
> >
> > >  **Question 2:** Please address the clarity issues I raised in my review.
> >
> > **Response to Question 2:** We appreciate your valuable feedback regarding clarity issues in our paper. We have updated our paper.

---

> > > ### Author Response · Authors · 2023-08-11
> > > **Response to Reviewer htJq [Part 3]**
> > >
> > > >  **Question 3:** Are there any experiments you could run to test the value of using hierarchical incentives? I'm curious how this would compare to a simpler method that infers a single set of incentives for each agent.
> > >
> > > **Response to Question 3:** Yes, we have performed additional experiments on a single set of incentives for each agent. According to the description between line 263 and 278, we name the approach that agents only uses instant incentive inference without behavioral incentive inferences as iPLAN-GAT. Similarly, we name the approach that agents only uses behavioral incentive inference without instant incentive inferences as iPLAN-BM. We have included the results performed by both approaches as a part of our ablation study and presented them in Section 5. Please check Section 5 in our updated submitted version of the paper for more details.
> > >
> > > We also included navigation metrics results generated over IPPO, iPLAN-GAT, iPLAN-BM, and iPLAN under mild and chaotic traffic scenarios of Heterogeneous Highway below.
> > >
> > > Navigation metrics over IPPO, iPLAN-GAT, iPLAN-BM, and iPLAN under mild traffic scenario over 64 tesing episodes performed over frozen models after 2,000,000 training time steps.
> > > |  Algorithm  | Avg. Speed | Avg. Survival Time | Success Rate |
> > > | ----------- | ------- | ------- |  ------- |
> > > | IPPO | 22.63 $\pm$ 0.17 | 66.13 $\pm$ 4.13  | 49.06 $\pm$ 7.35  |
> > > | iPLAN-GAT | 22.05 $\pm$ 0.11 | 75.54 $\pm$ 3.61  | 68.44 $\pm$ 6.64  |
> > > | iPLAN-BM | 22.61 $\pm$ 0.16 | 64.11 $\pm$ 4.28  | 45.63 $\pm$ 6.33  |
> > > | iPLAN | 22.91 $\pm$ 0.15 | 70.56 $\pm$ 3.81  | 68.44 $\pm$ 5.86  |
> > >
> > > Navigation metrics over IPPO, iPLAN-GAT, iPLAN-BM, and iPLAN under chaotic traffic scenario over 64 tesing episodes performed over frozen models after 2,000,000 training time steps.
> > > |  Algorithm  | Avg. Speed | Avg. Survival Time | Success Rate |
> > > | ----------- | ------- | ------- |  ------- |
> > > | IPPO | 22.28 $\pm$ 0.13 | 67.01 $\pm$ 3.64  | 42.50 $\pm$ 7.12  |
> > > | iPLAN-GAT | 20.91 $\pm$ 0.13 | 71.24 $\pm$ 3.83  | 61.88 $\pm$ 6.41  |
> > > | iPLAN-BM | 21.65 $\pm$ 0.28 | 63.20 $\pm$ 3.51  | 35.31 $\pm$ 5.66  |
> > > | iPLAN | 21.61 $\pm$ 0.16 | 76.20 $\pm$ 3.33  | 67.81 $\pm$ 5.91  |
> > >
> > >
> > > >  **Question 4:** I think a more clear demonstration on a more complex domain could also improve the paper's impact.
> > >
> > > **Response to Question 4:** Thank you for highlighting a potential avenue for expanding our research. We have conducted a preliminary experiment where we implemented iPLAN in a more challenging traffic scenario. This advanced setting, which we have termed "chaotic-VH", mirrors the existing traffic distribution of behavior-driven vehicles (Normal: Aggressive: Conservative = 4:3:3) in current chaotic scenario, but with a vehicle density that is twice as dense as our previously studied chaotic scenario. Due to computational constraints during the rebuttal phase, our exploration was limited to a single random seed over 1 million time steps. Despite this, the results of iPLAN evaluated under chaotic-VH are promising.
> > >
> > > We observe that though having a lower episodic reward curve than it used to be in the normal chaotic scenario, iPLAN outperforms MAPPO in the chaotic-VH scenario, in terms of episodic reward, average episode length, and success rate. For a comprehensive examination of the results and ensuing discussion, please refer to Appendix F.1 of our manuscript.
> > >
> > > Navigation metrics over iPLAN and MAPPO over 32 tesing episodes performed over frozen models after 1,000,000 training time steps.
> > > |  Algorithm  | Success Rate | Avg. Reward | Avg. Survival Time | Avg. Speed
> > > | ----------- | ------- | ------- | ------- | ------- |
> > > | iPLAN | 50.63 $\pm$ 9.33 | 41.15 $\pm$ 4.38  | 56.19 $\pm$ 6.62  | 19.77 $\pm$ 0.88|
> > > | MAPPO | 18.13 $\pm$ 4.37 | 32.49 $\pm$ 2.80  | 35.80 $\pm$ 3.42  | 24.02 $\pm$ 0.92|

---

> ### Author Response · Authors · 2023-08-13
> **Gentle Reminder to Reviewer htJq**
>
> Dear Reviewer htJq ,
>
> We appreciate the time and effort you've dedicated to reviewing our work. Your insights have been extremely useful, and we have taken care to address each concern raised. Should there be any additional points of discussion, we would be grateful for the opportunity to engage further.
>
> In light of the revisions and clarifications, we kindly and humbly request you to reconsider the evaluation of our work.
>
> Regards,
> Authors

---

> ### Author Response · Authors · 2023-08-14
> **Gentle Reminder to Reviewer htJq**
>
> Dear Reviewer htJq ,
>
> This is a gentle reminder. We appreciate the time and effort you have dedicated to reviewing our work. Your insights have been extremely useful, and we have taken care to address each concern raised. Should there be any additional points of discussion, we would be grateful for the opportunity to engage further.
>
> In light of the revisions and clarifications, we kindly and humbly request you to reconsider the evaluation of our work.
>
> Regards, Authors

---

> > ### Comment · Reviewer_htJq · 2023-08-14
> > **Thanks the updates (and for your patience)**
> >
> > Thank you for the revisions and clarifications. I believe the paper is more clear now and it is easier for me to understand what is going on with the approach discussed. Additionally, I appreciate moving descriptions of the experimental domains to the main body of the paper as well as making the method more reproducible through pseudocode. I have one follow-up question and two comments.
> >
> > First, can you articulate how your results justify the added complexity of using the behavioral/instant incentive? I can see the experimental results and there is some additional analysis. I would like to hear why the improvements you see between, e.g., IPLAN-GAT and IPLAN justify the additional complexity of IPLAN.
> >
> > Second, I think there is room for improvement in explaining and motivating the hierarchical incentives used here. For example, in describing the instant incentive, you say that collision avoidance becomes more important in heavy traffic. I expect a lot of readers will object to that reasoning because collision avoidance is always important --- it is simply easier to do in light traffic so one can drive faster. I guess that it is helpful because you don't have the right features. I.e., the preference over velocity is a simplification of a much more complicated set of preferences so it is easy for the inference to allow it to adapt.
> >
> > Finally, I do still notice a few typos and grammatical errors --- some additional editing is needed for the camera ready. Perhaps consider using Grammarly or a similar tool to check for missing words or incorrect verb tenses (e.g., l.165 "Behavioral incentive captures these inherent tendencies," should either say "The behavioral incentive captures" or "Behavioral incentives capture").
> >
> > I'll end by saying thank you for your efforts and your patience. I'm sorry that other responsibilities kept me from participating in the discussion phase as much as I would have liked.

---

> > > ### Author Response · Authors · 2023-08-14
> > > **Addressing reviewer htJq's remaining 1 question and 2 comments**
> > >
> > >
> > > > First, can you articulate how your results justify the added complexity of using the behavioral/instant incentive? I can see the experimental results and there is some additional analysis. I would like to hear why the improvements you see between, e.g., IPLAN-GAT and IPLAN justify the additional complexity of IPLAN.
> > >
> > > Thanks for the question! The main contribution of our work is a new algorithm the combines behavior modeling and trajectory prediction to enable efficient and state-of-the-art dencentralized MARL for applications in heterogeneous autonomous driving. Not only would our work greatly benefit the decentralized MARL research community in general, but also does **not** incur high computational complexity compared to the base controller.
> > >
> > > Specifically, we are happy to report that:
> > > - Each of the GAT and BM modules occupy only about **10%** of the size of the core controller (which is basically iPlan without the two incentive modules). We can provide specific parameter sizes if necessary.
> > > - In terms of inference time, both the core controller and the added incentive modules are of the **same order**. Specifically iPlan executes at **0.3646** seconds per timestep per episode. iPlan-GAT executes at **0.2512** seconds per timestep per episode and iPlan-BM executes at **0.0124** seconds per timestep per episode.
> > > - Finally in terms of training time (using the hyperparameters in the paper), iPlan takes **4.897 days** whereas iPlan-GAT takes **4.042** days. Note here, that the training time complexity also results from the complexity of the simulator environment.
> > >
> > > Let us know if you'd like us to analyze complexity in any other way.
> > >
> > > > Second, I think there is room for improvement in explaining and motivating the hierarchical incentives used here.
> > >
> > > We apologize for the confusion. Let us try again.
> > >
> > > Perhaps the most concrete way to explain the two incentives is to first differentiate between the objectives of the two incentives.
> > >
> > >
> > > - **Behavior incentive**:  Given the observations for the previous few seconds, behavior incentive performs high-level decision-making similar to action planning, and asks, "*What's the most likely action of this driver to take next?*". The answer is encoded via $\hat\beta^t_i$. This tells an agent whether it should speed up in empty traffic or slow down in dense traffic. It also is able to recognize conservative drivers and the possible need to overtake. Therefore, this incentive is able to reason between aggressive and conservative drivers.
> > > - **Instant incentive**: Instant incentive then asks "*How should I execute this maneuver/high-level action/plan using my controller so that I'm safe and still on track towards my goal?*". Instant incentive measures classical efficiency metrics defined in robotics literature such as collision avoidance (safety), distance from goal, and smoothness.
> > >
> > > **Toy Example**. Having gained an idea of what each incentive is responsible for, here's a toy example. Suppose Alice is driving behind Bob. Alice is a relatively more assertive and confident driver than Bob, who is driving very slowly. Now, Alice's' *behavior incentive* is tracking both Alice's and Bob's driving for the past few seconds and after observing for a short while will tell her to overtake Bob. At this point, her behavior incentive will inform her *instant incentive*, which will modify her trajectory and show her exactly how (what controls) to execute the overtake maneuver safely, as opposed to having her stuck behind Bob.
> > >
> > > Another way to look at it is that the instant incentive is akin to motion forecasting whereas the behavior incentive is akin to high-level decision making. Then, we can say that the behavior incentive biases the motion forecasting in a behavior-aware manner such that it is better suited for heterogeneous traffic. For evidence, note that in more homogeneous traffic, at 68.44% each, iPlan has similar success rate to iPlan-GAT (no behavior incentive), whereas in chaotic traffic, the success rate drops significantly for iPlan-GAT, compared to iPlan (61.88% versus 67.81%), indicating that you need behavior modeling to survive in more heterogeneous chaotic traffic
> > >
> > > We again apologize for the lack of clarity. Please let us know if this is better, and if so, we will incorporate this into the revised paper.
> > >
> > > We are happy to provide further clarification if needed.
> > >
> > > > Finally, I do still notice a few typos and grammatical errors.
> > >
> > > Thanks! We are currently fixing these. Will update this response once complete.

---

> > > > ### Comment · Reviewer_htJq · 2023-08-15
> > > > **Clarifying my comments on complexity**
> > > >
> > > > Thank you for your response about the memory and compute constraints that the new method provides. However, it doesn't really address my question. I'll try to rephrase it.
> > > >
> > > > Looking at your results, I don't think it will be clear to readers where and how the combined method improves over the ablated versions. My comment about complexity was more about methodological complexity (i.e., more interacting parts of the system, more hyperparameters to tune, more code to maintain, more design choices to make) than computational or memory complexity. In reading through your description of the results, it doesn't clearly make the case to me that the combination is a sufficient improvement to justify a more complicated method.

---

> > > > > ### Author Response · Authors · 2023-08-15
> > > > > **Justifying the combined system**
> > > > >
> > > > > Thanks for the clarification.
> > > > >
> > > > > Kindly note that our main contribution is an effective and efficient novel joint trajectory and intent prediction algorithm using MARL for autonomous driving in heterogeneous traffic. It is a strong novel conceptual contribution supported, not only by strong results, but also by good engineering.
> > > > >
> > > > > In general, it is well-known that training even simple MARL algorithms is hard. Yet, our approach extends MARL for trajectory planning research in autonomous driving to harder domains (heterogeneous traffic) under minimalistic assumptions (decentralized training, no weight sharing, variable agents etc.).
> > > > >
> > > > > Considering that getting decentralized MARL algorithms to even converge effectively in simpler environments is a challenge, the fact that our combined approach not only trains well, but also outperforms many state-of-the art baselines, is a significant achievement, in our opinion.
> > > > >
> > > > > Furthermore, our work opens new directions for further impactful research:
> > > > > - training and evaluation of decenrtalized MARL in heterogeneous environments with mobile robots. Applications include indoor social navigation, delivery robots in crowded grocery stores, restaurants etc.
> > > > > - robo-taxis like Cruise and Waymo are being deployed in geofenced locations in more and more urban areas. As the geofencing increases towards more challenging scenarios, there is a need for effective trajectory planning work for diverse, dense, and heterogenenous traffic.
> > > > > - understanding the theoretical implications and foundations for intent modeling in general MARL research. There is already a ton of work along the lines of opponent modeling etc. We would be excited to see if there are any theoretical insights to be gained along the way of our work.
> > > > >
> > > > > Another aspect that showcases the importance of both modules is the following: note that in more homogeneous traffic, at 68.44% each, iPlan has similar success rate to iPlan-GAT (no behavior incentive), whereas in chaotic traffic, the success rate drops significantly for iPlan-GAT, compared to iPlan (61.88% versus 67.81%). This clearly indicates that indicating that you need the behavior incentive module to survive in more heterogeneous chaotic traffic.
> > > > >
> > > > > In summary, **thinking of our contribution in terms of just the improved percentage points due to two added components is highly reductive**. Our *combined system* is a significant push along the direction of decentralized MARL for autonomous riving in heterogeneous traffic.

---

> > > > > > ### Author Response · Authors · 2023-08-15
> > > > > > **Follow-up to justification of combined system**
> > > > > >
> > > > > > Additionally, to make things easier for users and readers, we can:
> > > > > >
> > > > > > 1. Exactly list out all the extra hyperparameters, how to tune them, and which default values work best out of the box
> > > > > > 2. We can provide extensive documentation (and we have, just not available due to double-blind) for additional code and how to work with it.
> > > > > > 3. We can add tutorials and python notebooks to streamline how the combined system works.
> > > > > > 4. We can provide the exact default values and parameters of all the design choices.
> > > > > > 5. We can add individual config files to separately control each module.
> > > > > >
> > > > > > In summary, we will not hide the fact there is *some* (not a lot) added implementation due to the two modules, but the way we have set up our code and have integrated our system, we do not believe this will be much of a problem for users. The modules are highly compartmentalized in the sense that if users want to tweak, say the behavior incentive module, they can easily do so without touching the instant incentive module.

---

> > > > > > ### Comment · Reviewer_htJq · 2023-08-15
> > > > > > **One of your contributions is the 2-stream incentive model/inference mechanism you propose, it is reasonable to ask you to justify that choice**
> > > > > >
> > > > > > I'm sorry you think I'm being reductive. My comment was intended to say that your current draft expects the reader to connect the dots between your ablation experiments and the value of the two-stream inference mechanism that you claim as a contribution. From my perspective, I'm asking you to justify a design decision that is a central part of one of 3 contributions. I'm familiar with the difficulties of MARL and, as I've stated before, I do think that the approach has a lot of promise for several of the reasons you mentioned.
> > > > > >
> > > > > > To be specific, in your contributions, you claim the following:
> > > > > >
> > > > > > > We provide an explicit representation of agents’ private incentives to their strategies that include
> > > > > > > (i) Behavioral Incentive for long-term planning tied to an agent’s driving behavior or personality
> > > > > > > and (ii) Instant Incentive for short-term collision avoidance related to current traffic state. We
> > > > > > > propose a two-stream incentive inference mechanism that allows agents to infer incentives from
> > > > > > > their opponents and incorporate their inferred information into decision-making.
> > > > > >
> > > > > > If a single incentive performs similarly well, then that devalues the design of a two-stream incentive inference mechanism. If your claim is that this is a novel and sufficient contribution without the two-stage inference method, then you should only claim the contribution you intend to defend.
> > > > > >
> > > > > > It seems like there is a benefit in one setting of the combined module. Please make that benefit more clear to readers and provide some explicit justification for the design choices that you've made. Alternatively, you could reduce your emphasis on the two-stream aspect of the inference and only claim the explicit representation of the agent's private incentives.

---

> > > > > > > ### Author Response · Authors · 2023-08-15
> > > > > > > **We have reduced emphasis on two-stream inference. Added discussion. Updated paper**
> > > > > > >
> > > > > > > Thanks for being specific! We understand the concern much better now.
> > > > > > >
> > > > > > > We have updated the paper by removing the emphasis on two-stream inference from the abstract, contributions (2nd bullet), as well as from certain places in Section 3.
> > > > > > >
> > > > > > > In addition, we have also clarified the benefit of the combined system versus individual inference modules in Section 5.4 (Discussion)
> > > > > > >
> > > > > > > Please let us know if you'd suggest us to clarify/add anything further. Thank you.

---

> > > > > > > > ### Comment · Reviewer_htJq · 2023-08-16
> > > > > > > > **Thank you!**
> > > > > > > >
> > > > > > > > I'm sorry for the misunderstanding and I'm sorry that I was unclear. I think this change makes sense, given the results. I don't see the updated paper (this discussion thread has gotten somewhat long...) but I think those changes will be sufficient.
> > > > > > > >
> > > > > > > > As a final suggestion, perhaps a way to frame the results is to say that you experiment with different ways of representing/inferring the goals of other drivers. You find that this type of structure can be learned when it is there and that it improves the performance of the learning method. However, the overall experimental results are somewhat mixed, and future work can investigate in detail the benefits of different representational choices for incentive models/inference.
> > > > > > > >
> > > > > > > > Thanks for your patience with this process.

---

> > > > > > > > > ### Author Response · Authors · 2023-08-16
> > > > > > > > > **Framing the results - done**
> > > > > > > > >
> > > > > > > > > Thats strange. We will upload it again. Let us know if you still are unable to see the new revised version. Please check the PDF file under "Rebuttal by Paper223 Authors".
> > > > > > > > >
> > > > > > > > > Thank you for all your suggestions to make the paper stronger. We agree that such a framing will help set the tone appropriately for the results -- that sure, our approach works best in heterogeneous scenarios, but maybe a single incentive would suffice for simpler scenarios. We have added the following in Section 6:
> > > > > > > > >
> > > > > > > > > "We explored two incentives to represent and infer the objectives of other drivers to inform the ego-vehicle's motion planning. Our findings indicate that in diverse, dense, and heterogeneous settings, collectively inferring these incentives improves of the learning approach. However, in certain scenarios, such as in more straightforward or mixed conditions, the necessity of dual incentives remains ambiguous *i.e.* it might be that a singular incentive set is adequate. Future research could delve deeper into the advantages of specific representational selections for incentive or inference models across both simple and mixed contexts."

---

> > > > ### Comment · Reviewer_htJq · 2023-08-15
> > > > **Thank you for the example/improved description**
> > > >
> > > > I think the example really helps me to understand what is going on and your paragraph after the example helps substantially as well. I recommend updating the paper to use the distinction between high-level decision-making (i.e., setting subgoals) and motion control (i.e., executing a subgoal). You may want to connect your approach to Task and Motion Planning as a way to explain the approach. See, e.g., https://arxiv.org/abs/2010.01083.

---

> > > > > ### Author Response · Authors · 2023-08-15
> > > > > **Updated paper with the new description**
> > > > >
> > > > > Thank you. We have updated our paper accordingly in Section 3, page 4 and also connected to the TAMP literature - that was a good idea!

---

### Author Response · Authors · 2023-08-11
**General Summary**

We are immensely grateful for the insightful reviews provided by the esteemed reviewers. Their positive feedback and recognition of the novelty and relevance of our work are truly encouraging.

- **Reviewer htJq** acknowledged the ***interesting and promising approach*** of our paper, especially in the domain of autonomous driving. The potential for our work to become a high-impact paper was also highlighted. Notably, R1 stated the paper ***should be accepted*** once the presentation issues they raised were fixed
- **Reviewer qQsj** appreciated the relevance of our problem to the community and ***commended the rigor and readability*** of our methodology, particularly in Section 3.
- **Reviewer vQKv** praised the clarity of our problem setting, the precision of our formalisms, and the thoroughness of our related work section. They also recognized the difficulty of the problem setting and ***commended our achievements in this challenging domain***.

We believe that these positive sentiments underscore the potential and significance of our work. We have have made efforts to address the presentation and evaluation concerns raised by the reviewers. Specifically, we have:

**Experiments:**
1. We perform experiments over single set of incentives as our ablation study (Fig. 2).
2. We perform our experiments over more random seeds (Fig. 2).
3. We perform additional experiments to evaluate our approach over alternatives that allow weight sharing between agents in our module  (Appendix G.2).
4. We perform additional experiments to evaluate our approach over alternatives that incorporate our approach in a CTDE MARL framework like QMIX and MAPPO (Appendix G.3, H.2).
5. We refine our codebase and perform a rigorous testing phase over frozen models and perform standard statistical tests over testing results (Appendix G.4).
6. We explore the possibly of using other loss functions in our inference module (Appendix G.1).
7. We perform experiments to explore using graphical networks in the behavioral incentive inference module (Appendix H.1).
8. We perform experiments to evaluate our approach versus baselines under a more complex scenario (Appendix F.1).

**Text Changes:**
1. We modify the definition of behavioral and instant incentives in Section 3, and add some helpful examples to describe both incentive in real-life (Section 3).
2. We reorganize our paper with more description of environments we used in Section 5.1 of main paper (Section 5.1).
3. We clarify the ablation approaches we use with a more clear name (Section 5).
4. We refine our introduction by providing more clear concepts and definitions regarding centralized/distributed approaches, weight sharing in our paper (Section 1).
5. We correct the improper terminology "reconstruct" with a more suitable word in our paper (Section 4.1)

**Algorithms and Diagrams:**
1. We have modified Algorithm 1 in our paper with more detailed illustration of the training procedure of behavioral and instant incentive inference modules (Algorithm 1).
2. We add a flow diagram of our approach as a supplement to Algorithm 1. (Appendix F.2)
3. We correct the improper terminology "reconstruct" with a more suitable word in our framework (Fig. 1).
4. We add illustration caption to animations visualizing our environments (Attachment).

---

### Author Response · Authors · 2023-08-16
**General Summary of follow up dicussion after author-reviewer discussion period**

Dear Area Chairs,

We appreciate the engaging discussion of our paper with Reviewers htJq, qQsj, and vQKv. All of them raised insightful points that strengthened our paper, improving its clarity, exposition, and rigor.

Since the discussion thread is long, we summarize the discussions following the initial review and our follow-ups.
___
*Reviewer htJq* (**suggested acceptance** after the presentation concerns were addressed):
- All concerns (both initial review and follow up questions) have been **addressed** and **acknowledged**.
- In general finds the paper promising. We have addressed all presentation concerns raised by Reviewer htJq by:
    - revising the main contributions, abstract, parts of Section 3, Section 4, Section 5.4, and Section 6.
- More specifically, after the initial review, Reviewer htJq helped improve our paper's exposition by (1) simplifying the explanation of behavioral and instant incentives, (2) precisely stating our contributions, and (3) framing our results in a way that better aligns with the stated contributions.
- All concerns have been **addressed** and **acknowledged**.
___
*Reviewer qQsj* (raised the recommendation to **Weak Accept**):
- All technical concerns (both initial review and follow up questions) have been **addressed** and **acknowledged**.
- Agrees with all reviewers that paper has good contribution and good results. Primarily had issues with narrative and writing:
    - narrative in introduction (motivate approach without inaccurately demoting CTDE)
    - lack of proper discussion on limitations (generalization, high-dimensional input)
    - Correct terminology for centralized vs CTDE
- We have addressed these by:
    - rewriting the introduction in section 1,
    - adding a section in Section 6 on limitations that includes a discussion on the possible limitations w.r.t generalizability, high-dimensional input states, among other things.
    - We also expanded on the discussion of empirical analysis of centralized and decentralized methods in Section 5.4 along with a justification of design choice of incentives.
    - Fixing all terminology throughout the paper.
- All concerns have been **addressed** and **acknowledged**.
___
*Reviewer vQKv* (raised the recommendation to **Strong Accept**):
- All concerns (initial review only, no follow up concerns) have been **addressed** and **acknowledged**.
- Noted our responses to other reviewers regarding (1) clarity (2) CTDE v.s. DTDE (3) and found them sufficient
- Finds our paper very solid.

---

### Decision · Program_Chairs · 2023-08-30

**Decision:**

Accept (Oral)

**Comment:**

This paper presents a multi-agent RL approach in the context of path planning of autonomous vehicles to learn behavior of neighboring vehicles.

The reviewers have found that the proposed approach is interesting and technically sound. The reviewer concenrs on readability and contribution has been sufficiently resolved during the rebuttal phase.